# Multi-modal contrastive learning adapts to intrinsic dimensions of shared latent variables

Yu Gui[1] Cong Ma[2] Zongming Ma[3]

[1] Department of Statistics and Data Science, University of Pennsylvania
[2] Department of Statistics, University of Chicago
[3] Department of Statistics and Data Science, Yale University
yugui@wharton.upenn.edu, congm@uchicago.edu, zongming.ma@yale.edu

## Abstract

Multi-modal contrastive learning as a self-supervised representation learning technique has achieved great success in foundation model training, such as CLIP [60]. In this paper, we study the theoretical properties of the learned representations from multi-modal contrastive learning beyond linear representations and specific data distributions. Our analysis reveals that, enabled by temperature optimization, multi-modal contrastive learning not only maximizes mutual information between modalities but also adapts to intrinsic dimensions of data, which can be much lower than user-specified dimensions for representation vectors. Experiments on both synthetic and real-world datasets demonstrate the ability of contrastive learning to learn low-dimensional and informative representations, bridging theoretical insights and practical performance.

## 1 Introduction

The growing availability of data sources has enabled interdisciplinary research to leverage *multi-modal* data, which measures each unit from different aspects with various types of information. For example, the availability of data types including images, text, and audio fostered the advancement of cross-modal foundation models in recent years [39, 45, 70, 60]. Another notable example is single-cell multi-omics technologies [71] which utilize multi-modal measurements of individual cells for more informative scientific discovery [4, 29, 21, 50]. This emerging phenomenon raises a key question: *How can one efficiently integrate data from multi-modalities?*

Contrastive language-image pre-training (CLIP) was recently proposed in [60], which utilizes the self-supervised contrastive learning technique to train a vision-language model with unlabeled data and to obtain representations of data from multi-modalities. Mathematically, given observations from two modalities: continuous random vectors $X \in \mathbb{R}^{d_1}$ and $Y \in \mathbb{R}^{d_2}$, the goal of multi-modal contrastive learning is to learn representation maps $f \in \mathcal{H}_X : \mathbb{R}^{d_1} \to \mathbb{R}^d$ and $g \in \mathcal{H}_Y : \mathbb{R}^{d_2} \to \mathbb{R}^d$ that map data to representations supported on $\mathbb{R}^d$, where $d$ is the user-specified output dimension.

With a training set $\{(X_i, Y_i)\}_{i\in[N]}$, CLIP [60] minimizes the infoNCE contrastive loss:

$$\min_{(f,g,\tau)\in\mathcal{H}_X\times\mathcal{H}_Y\times\mathbb{R}_+} \mathcal{L}^N(f,g,\tau) = -\frac{1}{N}\sum_{i\in[N]} \log \frac{\exp\left(\frac{\sigma(f(X_i),g(Y_i))}{\tau}\right)}{N^{-1}\sum_{j\in[N]}\exp\left(\frac{\sigma(f(X_i),g(Y_j))}{\tau}\right)} - \frac{1}{N}\sum_{i\in[N]} \log \frac{\exp\left(\frac{\sigma(f(X_i),g(Y_i))}{\tau}\right)}{N^{-1}\sum_{j\in[N]}\exp\left(\frac{\sigma(f(X_j),g(Y_i))}{\tau}\right)}, \quad (1)$$

39th Conference on Neural Information Processing Systems (NeurIPS 2025).

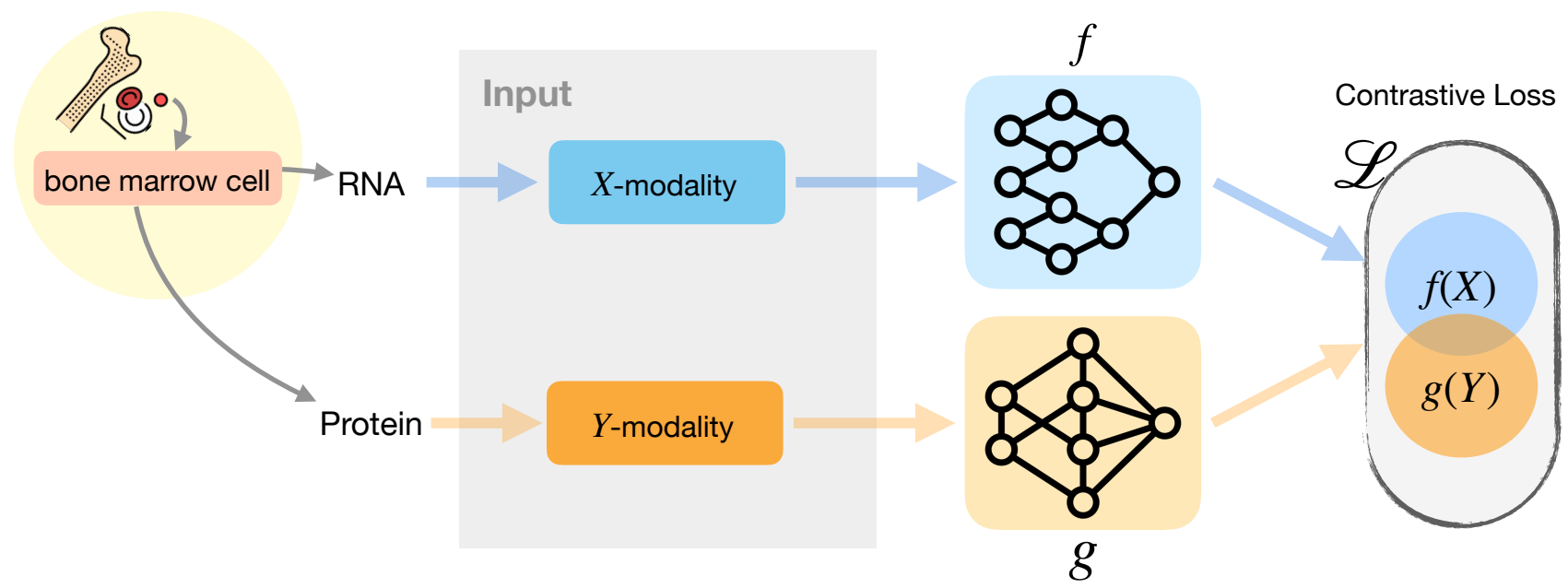

Figure 1: Multi-modal contrastive learning applied to the bone marrow single-cell CITE-seq data.

where $\sigma(\cdot, \cdot)$ is a bivariate similarity measure, with larger values of $\sigma(f(X), g(Y))$ indicating higher similarity between embeddings $f(X)$ and $g(Y)$. Here we take into account the optimization over temperature $\tau$ to align with the practice of CLIP [60]. In Figure 1, we plot the pipeline of CLIP [60] using a bone marrow single-cell CITE-seq dataset [21] for illustration.

CLIP has achieved outstanding performance in zero-shot accuracy for downstream tasks and has inspired a number of follow-up works across domains [17, 40, 81, 63]. The theoretical analysis of CLIP was later initialized in [62], [49], and [9], where the properties of the learned representation and its effect on the downstream accuracy are of interest. However, the distribution of multi-modal data is mainly restricted to factor models with shared latent variables, and the representation map is commonly assumed to be linear, neither of which is close to the actual practice of CLIP. In this paper, we aim to theoretically study the properties of multi-modal contrastive learning with data from general distributions and representation mappings in general function classes.

### 1.1 A motivating example: CLIP adapts to intrinsic dimension

We begin with describing several interesting phenomena arising from using CLIP on a synthetic data. The phenomena motivate our later theoretic studies.

Consider the synthetic setting where $X$ and $Y$ are generated from the following distribution with $k^* < \min\{d_1, d_2\}$:

$$Y_i \overset{i.i.d.}{\sim} \mathcal{N}(0, \mathbf{I}_{d_2}), \quad \xi_i \overset{i.i.d.}{\sim} \mathcal{N}(0, \mathbf{I}_{d_1 - k^*}), \quad X_i = (Y_{i1}, \cdots, Y_{ik^*}, \xi_i^\top)^\top.$$

We set $k^* = 2$, $d_1 = d_2 = 20$, and the output dimension of $f(X)$ and $g(Y)$ is set to be $d = 3$. We set the function class to be 5-layer ReLU neural networks with all middle-layer widths fixed at 50. We use a training set of size 12000 and a separate test set of size 2000. In the remaining part of this paper, we adopt the inner product with population-level normalization as the similarity measure[1], i.e., $\sigma(f(X), g(\widetilde{Y})) = \frac{\langle f(X), g(\widetilde{Y})\rangle}{\mathbb{E}\|f(X)\| \cdot \mathbb{E}\|g(\widetilde{Y})\|}$. In experiments, we leave out a fixed subset of size 2000 of the training set to estimate expected norms in each iteration. In Figure 2, we plot the *out-of-sample* similarities between positive and negative pairs, the change of estimated intrinsic dimension along training (see Section 4 for further details), and temperature $\tau$ along training. We note the following phenomena from the results:

1. For positive pairs, $\sigma(f(\widetilde{X}_i), g(Y_i))$ tends to concentrate around one, while similarities between negative pairs are capped by one;
2. Although the output dimension is 3, representations with intrinsic dimension 2 instead of 3 are preferred by infoNCE loss;
3. The optimized temperature converges to *zero*.

The observed phenomena, especially the intrinsic dimension selection and the convergence of temperature to zero, motivate us to understand multi-modal contrastive learning via CLIP from the theoretical perspective in this work.

[1] In Appendix G.5, we show that it is comparable to cosine similarity in terms of both representation intrinsic dimensions and downstream task performances.

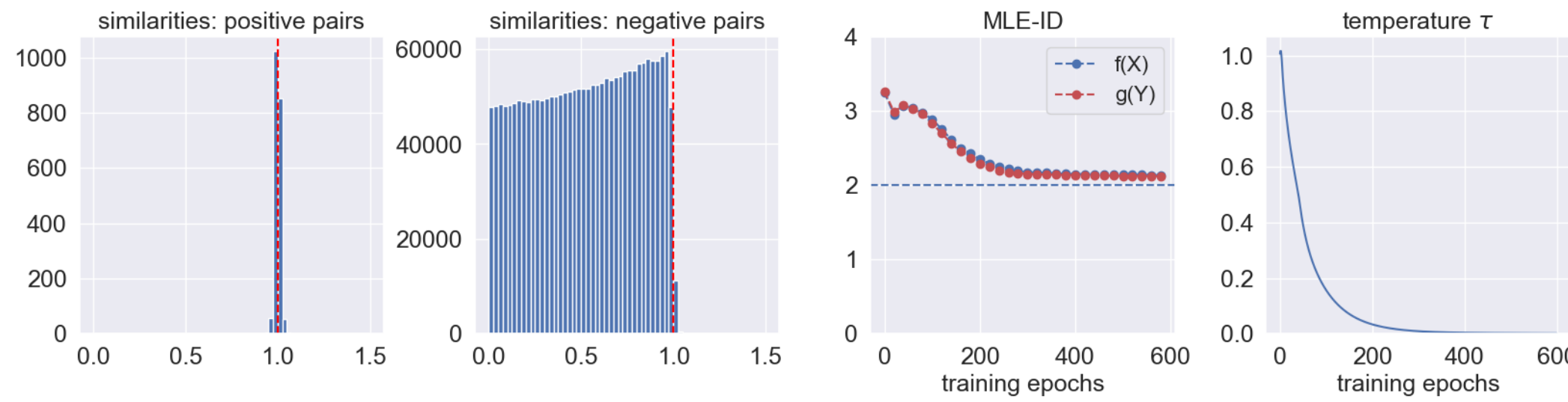

Figure 2: Histograms of out-of-sample similarities, change of intrinsic dimension, and convergence of temperature (linear setting: $k^* = 2$, $d = 3$).

### 1.2 Why existing understanding of multi-modal contrastive learning is insufficient?

Existing work explains the representations learned by contrastive learning from two different perspectives. Here, we elaborate on why neither can fully explain the phenomena observed in Section 1.1.

**Perspective from alignment and uniformity** [79] decomposes the population infoNCE loss $\mathcal{L}$ into alignment and uniformity terms such that $\mathcal{L}(f, g, \tau) = \mathcal{L}_{\mathsf{align}}(f, g, \tau) + \mathcal{L}_{\mathsf{unif}}(f, g, \tau)$. The first term $\mathcal{L}_{\mathsf{align}}$ favors alignment as it is minimized when $f(X) = g(Y)$ almost surely. In addition, the result in [79] implies that among all aligned representations, the second term $\mathcal{L}_{\mathsf{unif}}$ favors uniformity as it is minimized when $f(X)$ is uniformly distributed over the entire output space, which is restricted to be the unit hypersphere with *a user-specified dimension* in [79]. A crucial limitation of this result is that one needs the correct specification of a *true dimension* of data. In other words, if the aligned representation has a smaller dimension compared to the ambient output dimension (which arguably is always the case in practice) or the function class is not expressive enough to produce uniform representations on the entire output space, the theory established in [79] is not applicable.

**Perspective from mutual information maximization** Since the infoNCE loss can be viewed as a variational upper bound of the negative mutual information [53, 59] (see also Lemma 8 below), another line of research understands the global minimizer of infoNCE loss as the mutual information maximizer [76, 24, 75, 73, 84, 41]. Although existing works connect infoNCE loss to its mutual information bound, as is pointed out in [75], solely maximizing the mutual information may result in undesirable representations. For instance, in the example in Section 1.1, as long as $f(X) \approx g(Y)$ almost surely,[2] the mutual information between representations is infinite. Hence, this theory fails to explain why, in the motivating example, $f(X)$ and $g(Y)$ are supported on a 2-dimensional manifold as opposed to a curve or a number of standalone points, since the latter two can also maximize mutual information between representations at infinity. This indicates that infoNCE loss minimization is not simply mutual information maximization and requires a more fine-grained analysis.

### 1.3 Preview of our results and contributions

We now provide a preview of our main theoretical findings that allow us to explain the three phenomena observed in Section 1.1. A formal version will be presented in Section 3.

**Theorem 1** (Informal)**.** *Suppose there exist aligned representations that maximize the shared information between $X$ and $Y$. Then, any "minimizer" $(f, g, \tau)$ of the infoNCE loss satisfies*

1. $\sigma(f(X), g(Y)) =$ *constant almost surely, which caps similarities between negative pairs;*
2. *the intrinsic dimension of shared latent variables in the multi-modal data is exactly captured by* $(f(X), g(Y))$*.*
3. $\tau = 0^+$.

[2]Throughout this paper, "almost surely" refers to almost sure events with respect to the joint distribution of $(X, Y)$, unless otherwise specified. Likewise, "measurable" refers to measurability with respect to the Lebesgue measure, unless stated otherwise.

In addition, we summarize our contributions in this paper as follows.

- We investigate the theoretical properties of *learned representations* from multi-modal contrastive learning and take into account *temperature optimization* in theoretical analysis.
- Based on a precise (variational) decomposition of the infoNCE loss in Lemma 8, we theoretically show that temperature optimization enables multi-modal contrastive learning, which encourages multi-modal representations to be sufficient and to maximize the similarity measure, to also adapt to the intrinsic dimension of data. To be more concrete, different from [79], our theory does not require the existence of aligned representations that are uniform over the entire output space, i.e., no prior knowledge about the true intrinsic dimension is required.
- Moreover, the practical relevance of our theory is supported by empirical findings in real-world datasets, such as single-cell multi-omics and image-text datasets, which demonstrate that in many cases, there are a relatively small number of effective shared features that affect downstream tasks.

**Organization of paper.** The rest of this paper is organized as follows. The ideal properties of representations and the intrinsic dimension are formally defined in Section 2. Main theoretical results are shown in Section 3, which includes a formal statement of our theorem in Section 3.2. Additional related works are summarized in Section 5. Experimental results for both synthetic and `CITE-seq` single-cell datasets are presented in Section 4. Technical proofs, extensions (e.g., connection to sufficient dimension reduction in Appendix E.1), and additional numerical experiments are deferred to the appendix.

## 2 Ideal representations and intrinsic dimension

In Section 2.1, we define two key properties of the ideal representations $(f^\star(X), g^\star(Y))$ when learning from the paired data $(X, Y)$—alignment and maximal mutual information, and in Section 2.2, we define the intrinsic dimension. Throughout the paper, we define $\mathcal{H} = \mathcal{H}_X \times \mathcal{H}_Y$.

### 2.1 Alignment and maximal mutual information

**Alignment and similarity maximization.** Inspired by the seminal work [79] in single-modality contrastive learning, we propose the following notion of *alignment* for learning representations from multiple modalities.

**Definition 1.** We define the set of representation maps that realize alignment and similarity maximization as

$$\mathcal{A}(\mathcal{H}) = \left\{ (f, g) \in \mathcal{H} : \frac{f(X)}{\mathbb{E}\|f(X)\|} = \frac{g(Y)}{\mathbb{E}\|g(Y)\|}, \quad \sigma(f(X), g(Y)) = m_\sigma(f, g) \text{ almost surely} \right\}.$$

Here and after, $m_\sigma(f, g) := \operatorname{ess\,sup}_{X \perp\!\!\!\perp \widetilde{Y}} \sigma(f(X), g(\widetilde{Y}))$ for any $(f, g) \in \mathcal{H}$.

In our motivating example in Section 1.1, there exist representations $f(X) = (X_1, X_2)/\sqrt{X_1^2 + X_2^2} = (Y_1, Y_2)/\sqrt{Y_1^2 + Y_2^2} = g(Y)$ satisfying $(f, g) \in \mathcal{A}(\mathcal{H})$. In this case, $\sigma(f(X), g(Y)) = 1 = m_\sigma(f, g)$ almost surely.

**Maximal mutual information.** A classical notion to measure statistical dependence is mutual information: $I(X; Y) = D_{\mathrm{KL}}(P_{X,Y} \,\|\, P_X \otimes P_Y)$. It is tempting to ask for representations $(f, g)$ such that the mutual information $I(f(X); g(Y))$ is maximized. This is known as the infoMax principle [75, 6]. However, as we have argued in Section 1.2, this vanilla definition of mutual information is not fine-grained enough to compare aligned representations: whenever continuous representations $f(X) = g(Y)$ almost surely, we have $I(f(X); g(Y)) = +\infty$ [33, 74].

To mitigate the deficiency, we adopt a fine-grained *order* for mutual information. For each integer $M \geq 1$, we consider a discretized version[3] $(f_M, g_M)$ of the representations $(f, g)$. As $(f_M, g_M)$

[3] See Section A.2 for precise definitions of the discretizations. Related results on approximating mutual information based on discretization and binning can also be found in [32, 56, 12].

are supported on finitely many points, the mutual information $I(f_M(X); g_M(Y))$ is always finite, albeit $\lim_{M\to+\infty} I(f_M(X); g_M(Y))$ is possibly infinite.

With the discretization in place, we can define the set of representations with maximal mutual information as follows.

**Definition 2** (Maximal mutual information)**.** We define the following set of pairs $(f, g) \in \mathcal{H}$ that sufficiently capture the dependence between $X$ and $Y$:

$$\mathcal{W}(\mathcal{H}) = \{(f, g) \in \mathcal{H} : \liminf_{M\to+\infty} \big(I(f_M(X); g_M(Y)) - I^*_M(\mathcal{H})\big) \geq 0\},$$

where $I^*_M(\mathcal{H}) = \sup_{(f,g)\in\mathcal{H}} I(f_M(X); g_M(Y))$.

Roughly speaking, representations $(f, g)$ have maximal mutual information if at every discretization level $M$, the mutual information $I(f_M(X); g_M(Y))$ is comparable to the maximal discrete mutual information $I^*_M(\mathcal{H})$ achievable by the function class. In our motivating example in Section 1.1, where the shared latent variables between $X$ and $Y$ are two-dimensional, the learned representations can maximize mutual information only if they can capture all the latent features and have the intrinsic dimension of 2.

In the end, we define $\mathcal{V}(\mathcal{H}) = \mathcal{A}(\mathcal{H}) \cap \mathcal{W}(\mathcal{H})$. Throughout the paper, we assume $\mathcal{V}(\mathcal{H}) \neq \varnothing$, i.e., there exist aligned representations with maximal mutual information.

### 2.2 Intrinsic dimension

We now move on to define the intrinsic dimension of a representation function $f$. To begin with, we define the range of a function $f$ to be $R(f) = \{f(x) : x \in \mathbb{R}^{d_1}\}$, and define the usual linear dimension as $\dim(R(f)) = \dim(\text{span}(R(f)))$. However, this vanilla dimension is not able to capture the nonlinearity in $f$, and the possible manifold structure in the range of $f$. With this in mind, we adopt the following notion of intrinsic dimension, which is closely related to the dimension of manifolds [34].

**Definition 3** (Intrinsic dimension.)**.** We define the intrinsic dimension of $f \in \mathcal{H}_X$, denoted by $\texttt{ID}(f)$, as the smallest integer $k$ such that there exist a measurable function $h : \mathbb{R}^{d_1} \to \mathbb{R}^d$ with $\dim(R(h)) = k$ and an injective measurable function $\phi : R(h) \to \mathbb{R}^d$ such that $f(x) = (\phi \circ h)(x)$ almost everywhere.

As an example, with a full-rank matrix $\mathbf{A} \in \mathbb{R}^{k\times d_Z}$ and an injective map $\zeta : \mathbb{R}^k \to \mathbb{R}^d$, the representation $f(Z) = \zeta(\mathbf{A}Z)$, which is usually known as the multi-index model [80, 38], has intrinsic dimension exactly $k$ where we can choose $h(Z) = \mathbf{A}Z$ and $\phi = \zeta$.

**Proposition 1.** *Suppose $\mathcal{V}(\mathcal{H}) \neq \varnothing$. Then all ideal representations $(f, g) \in \mathcal{V}(\mathcal{H})$ have the same intrinsic dimension $k^\star$, that is, for all $(f, g) \in \mathcal{V}(\mathcal{H})$, we have $\texttt{ID}(f) = \texttt{ID}(g) = k^*$.*

To further understand the definition of $k^*$, we recall the example in Section 1.1. As the shared feature between $X$ and $Y$ has dimension 2, any aligned representations supported on a curve cannot maximize the mutual information with a certain function class. Also, any representations with intrinsic dimension larger than 2 will have additional randomness conditioning on the 2-dimensional shared feature, and thus cannot align.

## 3 Global minimizers of InfoNCE and dimension adaptation

In this section, we analyze the global minimizers of the population infoNCE loss and prove that CLIP adapts to the intrinsic dimension $k^*$ of the ideal representations.

### 3.1 InfoNCE loss and its minimizers

Similar to prior work [79, 44], throughout the paper, we consider the population infoNCE loss

$$\mathcal{L}(f,g,\tau) = -\mathbb{E}_{X,Y}\left[\log \frac{\exp\left(\frac{\sigma(f(X),g(Y))}{\tau}\right)}{\mathbb{E}_{\widetilde{Y}}\exp\left(\frac{\sigma(f(X),g(\widetilde{Y}))}{\tau}\right)}\right] - \mathbb{E}_{X,Y}\left[\log \frac{\exp\left(\frac{\sigma(f(X),g(Y))}{\tau}\right)}{\mathbb{E}_{\widetilde{X}}\exp\left(\frac{\sigma(f(\widetilde{X}),g(Y))}{\tau}\right)}\right], \quad (2)$$

where $\widetilde{Y} \stackrel{d}{=} Y$, $\widetilde{X} \stackrel{d}{=} X$, and $(\widetilde{X}, \widetilde{Y}) \perp\!\!\!\perp (X,Y)$. Based on [79] (Theorem 1) and [44] (Theorem 2.1), the population loss (2) is indeed the large-sample limit of the empirical loss (1) as $\lim_{N\to+\infty} |\mathcal{L}^N(f,g,\tau) - \mathcal{L}(f,g,\tau)| = 0$, for any fixed $f$, $g$, and $\tau > 0$.

**Temperature optimization.** We consider the regime where the temperature $\tau \geq 0$ is also optimized in the training process, which aligns with the practice [60]. In previous theoretical works on CLIP, the temperature parameter was either treated as fixed [79] or not taken into account [49, 22]. However, it has been revealed in both empirical and theoretical studies that different choices of $\tau$ can lead to extremely different properties of the learned representations [79, 78, 18, 20]. Thus, to theoretically understand properties of representations learned by CLIP in practice, we take temperature optimization into account and write $\tau$ as an argument of the loss function.

**Minimizers of the infoNCE loss.** Challenges arise when defining the tuple $(f, g, \tau)$ that minimizes the infoNCE loss in (2). Consider two representations $(f_1, g_1)$ and $(f_2, g_2)$. Both are aligned and continuous. It can be shown that $\lim_{\tau\to 0^+} \mathcal{L}(f_1, g_1, \tau) = -\infty = \lim_{\tau\to 0^+} \mathcal{L}(f_2, g_2, \tau)$. In other words, the infoNCE loss cannot differentiate among aligned continuous representations, due to the unboundedness of mutual information for aligned continuous representations.

To address this issue, we use the same discretization idea in Section 2.1. Precisely, for each slackness $\eta > 0$, we define the set of near-minimizers to be

$$O_{\mathcal{L},\eta}(\mathcal{H}) = \left\{(f,g) \in \mathcal{H} :\ \exists\, \tau \geq \varepsilon(\eta),\ \limsup_{M\to+\infty}\ \left(\mathcal{L}(f_M, g_M, \tau) + 2I^*_M(\mathcal{H})\right) \leq 2\eta\right\}, \quad (3)$$

where $\varepsilon(\eta) > 0$ is nondecreasing[4] in $\eta$ with $\varepsilon(0) = 0$. Correspondingly, we define the set of minimizers to be $\cap_{\eta\geq 0}\mathcal{O}_{\mathcal{L},\eta}(\mathcal{H})$. A few remarks are in order.

- First, instead of looking at $\mathcal{L}(f,g,\tau)$ which could be $-\infty$, we compare its discretized version $\mathcal{L}(f_M, g_M, \tau)$ against a benchmark $-2I^*_M(\mathcal{H})$. Both quantities are finite.
- Second, the lower bound $\varepsilon(\eta)$ on the temperature measures the sensitivity of the solution $(f,g)$ with respect to the change of temperature $\tau$. Then, with any tolerance $\eta > 0$, the set $O_{\mathcal{L},\eta}(\mathcal{H})$ only contains representation maps that achieve small loss with temperature no less than the threshold $\varepsilon(\eta)$ and hence rules out the representations that can only be optimal in the singular case when the temperature is always zero, which is in line with the fact that the temperature can only decrease to zero at a certain rate in the actual training process.

### 3.2 CLIP automatically adapts to intrinsic dimensions

Recall the definition of $k^*$ in Proposition 1. We now present the main result, assuming that $\mathcal{H}$ includes all measurable functions of $X$ and $Y$.

**Theorem 2.** *Assume that $\mathcal{V}(\mathcal{H}) \neq \varnothing$. We have $\bigcap_{\eta\geq 0}\mathcal{O}_{\mathcal{L},\eta}(\mathcal{H}) \neq \varnothing$. In addition, for any $(f,g) \in \bigcap_{\eta\geq 0}\mathcal{O}_{\mathcal{L},\eta}(\mathcal{H})$, we have*

1. ***(similarity maximization)*** *$\sigma(f(X), g(Y)) = m_\sigma(f,g)$ almost surely;*
2. ***(intrinsic dimension adaptation)*** *$\mathtt{ID}(f) = \mathtt{ID}(g) = k^*$;*
3. ***(monotonicity in temperature)*** *infoNCE loss $\mathcal{L}(f,g,\tau)$ is increasing in $\tau$;*
4. ***(mutual information maximization)*** *$(f,g) \in \mathcal{W}(\mathcal{H})$.*

[4] See Appendix C.2 for the exact definition of $\varepsilon(\eta)$.

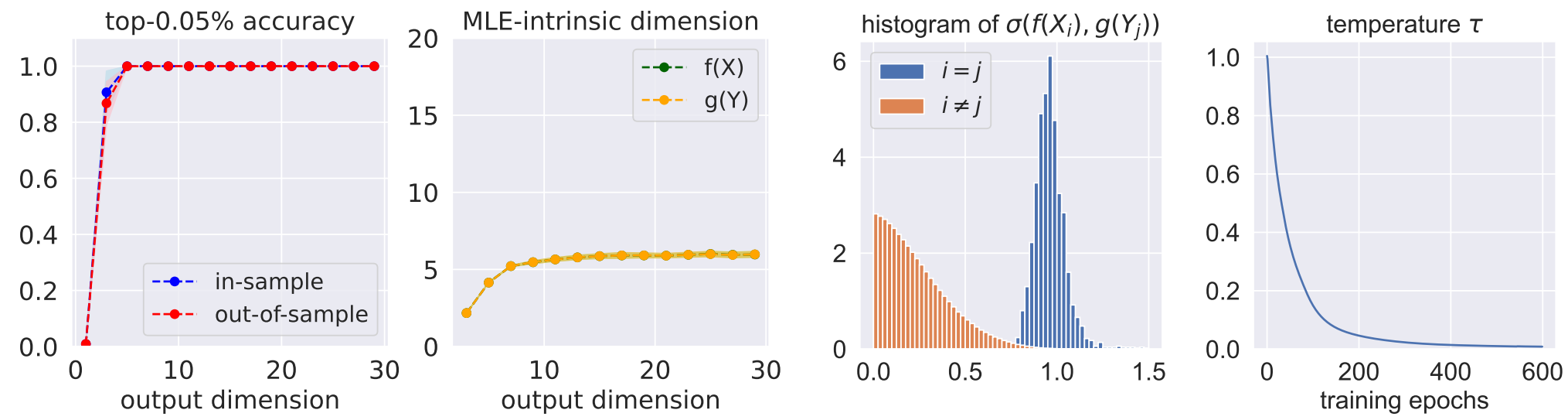

(a) Accuracy and intrinsic dimension with varying $d$. (b) Similarities and convergence of $\tau$ ($d = 20$).

Figure 3: Results with synthetic data: linear setting.

Theorem 2 implies that (approximate) minimizers of the InfoNCE loss have an intrinsic dimension exactly equal to $k^*$. We note that the existence of aligned and uniformly distributed representations over the entire output space, as required in [79], provides a sufficient condition for $\mathcal{V}(\mathcal{H}) \neq \varnothing$. In contrast, our result shows that even without requiring representations to be aligned and uniform over the entire output space, CLIP can still adapt to the intrinsic dimension of multi-modal data. As a corollary, in Appendix E.2, we further show that when the output dimension $d$ is correctly specified, the set of minimizers $\bigcap_{\eta \geq 0} \mathcal{O}_{\mathcal{L},\eta}(\mathcal{H})$ coincides with the set of aligned and uniform representations that maximize mutual information. We also connect our findings to sufficient dimension reduction in Appendix E.1.

We note that, in the concurrent work [52, 43], sufficiency (Corollary 1) is obtained when there exists a pair of encoders that has infoNCE loss coinciding with the minimum over all similarity measures. In contrast, we consider the family of similarity measures taking the form $S_\tau(U, V) = \tau^{-1}\sigma(U, V)$ adopted in CLIP and take into account the optimization of temperature, which is relevant to the result in [53, 43] and [52], but also reveals and explains new phenomena in practice, such as the convergence of temperature and the adaptation to intrinsic dimension. More importantly, our paper puts more emphasis on the exact statistical properties of learned representations, such as sufficiency and low-dimensionality that are shown in Theorem 2, while [52, 43], through the sufficiency measure, focus on the bounds for downstream accuracy and the learnability (via excess infoNCE loss) of architectures (e.g., Transformers) under certain structured models.

## 4 Numerical experiments

In this section, we further justify the theoretical findings with both synthetic and real-world datasets. Starting with a synthetic dataset in Section 4.1, we further consider real datasets: a `CITE-seq` dataset [68, 69] in Section 4.2, `ImageNetV2` dataset [61] in Appendix 4.3, and `YFCC` dataset [72] in Appendix G.7. Throughout this section, we fit the representation map with a 5-layer ReLU neural network with middle-layer widths all fixed at 50 and the output space $\mathbb{R}^d$. This function class is denoted by $\mathcal{F}_{\text{NN}}^{p,d}$, where $p$ is the input dimension adjusted for each setting. We estimate the global intrinsic dimension of data using the MLE-based approach proposed in [35], which is implemented in the `skdim.id` package.[5]

### 4.1 Results with synthetic data

We start with two synthetic datasets with $k^* < \min\{d_1, d_2\}$:

1. Linear setting: consider $N$ i.i.d. draws from the same distribution shown in Section 1.1.
2. Nonlinear setting: consider $N$ i.i.d. draws from the following distribution: $Y_i \overset{i.i.d.}{\sim} \mathcal{N}(0, \mathbf{I}_{d_2})$, $\xi_i \overset{i.i.d.}{\sim} \mathcal{N}(0, \mathbf{I}_{d_1 - k^*})$, and

$$X_i = (0.2Y_{i1}^3, \sin(Y_{i2}Y_{i2}), \log(Y_{i3}^2), \cdots, \log(Y_{ik^*}^2), \xi_i^\top)^\top.$$

[5] https://scikit-dimension.readthedocs.io/en/latest/skdim.id.MLE.html.

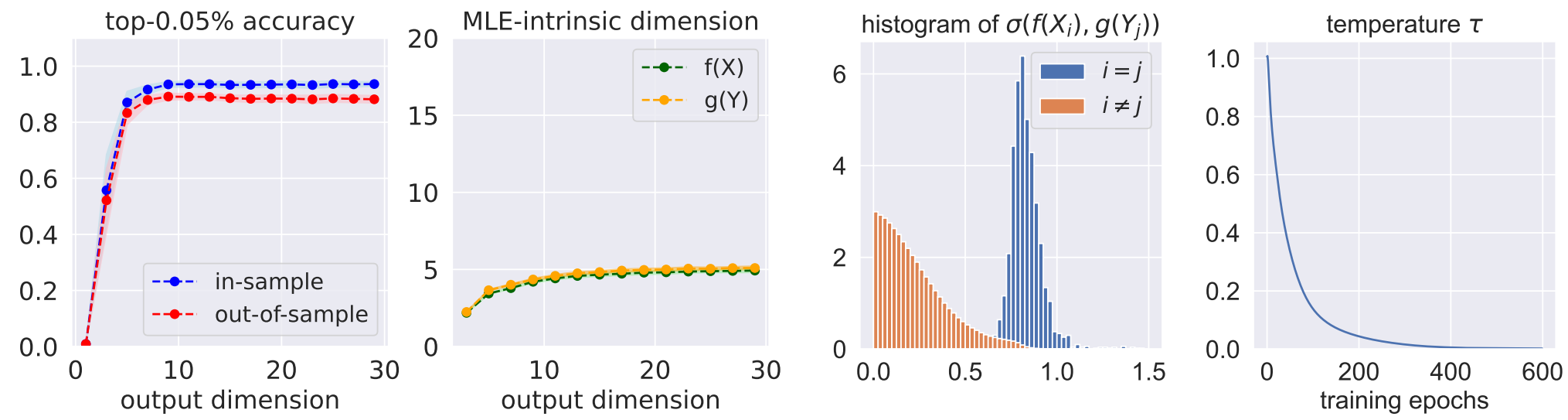

(a) Accuracy and intrinsic dimension with varying $d$. (b) Similarities and convergence of $\tau$ ($d = 20$).

Figure 4: Results with synthetic data: nonlinear setting.

Here we set $d_1 = d_2 = 20$ and $k^* = 5$. The total $N = 14000$ data points are partitioned into a training set $\mathcal{D}_{\text{train}}$ with $|\mathcal{D}_{\text{train}}| = 10000$, a test set $\mathcal{D}_{\text{test}}$ with $|\mathcal{D}_{\text{test}}| = 2000$, and a separate set with size 2000 for estimating the expected norm at each epoch. With representation maps $\widehat{f} \in \mathcal{F}_{\text{NN}}^{d_1,d}$ and $\widehat{g} \in \mathcal{F}_{\text{NN}}^{d_2,d}$ learned with $\mathcal{D}_{\text{train}}$, we consider the downstream task as the top-$\alpha\%$ matching of representations. More concretely, with $\mathcal{D} = \mathcal{D}_{\text{train}}$ or $\mathcal{D}_{\text{test}}$, for any $i \in \mathcal{D}$, denote $\mathcal{N}_\alpha(i; \mathcal{D})$ as all the indices $j \in \mathcal{D}$ such that $\|\widehat{f}(X_i) - \widehat{g}(Y_j)\|$ is the $\lceil \alpha|\mathcal{D}| \rceil$-smallest among $\|\widehat{f}(X_i) - \widehat{g}(Y_k)\|, k \in \mathcal{D}$, and accordingly,

$$\texttt{Acc}_\alpha(\mathcal{D}) = \frac{1}{|\mathcal{D}|} \sum_{i \in \mathcal{D}} \mathbb{1}\{i = N_\alpha(i; \mathcal{D})\}.$$

Then, we can define the in-sample and out-of-sample accuracy by $\texttt{Acc}_\alpha(\mathcal{D}_{\text{train}})$ and $\texttt{Acc}_\alpha(\mathcal{D}_{\text{test}})$, respectively. Particularly, with synthetic data, we choose $\alpha\% = 1/|\mathcal{D}_{\text{test}}| = 0.05\%$, which refers to the top-1 matching accuracy for out-of-sample matching.

Figures 3 and 4 report the results averaged over 50 repetitions in linear and nonlinear settings, respectively. We can see that in both settings, when the output dimension exceeds 5, both the in-sample and out-of-sample accuracy, and the MLE-based estimated intrinsic dimensions tend to saturate. Specifically, the estimated intrinsic dimensions approach the true value $k^* = 5$, which validates that minimizing the multi-modal contrastive loss automatically adapts to the underlying intrinsic dimension of data when the $d \geq k^*$. In addition, in each setting, we fix $d = 20$ and present the histogram of similarities as well as the convergence of $\tau$ in training in Figure 3b and Figure 4b, which validates the theoretical prediction that $\tau$ converges to zero and that the similarity measure between positive pairs will concentrate at a constant that dominates the similarity measure between negative pairs.

### 4.2 Results with `CITE-seq` dataset

The `CITE-seq` dataset contains simultaneous measurements of transcriptomes and cell-surface proteins from the same cell, and we get access to `CITE-seq` dataset via `Seurat`[6] [21] in R. We focus on the `CITE-seq` healthy human bone marrow cells (BMCs) data, consisting of 30672 measured scRNA-seq profiles from bone marrow [69], each with an additional panel of 25 antibodies. Following the preprocessing procedure in `Seurat`, we obtain a two-modal dataset with 24-dimensional protein data and 200-dimensional RNA data. More details of data preprocessing are presented in Appendix G, and ablation studies with a Transformer architecture are presented in Appendix G.6.

In each repetition of the experiments, we randomly sample 20000 rows without replacement from the preprocessed dataset and randomly split the subset into a training set $\mathcal{D}_{\text{train}}$ with $|\mathcal{D}_{\text{train}}| = 10000$, a test set $\mathcal{D}_{\text{test}}$ with $|\mathcal{D}_{\text{test}}| = 2000$, and a separate set with size 8000 for estimating the expected norm at each epoch. Representation maps $\widehat{f}, \widehat{g} \in \mathcal{F}_{\text{NN}}^{d_2,d}$ are learned with $\mathcal{D}_{\text{train}}$ via contrastive loss and the quality of representations is evaluated on $\mathcal{D}_{\text{test}}$ in terms of the classification accuracy of two groups of cell types and the top-$\alpha\%$ matching accuracy ($\alpha\% = 0.5\%$). We vary the output dimension $d$ of the neural network architecture from 1 to 29, and results averaged after 50 repetitions with varying $d$ are presented in Figure 5, in which, both in-sample and out-of-sample accuracy tend to

[6] `https://satijalab.org/seurat/`.

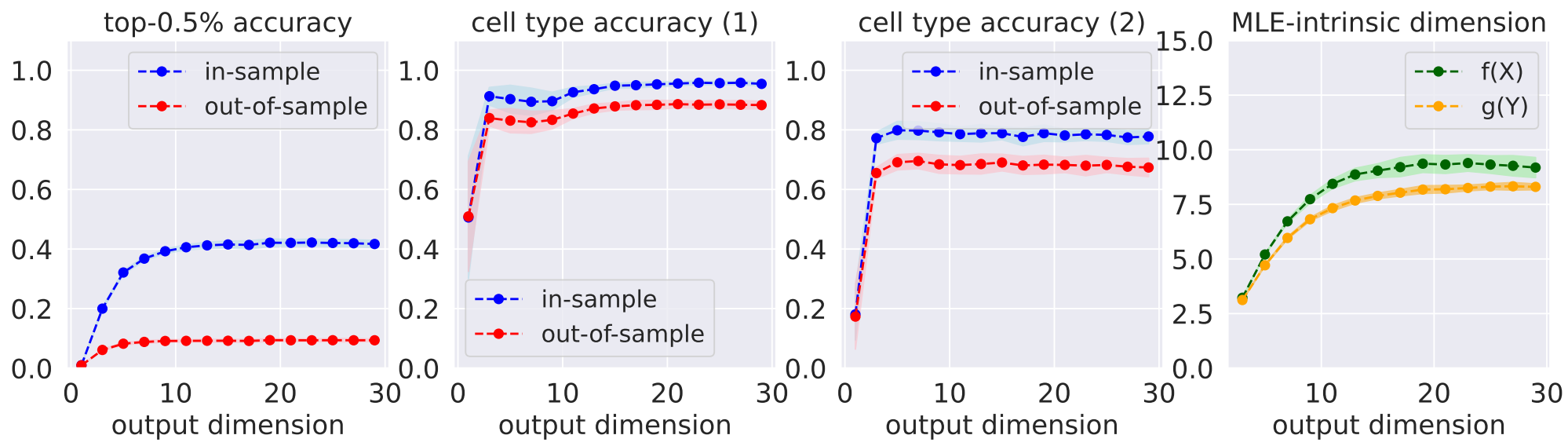

Figure 5: Results with `CITE-seq` dataset.

saturate when the output dimension exceeds $10$. In addition, in the right panel of Figure 5, the MLE-based estimated intrinsic dimension of in-sample representations does not exceed $10$ regardless of the choice of output dimension. This demonstrates that the multi-modal contrastive loss effectively extracts intrinsically low-dimensional features from the `CITE-seq` dataset, which successfully capture the information to differentiate two levels of cell types. Moreover, as is shown in Appendix G.4, with different choices of output dimension $d$, the temperature tends to converge to zero in training, which offers evidence for the existence of nonempty $\mathcal{V}(\mathcal{H})$ when $\mathcal{H}$ is a multi-layer ReLU network.

### 4.3 Results with `ImageNetV2` dataset

We also consider the `ImageNetV2` dataset[7] and focus on two modalities: images and text labels. Here we use the text "This is a photo of a/an `label`" as the input of the text encoder for each image. In addition, each image in `ImageNetV2` dataset can be classified by coarser classes with 67 levels [13], which will be adopted in a downstream classification task. We first use a pretrained image encoder (`ViT-L14`) and text encoder (a masked self-attention Transformer) to obtain preprocessed inputs: $1024$-dimensional image embeddings and $768$-dimensional text embeddings, which are used as inputs to the 5-layer ReLU neural networks.[8]

Similar to previous sections, with $|\mathcal{D}_{\text{train}}| = 8000$, $|\mathcal{D}_{\text{test}}| = 1000$, and a separate dataset with size $1000$ to estimate the expected norms, we consider image classification and the top-$\alpha\%$ matching ($\alpha\% = 0.5\%$) as the downstream tasks. In addition, since images from the class share the same text input, to avoid the singularity when estimating the MLE-based intrinsic dimension, we add independent entrywise perturbations drawn from $\mathcal{N}(0, 0.01)$ to encoded text inputs before CLIP training for both $\mathcal{D}_{\text{train}}$ and $\mathcal{D}_{\text{test}}$. We can see that both accuracies tend to saturate when $d$ is approaching $20$ and the MLE-based estimation of the intrinsic dimension for the image embeddings is approximately $8$ when $d$ keeps increasing. Since the text embeddings have a cluster structure due to discrete labels, the MLE-based intrinsic dimensions are slightly smaller than those of image embeddings. Note that we use a 5-layer ReLU network for CLIP training only for illustration purposes, and better results can potentially be obtained with more sophisticated network architectures.

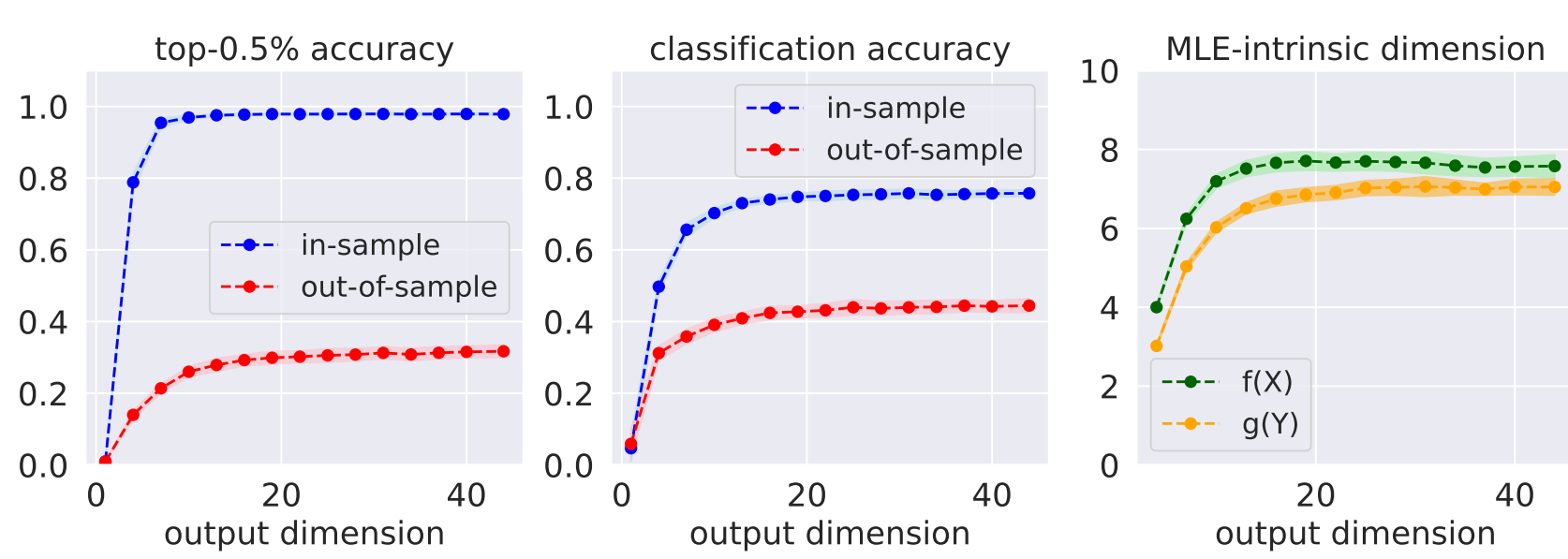

Figure 6: Results with `ImageNetV2` dataset.

[7] We load the dataset from `https://github.com/modestyachts/ImageNetV2_pytorch`

[8] More details of preprocessing are deferred to Appendix G.

## 5 Related work

The integration of multi-modal data has been studied for decades in machine learning. The simplest practice is to concatenate features from all modalities, which is known as early fusion [5], but the direct concatenation can lead to redundancy and high dimensionality [48]. As a dimension reduction technique, canonical correlation analysis (CCA) [23, 2, 26, 28, 25] is widely adopted to learn maximally correlated (linear) projections of modalities as the shared representation, which is further extended to nonlinear projections by kernel CCA [1] and deep CCA [3]. With the initial purpose of integrating acoustic and visual speech signals [82, 57], multi-modal learning has witnessed consistent progress in machine learning [7]. One line of research proposed fine-grained neural network architectures that can output joint representations of the multi-modal input [47, 51, 64]. Another line of research adopts probabilistic graphical models with latent variables to characterize multi-modal data, such as multi-modal deep belief networks [66, 30] and deep Boltzmann machines [67]. Different architectures for modalities are also explored in the field, where certain (constrained) measures of cross-modal similarities are proposed for representation learning [31, 83, 55].

Despite the wide applications of multi-modal learning, the theoretical understanding of its empirical success has drawn attention from the community only in recent years. Earlier works mainly interpret the advantage of multi-modalities from the information-theoretical perspective [65, 14, 42]. A recent work [27] compares the generalization error with different subsets of modalities theoretically, and the result is further discussed in [46].

Similar to our scope, more recent papers include [76], which focuses on the mutual information of multi-modal features as the upper bound of the additive inverse of contrastive loss, and [44], which analyzes the solution of CLIP via the lens of independent component analysis (ICA) and depends on specific data generating process to ensure the identifiability. In addition, concurrent papers [52, 43] introduce the notion of approximate sufficiency and show that this property is satisfied by learned representations by CLIP and also connect the sufficiency measure to the accuracy of downstream tasks with a goal similar to [9].

## 6 Discussion

In this paper, we have characterized the statistical properties of learned representations in multi-modal contrastive learning, and specifically, have shown that the solution can adapt to the intrinsic dimension of data in the setting with $\mathcal{V}(\mathcal{H}) \neq \varnothing$. This property is also relevant to sufficient dimension reduction as is demonstrated in Appendix E.1. The theoretical result is also justified by both synthetic and real datasets, where the estimated intrinsic dimension as well as the downstream accuracy in various tasks tend to saturate regardless of the output dimension (once it exceeds the intrinsic dimension), which implies that the underlying information shared by modalities can indeed be captured by a low-dimensional structure. It is also worth noting that our theory suggests a two-stage fitting strategy when using CLIP. In the first stage, one selects a large output dimension so that the function class is expressive. In the second stage, one can use the intrinsic dimension discovered from the first stage and post-process the representation to a lower dimension. This could potentially accelerate the inference speed.

Our findings also suggest several future directions. Although $\mathcal{V}(\mathcal{H}) \neq \varnothing$ relaxes the condition in [79], it would be of theoretical interest for future work to characterize the precise limit of optimized temperature in general settings beyond this regime, which may also guide the practical choice of $\tau$. In addition, it is interesting to study the finite-sample behavior of the infoNCE loss as well as its minimizers.

## Acknowledgements

C.M. was partially supported by the National Science Foundation via grant DMS-2311127 and DMS CAREER Award 2443867. Z.M. was partially supported by the National Institutes of Health via grant U01CA294514 and the National Science Foundation via grants DMS-2345215 and DMS-2245575.

## Broad impact

This paper provides the theoretical understanding of CLIP, especially its ability to adapt to the intrinsic dimensions of datasets when the temperature parameter is optimized. The theoretical results are justified by both synthetic and real datasets. These findings will enhance the interpretability and guide the postprocessing of learned representations from multi-modal contrastive learning.

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

# A Preliminaries

We start with preliminaries, including notations and the discretization of $\widetilde{B}_d$ introduced in Section 2.1.

## A.1 Notations

We start by summarizing important notations in Table 1.

| | |
|---|---|
| $\mathbb{R}^d$ | $d$-dimensional Eudlidean space |
| $\widetilde{B}_d$ | $\{x \in \mathbb{R}^d : \ \|x\| \leq \sqrt{\Omega}\}$ |
| $\mathtt{ID}(f)$ | intrinsic dimension of $f$ (Definition 3) |
| $C_M = \{c_{M,i} : \ i \in [M]\}$ | a disjoint partition of $\widetilde{B}_d$ |
| $\widehat{\mathcal{B}}_{d,M} = \{z_{M,i} \in c_{M,i} : \ i \in [M]\}$ | discretization of $\widetilde{\mathcal{B}}_d$ under $C_M$ |
| $f_M, g_M$ | discretizations of $(f, g) \in \mathcal{H}$ |
| $\mathcal{H}_M$ | $\{(f_M, g_M) : \ (f, g) \in \mathcal{H}\}$ |
| $m_\sigma(f, g)$ | $\text{ess sup}_{X \perp\!\!\!\perp \widetilde{Y}} \ \sigma(f(X), g(\widetilde{Y}))$ |
| $\mathcal{A}(\mathcal{H}')$ | $\{(f, g) \in \mathcal{H}' : \ \frac{f(X)}{\mathbb{E}\|f(X)\|} = \frac{g(Y)}{\mathbb{E}\|g(Y)\|}$ a.s. and $m_\sigma(f, g) = 1\}$ |
| $I^*_M(\mathcal{H})$ | $\sup_{(f,g)\mathcal{H}} I(f_M(X), g_M(Y))$ |
| $\mathcal{W}(\mathcal{H}')$ | $\{(f, g) \in \mathcal{H}' : \ \liminf_{M\to+\infty}(I(f_M(X); g_M(Y)) - I^*_M(\mathcal{H})) \geq 0\}$ |
| $\mathcal{V}(\mathcal{H}')$ | $\mathcal{A}(\mathcal{H}') \cap \mathcal{W}(\mathcal{H}')$ |
| $Q_{f(X),g(Y),\tau}, \widetilde{Q}_{f(X),g(Y),\tau}$ | smoothed distributions defined in (6) |
| $Q^M_{f(X),g(Y),\tau} = Q^M_\tau, \widetilde{Q}^M_{f(X),g(Y),\tau} = \widetilde{Q}^M_\tau$ | discretized distributions defined with $(f_M, g_M)$ |
| $\varepsilon(\eta)$ | temperature threshold with tolerance $\eta$ defined in (9) |
| $O_{\mathcal{L},\varepsilon,\eta}(\mathcal{H})$ (7) | Set of minimizers up to tolerance $\tau$ with temperature threshold $\varepsilon$ |
| $O_{\mathcal{L},\eta}(\mathcal{H})$ (3) | Set of minimizers up to tolerance $\tau$ with temperature threshold $\varepsilon(\eta)$ |

Table 1: Table of notations

## A.2 Discretization of $\widetilde{B}_d$

With the defined similarity measure $\sigma(\cdot,\cdot)$, to analyze the mutual information between $f(X)$ and $g(Y)$, it is equivalent to considering mutual information between $\widetilde{f}(X) = f(X)/\mathbb{E}\|f(X)\|$ and $\widetilde{g}(Y) = g(Y)/\mathbb{E}\|g(Y)\|$, which satisfies $\mathbb{E}\|\widetilde{f}(X)\| = \mathbb{E}\|\widetilde{g}(Y)\| = 1$, $\|\widetilde{f}(X)\|, \|\widetilde{g}(Y)\| \leq \sqrt{\Omega} < \infty$ almost surely. Then, we denote the range of normalized representations by $\widetilde{B}_d$, which is compact.

For any fineness $M \in \mathcal{M} \subseteq \mathbb{N}$, we consider a discretization $C_M = \{c_{M,i}\}_{i\in[M]}$ of $\widetilde{B}_d$ such that the compact set $\widetilde{B}_d$ is divided into $M$ connected portions, and representatives from each portion $c_{M,i}$ form the set $\widehat{\mathcal{B}}_{d,M} = \{z_{M,i}\}_{i\in[M]}$. We also denote $\mathcal{C}_M$ as the set of all discretizations $C_M$ of $\widetilde{B}_d$ such that

$$\max_{i\in[M]} \text{diam}(c_{M,i}) \to 0 \qquad \text{as} \qquad M \to +\infty.$$

Particularly, we consider the discretization $C_M \in \mathcal{C}_M$ such that each portion $c_{M,i}$ has the same Lebesgue measure $\Delta_M$, which satisfies $\lim_{M\to+\infty} \Delta_M = 0$. In addition, we assume the sequence of discretizations $\{C_M\}_{M\in\mathcal{M}}$ is nested such that $C_{M'} \subseteq C_M$ for any $M' \leq M$.

In the rest of the proof, we write $M \to +\infty$ to denote $\mathcal{M} \ni M \to +\infty$. Accordingly, for any function $f$, we define

$$f_M(x) = z_{M,i} \qquad \text{if } x \in f^{-1}(c_{M,i}),$$

and the function class $\mathcal{H}_M = \{(f_M, g_M) : (f,g) \in \mathcal{H}\}$, which is also nested in the sense that $\mathcal{H}_M \subseteq \mathcal{H}_{\widetilde{M}}$ for all $M \leq \widetilde{M}$. We also notice that $\mathcal{H}_M$ can be viewed as the class of piecewise constant functions. For any $1 \leq k \leq d$, the set $\mathcal{H}_M^k$ is defined similarly.

Abusing notations, for any random vector $U$ supported on $\widetilde{B}_d$, we write $U_M = U(\widehat{\mathcal{B}}_{d,M})$ to denote the discretization of $U$, for which

$$\mathbb{P}(U_M = z_{M,i}) = \mathbb{P}(U \in c_{M,i}) \qquad \text{and} \qquad \mathbb{P}(U_M \notin \widehat{\mathcal{B}}_{d,M}) = 0.$$

If $U$ is supported on $\widehat{\mathcal{B}}_{d,M}$, we have $U_M = U$ almost surely and, particularly, $f_M(X) = (f(X))_M$ almost surely.

### A.2.1 A fine-grained order

For any pairs of random vectors $U = (U_1, \cdots, U_l), \widetilde{U} = (\widetilde{U}_1, \cdots, \widetilde{U}_l) \in (\widetilde{B}_d)^l$ and any functional $G$ of random vectors, we define the order $\preceq_{\mathcal{M}}$ as follows.

$$G(U) \preceq_{\mathcal{M}} G(\widetilde{U}) \iff \limsup_{M \to +\infty} \left\{ G(U_M) - G(\widetilde{U}_M) \right\} \leq 0,$$

and moreover, the inequality is strict, i.e. $G(U) \prec_{\mathcal{M}} G(\widetilde{U})$, if and only if

$$\limsup_{M \to +\infty} \left\{ G(U_M) - G(\widetilde{U}_M) \right\} < 0.$$

We also write $G(U) =_{\mathcal{M}} G(\widetilde{U})$ if $\lim_{M \to +\infty} \left\{ G(U_M) - G(\widetilde{U}_M) \right\}$ exists and is zero.

### A.2.2 Approximation by discretization

In this section, we study the properties of the aforementioned discretization. To start with, for a discrete random vector $U$ with the probability mass function $p_U$, we define the entropy of $U$ by $H(U) = -\mathbb{E}[\log p_U(U)]$. For a continuous random vector $U$ with probability density function $p_U$, we define the differential entropy of $U$ by $\widetilde{H}(U) = -\mathbb{E}[\log p_U(U)]$.

**Approximation of entropy and relative entropy.** To start with, consider the random vector $U \in \widetilde{B}_d$ with probability density function $p$ and the discretized random variable $U_M \in \widetilde{B}_d$ such that $p_i = \mathbb{P}(U_M = z_{M,i}) = \mathbb{P}(U \in c_{M,i})$. As $\int_{c_{M,i}} p(u) \mathsf{d}u \approx p(z_{M,i}) \Delta_M$, we have the following result that connects the Shannon entropy of $U_M$ and the differential entropy of $U$.

**Lemma 1.** *[12, Theorem 8.3.1] As $M \to +\infty$ and $\operatorname{diam}(c_{M,i}) \to 0$, it holds that*

$$\lim_{M \to +\infty} (H(U_M) + \log \Delta_M) = \widetilde{H}(U).$$

See proof in Section F.

In addition, for two random vectors $U, V$ supported on $\widetilde{B}_d$, we recall [77, Theorem 21] in the following lemma.

**Lemma 2.** *[77, Theorem 21] For any $C_M \in \mathcal{C}_M$ with $M \to +\infty$ and $\operatorname{diam}(c_{M,i}) \to 0$, it holds that for any $M$,*

$$D_{\mathrm{KL}}(U_M \,||\, V_M) \leq D_{\mathrm{KL}}(U_{M+1} \,||\, V_{M+1}) \leq D_{\mathrm{KL}}(U \,||\, V),$$

*and moreover,*

$$\lim_{M \to +\infty} D_{\mathrm{KL}}(U_M \,||\, V_M) = D_{\mathrm{KL}}(U \,||\, V).$$

**Approximation of conditional entropy.** For later use, we also consider the approximation of the following quantity

$$\widetilde{H}(\phi(U,V) \mid U),$$

where $U, V$ are random vectors on $\widetilde{B}_d$ with joint density $p(u,v)$ and $\phi : \widetilde{B}_d \times \widetilde{B}_d \to \mathbb{R}$ is continuous. We have the following result.

**Lemma 3.** *Assume $\phi : \widetilde{B}_d \times \widetilde{B}_d \to \mathbb{R}$ is continuous. For any $C_M \in \mathcal{C}_M$ with $M \to +\infty$ and $\operatorname{diam}(c_{M,i}) \to 0$, if $\widetilde{H}(\phi(U,V) \mid U) > 0$, it holds that*

$$\liminf_{M \to +\infty} H(\phi(U_M, V_M) \mid U_M) > 0.$$

See proof in Section F.

**Approximation of (conditional) mutual information.** Consider random vectors $U$, $V$ on $\widetilde{B}_d$ and continuous maps $\phi, \psi : \widetilde{B}_d \times \widetilde{B}_d \to \mathbb{R}$. We are interested in the approximation of $I(\phi(U,V); \psi(U,V))$ and $I(\phi(U,V); \psi(U,V) \mid U)$ by its discretized version. To start with, we have the following result.

**Lemma 4.** *Assume $\phi$ and $\psi$ are continuous and $\mathbb{P}(\phi(U,V) = \psi(U,V)) < 1$. For any $C_M \in \mathcal{C}_M$ with $M \to +\infty$ and $\operatorname{diam}(c_{M,i}) \to 0$, it holds that*

$$\lim_{M \to +\infty} I(\phi(U_M, V_M); \psi(U_M, V_M)) = I(\phi(U,V); \psi(U,V)).$$

See proof in Section F.

For the conditional mutual information, note that

$$I(\phi(U,V); \psi(U,V) \mid U) = I(\phi(U,V); \psi(U,V), U) - I(\phi(U,V); U).$$

Lemma 4 can be applied to both terms, which indicates that

$$\lim_{M \to +\infty} I(\phi(U_M, V_M); \psi(U_M, V_M) \mid U_M) = I(\phi(U,V); \psi(U,V) \mid U).$$

Related results on estimating mutual information based on discretization and binning can be found in [32, 56, 12].

**Approximation of infoNCE loss.** For any pair of representation maps $(f, g) \in \mathcal{H}$, we have the following result.

**Lemma 5.** *For any $(f, g) \in \mathcal{H}$ with $\mathbb{E}\|f(X)\| = \mathbb{E}\|g(Y)\| = 1$ and any $\tau > 0$, it holds that*

$$\lim_{M \to +\infty} \mathcal{L}(f_M, g_M, \tau) = \mathcal{L}(f, g, \tau).$$

See proof in Section F.

### A.3 Intrinsic dimension and its properties

We revisit the definition of intrinsic dimension in Definition 3 for which we have the following property for $\mathtt{ID}(f)$.

**Lemma 6.** *Suppose $f : \mathbb{R}^{d_1} \to \mathbb{R}^d$ satisfies $\mathtt{ID}(f) = k$. Then, for any measurable function $\psi : \mathbb{R}^d \to \mathbb{R}^d$, it holds that $\mathtt{ID}(\psi \circ f) \leq k$.*

#### A.3.1 Proof of Lemma 6

Suppose $\mathtt{ID}(f) = k$. Then, by Definition 3, there exist a measurable function $h : \mathbb{R}^{d_1} \to \mathbb{R}^d$ with $\dim(R(h)) = k$ and an injective function $\phi : \mathbb{R}^d \to \mathbb{R}^d$ such that $f = \phi \circ h$.

For any measurable function $\psi : \mathbb{R}^d \to \mathbb{R}^d$, if $\psi$ is injective, then $\phi \circ f = (\psi \circ \phi) \circ h$. Since $\psi \circ \phi$ is injective, by definition, we have $\mathtt{ID}(\psi \circ f) = k$.

If $\psi$ is not injective, for any $x \in \mathbb{R}^{d_1}$, define

$$\mathcal{T}(x) = \{x' : (\psi \circ \phi \circ h)(x') = (\psi \circ \phi \circ h)(x)\},$$

and accordingly, we define the equivalence relation $\sim$: $x \sim x'$ if and only if $\mathcal{T}(x) = \mathcal{T}(x')$, with which $\Pi(x) = \{x' : x' \sim x\}$. Then, denote $x^{/\sim}$ as the representative of $\Pi(x)$. Based on the notations above, define

$$\widetilde{h} : x \mapsto h(x^{/\sim}).$$

We can verify that for any $x \in \mathbb{R}^{d_1}$, it holds that

$$(\psi \circ \phi \circ h)(x) = (\psi \circ \phi \circ h)(x^{/\sim}) = (\psi \circ \phi \circ \widetilde{h})(x).$$

In addition, for any $z_1, z_2 \in R(\phi \circ \widetilde{h})$, it holds that $\psi(z_1) \neq \psi(z_2)$. Otherwise, suppose there exist $z_1 \neq z_2 \in R(\phi \circ \widetilde{h})$, i.e., there exist $x_1, x_2 \in \mathbb{R}^{d_1}$ satisfying $z_i = (\phi \circ \widetilde{h})(x_i)$ with $i \in \{1, 2\}$, such that $\psi(z_1) = \psi(z_2)$. Then, by the definition of the equivalence relation $\sim$ and the definition of $\widetilde{h}$, we obtain $x_1 \sim x_2$, thus $x_1^{/\sim} = x_2^{/\sim}$ and

$$z_1 = (\phi \circ \widetilde{h})(x_1) = (\phi \circ \widetilde{h})(x_2) = z_2,$$

which draws the contradiction. Hence, $\psi|_{R(\phi\circ\widetilde{h})}$, i.e., $\psi$ restricted on the range of $\phi \circ \widetilde{h}$ is injective. Then, $(\psi\circ\phi)|_{R(\widetilde{h})}$ is also injective. Consequently, we have constructed an injective map $(\psi\circ\phi)|_{R(\widetilde{h})}$ and a measurable function $\widetilde{h}$ with $\dim(R(\widetilde{h})) \leq \dim(R(h)) = k$ such that

$$(\psi \circ f)(x) = [(\psi \circ \phi) \circ \widetilde{h}](x) \qquad \text{almost everywhere},$$

which, by Definition 3, implies that $\texttt{ID}(\psi \circ f) \leq \texttt{ID}(f) = k$.

# B Properties of $\mathcal{V}(\mathcal{H})$

In this section, we focus on the properties of the set $\mathcal{V}(\mathcal{H})$. To begin with, for any subset $\mathcal{H}' = \mathcal{H}'_X \times \mathcal{H}'_Y \subseteq \mathcal{H}$, we generalize the definitions of $\mathcal{A}(\mathcal{H})$ and $\mathcal{W}(\mathcal{H})$ as follows.

**Definition 4.** We define the set of representation maps that realize alignment and similarity maximization as

$$\mathcal{A}(\mathcal{H}') = \left\{(f, g) \in \mathcal{H}' : \frac{f(X)}{\mathbb{E}\|f(X)\|} = \frac{g(Y)}{\mathbb{E}\|g(Y)\|} \text{ almost surely and } m_\sigma(f, g) = 1\right\}.$$

Here $m_\sigma(f, g) \coloneqq \operatorname{ess\,sup}_{X \perp\!\!\!\perp \widetilde{Y}} \sigma(f(X), g(\widetilde{Y}))$ for any $(f, g) \in \mathcal{H}$.

**Definition 5** (Maximal mutual information)**.** We define the following set of pairs $(f, g) \in \mathcal{H}$ that sufficiently capture the dependence between $X$ and $Y$:

$$\mathcal{W}(\mathcal{H}') = \{(f, g) \in \mathcal{H}' : \liminf_{M\to+\infty} \big(I(f_M(X); g_M(Y)) - I^*_M(\mathcal{H})\big) \geq 0\},$$

where $I^*_M(\mathcal{H}) = \sup_{(f,g)\in\mathcal{H}} I(f_M(X); g_M(Y))$.

Note that in the definition of $\mathcal{W}(\mathcal{H}')$ for any $\mathcal{H}' \subseteq \mathcal{H}$, the benchmark mutual information $I^*(\mathcal{H})$ is defined for the entire class $\mathcal{H}$. Accordingly, we define $\mathcal{V}(\mathcal{H}') = \mathcal{A}(\mathcal{H}') \cap \mathcal{W}(\mathcal{H}')$.

Then, we are ready to present the following lemma on properties of $\mathcal{V}(\mathcal{H})$.

**Lemma 7.** *Assume $\mathcal{V}(\mathcal{H}) \neq \varnothing$. There exists $k^* \in [d]$ such that for any $(f^*, g^*) \in \mathcal{V}(\mathcal{H})$,*

*(1)* ***Intrinsic dimension adaptation****. $\texttt{ID}(f^*) = \texttt{ID}(g^*) = k^*$.*

*(2)* ***Sufficiency****. If $k^* < d$, for any $(f, g) \in \mathcal{H}$ with $\max\{\texttt{ID}(f), \texttt{ID}(g)\} > k^*$, it holds that*

$$f(X) \perp\!\!\!\perp g(Y) \mid f^*(X).$$

See proof in Section B.1 and Section B.2, respectively. We note that Proposition 1 is directly implied by Lemma 7 (1).

## B.1 Proof of Lemma 7 (1)

To start with, for any integer $k \in [d]$, we define $\mathcal{H}^k = \{(f, g) \in \mathcal{H} : \texttt{ID}(f) = \texttt{ID}(g) = k\}$ and $\mathcal{V}(\mathcal{H}^k)$ is defined accordingly by

$$\mathcal{V}(\mathcal{H}^k) = \{(f, g) \in \mathcal{V}(\mathcal{H}) : \texttt{ID}(f) = \texttt{ID}(g) = k\}.$$

Then, we define

$$\widetilde{k} = \max\left\{k \in [d] : \mathcal{V}(\mathcal{H}^k) \neq \varnothing\right\}.$$

By definitions of $\mathcal{V}(\mathcal{H})$ and $\mathcal{V}(\mathcal{H}^k)$, we have $\mathcal{V}(\mathcal{H}^{\widetilde{k}}) \subseteq \mathcal{V}(\mathcal{H})$, and we will show that $\mathcal{V}(\mathcal{H}) \subseteq \mathcal{V}(\mathcal{H}^{\widetilde{k}})$.

**Case 1.** If $\widetilde{k} = 1$, recall the definition of $\widetilde{k} = \max\{k \in [d] : \mathcal{V}(\mathcal{H}^k) \neq \varnothing\}$. For any $(f, g) \in \mathcal{V}(\mathcal{H})$, suppose $\mathtt{ID}(f) = k > 1$, it holds that $(f, g) \in \mathcal{H}^k$, thus $\widetilde{k} \geq k > 1$, which draws the contradiction. Hence, for any $(f, g) \in \mathcal{V}(\mathcal{H})$, we have$(f, g) \in \mathcal{H}^1$, i.e. $\mathcal{V}(\mathcal{H}^{\widetilde{k}}) = \mathcal{V}(\mathcal{H})$.

**Case 2.** Consider the case with $\widetilde{k} > 1$. Suppose there exist $(f^*, g^*) \in \mathcal{V}(\mathcal{H}^{\widetilde{k}})$ and $(f, g) \in \mathcal{V}(\mathcal{H}^k)$ but $k < \widetilde{k} \leq d$. Since both mutual information and infoNCE loss are scale-invariant with respect to both $f$ and $g$, without loss of generality, we assume $\mathbb{E}\|f(X)\| = \mathbb{E}\|g(Y)\| = \mathbb{E}\|f^*(X)\| = \mathbb{E}\|g^*(X)\| = 1$. Then, since $\max\{\mathtt{ID}(f), \mathtt{ID}(g)\} = k < \widetilde{k} \leq d$, by Definition 3, there exist aligned $F(X), G(Y) \in \mathbb{R}^k$ and injective measurable functions $\phi, \psi : \mathbb{R}^k \to \mathbb{R}^d$ such that

$$f(X) = \phi\left(F(X)\right) \qquad g(Y) = \psi\left(G(Y)\right).$$

Here we define $H(U)$ as the entropy of random vector $U$ and $U_M = \mathtt{id}_M(U)$, where $\mathtt{id}$ is the identical map on $\mathbb{R}^d$. Then, there exists $\Psi : \mathbb{R}^d \times \mathbb{R}^d \to \mathbb{R}$ such that

$$0 < \widetilde{H}(\Psi(f^*(X), f(X)) \mid f(X)).$$

Otherwise, there exists a measurable function $\zeta$ such that $f^*(X) = \zeta(f(X))$ almost surely. By Definition 3, it holds that $\mathtt{ID}(f) \leq \mathtt{ID}(f^*) = \widetilde{k} < k$, which draws the contradiction.

Then, as $\widetilde{H}(\Psi(f^*(X), f(X)) \mid f(X)) = \widetilde{H}(\Psi(f^*(X), f(X)), f(X)) - \widetilde{H}(f(X))$, by Lemma 3, we also have

$$0 \prec_{\mathcal{M}} H(\Psi(f^*(X), f(X)) \mid f(X)).$$

Otherwise, for any $\Psi$, it holds that $\Psi(f^*(X), f(X))$ is a deterministic function of $f(X)$. Specifically, $f^*(X)$ is a function of $f(X)$ and by Definition 3, we obtain $\widetilde{k} = \mathtt{ID}(f^*) \leq \mathtt{ID}(f) = k$, which draws the contradiction.

Hence, we can define

$$\bar{F}(X) = \begin{pmatrix} F(X) \\ \mathbf{0}_{d-k-1} \\ \Psi(f(X), f^*(X)) \end{pmatrix} \qquad \text{and} \qquad \bar{G}(Y) = \begin{pmatrix} G(Y) \\ \mathbf{0}_{d-k-1} \\ \Psi(g(Y), g^*(Y)) \end{pmatrix},$$

with which

$$\begin{aligned} -I(\bar{F}(X); \bar{G}(Y)) &=_{\mathcal{M}} -H(\bar{F}(X)) \\ &=_{\mathcal{M}} -H(F(X)) - H(\Psi(f^*(X), f(X)) \mid F(X)) \\ &\prec_{\mathcal{M}} -H(F(X)) \\ &=_{\mathcal{M}} -I(F(X); G(Y)). \end{aligned}$$

Since $\phi$ and $\psi$ are injective, we further have

$$-I(\bar{F}(X); \bar{G}(Y)) \prec_{\mathcal{M}} -I(F(X); G(Y)) =_{\mathcal{M}} -I(f(X); g(Y)),$$

which contradicts the fact that $(f, g) \in \mathcal{W}(\mathcal{H})$, which completes the proof. Hence, in the remaining part of the proof, we can define $k^* = \widetilde{k}$.

### B.2 Proof of Lemma 7 (2)

Suppose the statement is not true. Then, there exists $\widetilde{\Phi}_1, \widetilde{\Phi}_2 : \mathbb{R}^d \times \mathbb{R}^d \to \mathbb{R}$ such that

$$0 < I(\widetilde{\Phi}_1(f(X), f^*(X)); \widetilde{\Phi}_2(g(Y), g^*(Y)) \mid f^*(X)),$$

which, by Lemma 4, further indicates that

$$0 \prec_{\mathcal{M}} I(\widetilde{\Phi}_1(f(X), f^*(X)); \widetilde{\Phi}_2(g(Y), g^*(Y)) \mid f^*(X)). \tag{4}$$

To see this, if the statement is not true, it holds that for any $\widetilde{\Phi}_1, \widetilde{\Phi}_2 : \mathbb{R}^d \times \mathbb{R}^d \to \mathbb{R}$,

$$\widetilde{\Phi}_1(f(X), f^*(X)) \perp\!\!\!\perp \widetilde{\Phi}_2(g(Y), g^*(Y)) \mid f^*(X).$$

Specifically, we have $f(X) \perp\!\!\!\perp g(Y) \mid f^*(X)$, which draws the contradiction.

Since $k^* < d$, there exist injective measurable functions $\phi^*, \psi^* : \mathbb{R}^{k^*} \to \mathbb{R}^d$ and $F^*(X), G^*(Y) \in \mathbb{R}^{k^*}$ such that

$$f^*(X) = \phi^*\left(F^*(X)\right) \qquad g^*(Y) = \psi^*\left(G^*(Y)\right).$$

Then, we can define

$$\check{F}(X) = \begin{pmatrix} F^*(X) \\ \mathbf{0}_{d-k^*-1} \\ \widetilde{\Phi}_1(f(X), f^*(X)) \end{pmatrix} \qquad \text{and} \qquad \check{G}(Y) = \begin{pmatrix} G^*(Y) \\ \mathbf{0}_{d-k^*-1} \\ \widetilde{\Phi}_2(g(Y), g^*(Y)) \end{pmatrix},$$

with which, we have

$$-I(\check{F}(X); \check{G}(Y)) \preceq_{\mathcal{M}} -I(F^*(X); \check{G}(Y)) \preceq_{\mathcal{M}} -I(F^*(X); G^*(Y)).$$

Here $=_{\mathcal{M}}$ holds in the first inequality if and only if $\check{G}(Y) \perp\!\!\!\perp \widetilde{\Phi}_1(f(X), f^*(X)) \mid F^*(X)$, i.e.

$$\widetilde{\Phi}_2(g(Y), g^*(Y)) \perp\!\!\!\perp \widetilde{\Phi}_1(f(X), f^*(X)) \mid f^*(X),$$

which draws the contradiction to the construction of $\widetilde{\Phi}_1$ and $\widetilde{\Phi}_2$ in (4). Since $\phi^*$ and $\psi^*$ are injective, we further obtain

$$-I(\check{F}(X); \check{G}(Y)) \prec_{\mathcal{M}} -I(f^*(X); g^*(Y)),$$

which draws the contradiction to the fact that $(f^*, g^*) \in \mathcal{W}(\mathcal{H})$. Thus, we obtain

$$f(X) \perp\!\!\!\perp g(Y) \mid f^*(X).$$

# C Properties of infoNCE loss

Before presenting the technical proof of the main results on intrinsic dimension adaptation, we summarize some properties of the infoNCE loss.

## C.1 A decomposition of the infoNCE loss

A pillar of our proof is the following decomposition of the infoNCE loss that characterizes the gap between the infoNCE loss and mutual information. Similar decompositions have appeared in [53, 79, 52].

**Lemma 8.** *Let $(f(X), g(Y)) \sim P_{f(X),g(Y)}$. If $I(f(X), g(Y)) < +\infty$, there exist joint distributions $Q_{f(X),g(Y),\tau}$ and $\widetilde{Q}_{f(X),g(Y),\tau}$ such that $\mathcal{L}(f, g, \tau)$ can be decomposed as*

$$-2I(f(X); g(Y)) + D_{\mathrm{KL}}\left(P_{f(X),g(Y)} \,||\, Q_{f(X),g(Y),\tau}\right) + D_{\mathrm{KL}}\left(P_{f(X),g(Y)} \,||\, \widetilde{Q}_{f(X),g(Y),\tau}\right) \tag{5}$$

The formal definitions of $Q_{f(X),g(Y),\tau}$ and $\widetilde{Q}_{f(X),g(Y),\tau}$ are presented in Appendix C.1.1. In short, they can be viewed as the kernel smoothing of $P_{f(X),g(Y)}$ in the product space.

Conceptually, given the decomposition (5), minimizing the infoNCE loss is equivalent to simultaneously maximizing the mutual information between $f(X)$ and $g(Y)$, and minimizing the KL-divergence between $P_{f(X),g(Y)}$ and its smoothed versions. Here, the latter encourages the joint distribution of $P_{f(X),g(Y)}$ to spread across the support and implicitly enforces uniformity of the distribution $P_{f(X),g(Y)}$, which is in line with [79]. In our actual proof, the lemma will be used together with discretizations introduced in Section 2.1 and (3) that ensure the finiteness of mutual information.

This decomposition is also relevant to the analysis in [53, 52], where the minimum of infoNCE loss over all possible similarity measures is shown to be bounded from below by $-2I(f(X); g(Y))$. However, in our paper, we focus on the inner product-based similarity that is in line with the implementation of CLIP and does not rely on any unknown information about the distributions of $(X, Y)$.

#### C.1.1 Proof of Lemma 8

Recall the definition of mutual information

$$I(X;Y) = D_{\mathrm{KL}}\left(P_{X,Y} \,||\, P_X \otimes P_Y\right),$$

where $P_{X,Y}$ is the joint distribution of $(X,Y)$ and $P_X$, $P_Y$ are marginal distributions. For any map $f$ and $g$, we have the

$$I(X;Y) \geq I(f(X); g(Y)).$$

When $f$ and $g$ are one-to-one with measurable inverse maps, it holds that $I(X;Y) = I(f(X); g(Y))$.

To characterize the dependence structure between $f(X)$ and $g(Y)$, the conditional distribution $P_{f(X)|g(X)}$ is often of interest, but the estimation is usually not tractable in practice. As an alternative, consider a hypothesized conditional distribution $Q_{f(X)|g(Y),\tau}$ with density function $q_{f(X)|g(Y),\tau}$, with which we have

$$\begin{aligned}
I(f(X); g(Y)) &= \mathbb{E}_P\left[\log \frac{p_{f(X),g(Y)}(U,V)}{p_{f(X)}(U) p_{g(Y)}(V)}\right] \\
&= \mathbb{E}_P\left[\log \frac{p_{f(X)|g(Y)}(U \mid V) q_{f(X)|g(Y),\tau}(U \mid V)}{p_{f(X)}(U) q_{f(X)|g(Y),\tau}(U \mid V)}\right] \\
&= \mathbb{E}_P\left[\log \frac{q_{f(X)|g(Y),\tau}(U \mid V)}{p_{f(X)}(U)}\right] + \mathbb{E}_{P_{g(Y)}}\left[D_{\mathrm{KL}}\left(P_{f(X)|g(Y)} \,||\, Q_{f(X)|g(Y),\tau}\right)\right].
\end{aligned}$$

Similarly, if we have a hypothesized conditional distribution $\widetilde{Q}_{g(Y)|f(X),\tau}$ with density function $\widetilde{q}_{g(Y)|f(X),\tau}$, we can symmetrize the foregoing argument to obtain

$$\begin{aligned}
I(f(X); g(Y)) =& \frac{1}{2}\mathbb{E}_P\left[\log \frac{q_{f(X)|g(Y),\tau}(U \mid V)}{p_{f(X)}(U)}\right] + \frac{1}{2}\mathbb{E}_{P_{g(Y)}}\left[D_{\mathrm{KL}}\left(P_{f(X)|g(Y)} \,||\, Q_{f(X)|g(Y),\tau}\right)\right] \\
&+ \frac{1}{2}\mathbb{E}_P\left[\log \frac{\widetilde{q}_{g(Y)|f(X)}(V \mid U)}{p_{g(Y)}(V)}\right] + \frac{1}{2}\mathbb{E}_{P_{f(X)}}\left[D_{\mathrm{KL}}\left(P_{g(Y)|f(X)} \,||\, \widetilde{Q}_{g(Y)|f(X),\tau}\right)\right].
\end{aligned}$$

We consider the following families of conditional distributions:

$$\begin{aligned}
\mathcal{H}_{f|g} &= \left\{ q_{f(X)|g(Y),\tau}(u \mid v) = p_{f(X)}(u) \cdot \frac{e^{\sigma(u,v)/\tau}}{\mathbb{E}_{P_{g(Y)}} e^{\sigma(u,V)/\tau}} \right\}, \\
\mathcal{H}_{g|f} &= \left\{ \widetilde{q}_{g(Y)|f(X),\tau}(v \mid u) = p_{g(Y)}(v) \cdot \frac{e^{\sigma(u,v)/\tau}}{\mathbb{E}_{P_{f(X)}} e^{\sigma(U,v)/\tau}} \right\},
\end{aligned}$$

where $\sigma(U,V)$ is a similarity measure between random vectors $U$ and $V$. Then, in this case, for any $q_{f(X)|g(Y),\tau}(u \mid v) \in \mathcal{H}_{f|g}$ and $\widetilde{q}_{g(Y)|f(X),\tau}(v \mid u) \in \mathcal{H}_{g|f}$, one can verify that

$$\mathbb{E}_P\left[\log \frac{q_{f(X)|g(Y),\tau}(U \mid V)}{p_{f(X)}(U)}\right] = \frac{1}{\tau}\mathbb{E}_P\left\{\sigma(f(X), g(Y))\right\} - \mathbb{E}_X\left\{\log \mathbb{E}_{\widetilde{Y}}\left[\exp\left(\frac{\sigma(f(X), g(\widetilde{Y}))}{\tau}\right)\right]\right\},$$

$$\mathbb{E}_P\left[\log \frac{\widetilde{q}_{g(Y)|f(X)}(V \mid U)}{p_{g(Y)}(V)}\right] = \frac{1}{\tau}\mathbb{E}_P\left\{\sigma(f(X), g(Y))\right\} - \mathbb{E}_{\widetilde{Y}}\left\{\log \mathbb{E}_X\left[\exp\left(\frac{\sigma(f(X), g(\widetilde{Y}))}{\tau}\right)\right]\right\}.$$

In addition, if we define the joint distributions

$$Q_{f(X),g(Y),\tau} = Q_{f(X)|g(Y),\tau} \otimes P_{g(Y)} \quad \text{and} \quad \widetilde{Q}_{f(X),g(Y),\tau} = \widetilde{Q}_{g(Y)|f(X),\tau} \otimes P_{f(X)}, \qquad (6)$$

we obtain the variational form of mutual information as follows:

$$
\begin{aligned}
I(f(X);g(Y)) = -\frac{1}{2}\mathcal{L}(f,g,\tau) \\
&+ \frac{1}{2}D_{\mathrm{KL}}\left(P_{f(X),g(Y)} \,||\, Q_{f(X),g(Y),\tau}\right) + \frac{1}{2}D_{\mathrm{KL}}\left(P_{f(X),g(Y)} \,||\, \widetilde{Q}_{f(X),g(Y),\tau}\right) \\
&=: -\frac{1}{2}\mathcal{L}(f,g,\tau) + \Delta(P; Q_{f(X),g(Y),\tau}, \widetilde{Q}_{f(X),g(Y),\tau}).
\end{aligned}
$$

Hence, minimizing $\mathcal{L}(f,g,\tau)$ is equivalent to the following optimization problem:

$$
\begin{aligned}
\min_{f,g,\tau} \quad & \Big\{ - I(f(X);g(Y)) + \Delta(P; Q_{f(X),g(Y),\tau}, \widetilde{Q}_{f(X),g(Y),\tau}) \Big\} \\
\text{subject to} \quad & \mathsf{d}Q_{f(X)|g(Y),\tau} \in \mathcal{H}_{f|g},\ \mathsf{d}\widetilde{Q}_{g(Y)|f(X),\tau} \in \mathcal{H}_{g|f}.
\end{aligned}
$$

### C.2 Properties of (approximate) minimizers

Since temperature $\tau$ is also optimized, to analyze the solution path with varying $\tau$, for any $\varepsilon > 0$, we consider the set

$$
O_{\mathcal{L},\varepsilon,\eta}(\mathcal{H}) = \Big\{(f,g) \in \mathcal{H} :\ \exists\, \tau \geq \varepsilon \text{ such that } \limsup_{M\to+\infty} \big(\mathcal{L}(f_M, g_M, \tau) + 2I^*_M(\mathcal{H})\big) \leq 2\eta\Big\}. \tag{7}
$$

Note that we adopt the natural order $\leq$ on $\mathbb{R}$ since with discretization, it always holds that $I^*_M(\mathcal{H}) < +\infty$. Recall that

$$
O_{\mathcal{L},\eta}(\mathcal{H}) = \Big\{(f,g) \in \mathcal{H} :\ \exists\, \tau \geq \varepsilon(\eta) \text{ such that } \limsup_{M\to+\infty} \big(\mathcal{L}(f_M, g_M, \tau) + 2I^*_M(\mathcal{H})\big) \leq 2\eta\Big\},
$$

where we define $\varepsilon(\eta)$ as follows. Denote the set of nondecreasing univariate functions by $\mathcal{C}$ and in addition, the subset

$$
\mathcal{C}^* = \left\{\omega(\cdot) \in \mathcal{C} :\ \bigcap_{\eta\geq 0} O_{\mathcal{L},\omega(\eta),\eta}(\mathcal{H}) \neq \varnothing,\ \lim_{\eta\to 0+} \omega(\eta) = 0\right\}. \tag{8}
$$

The set $\mathcal{C}^*$ is shown to be nonempty when $\mathcal{V}(\mathcal{H}) \neq \varnothing$ in Lemma 9. We define $\omega_1(\cdot) \preceq \omega_2(\cdot)$ if for any $\eta \geq 0$, it holds that $\omega_1(\eta) \leq \omega_2(\eta)$. Then, we choose $\varepsilon(\cdot)$ to be any item in $\mathcal{C}^*$ such that

$$
\{\omega \in \mathcal{C}^* :\ \varepsilon(\cdot) \preceq \omega(\cdot)\} = \varnothing. \tag{9}
$$

Note that if there exists a global minimizer $(f,g)$ that minimizes $\mathcal{L}$ for any $\tau \geq 0$, we can simply define

$$
\varepsilon(\eta) = \sup\Big\{\varepsilon \geq 0 :\ \mathcal{O}_{\mathcal{L},\eta,\varepsilon}(\mathcal{H}) \neq \varnothing\Big\}.
$$

To start with, we have the following lemma.

**Lemma 9.** *Assume $\mathcal{V}(\mathcal{H}) \neq \varnothing$. Then, it holds that*

$$
\bigcap_{\eta\geq 0} \mathcal{O}_{\mathcal{L},\eta}(\mathcal{H}) \neq \varnothing.
$$

*In addition,* $\liminf_{\eta\to 0}\ \varepsilon(\eta) = 0$.

See proof in Section C.2.1.

Then, we are ready to present the main result on the properties of $\cap_{\eta\geq 0}\mathcal{O}_{\mathcal{L},\eta}(\mathcal{H})$. For a general similarity measure $\sigma(U,V)$ and any $(f,g) \in \cap_{\eta\geq 0}\mathcal{O}_{\mathcal{L},\eta}(\mathcal{H})$, we have the following results.

**Lemma 10.** *Assume $(f,g) \in \cap_{\eta\geq 0}\mathcal{O}_{\mathcal{L},\eta}(\mathcal{H})$ and $m_\sigma(f,g) < \infty$. Then, the following properties hold.*

(1) ***Maximal mutual information***. $(f,g) \in \mathcal{W}(\mathcal{H})$.
(2) ***Maximal similarity***. $\sigma(f(X), g(Y)) = m_\sigma(f,g)$ *almost surely.*
(3) ***Monotonicity in*** $\tau$. $\Delta(P; Q_{f(X),g(Y),\tau}, \widetilde{Q}_{f(X),g(Y),\tau})$ *is increasing in* $\tau$.

### C.2.1 Proof of Lemma 9

**Nonemptiness.** To see the nonemptiness, it suffices to show that for any $(f^*,g^*) \in \mathcal{V}(\mathcal{H})$,

$$0 = \lim_{\tau\to 0+}\left\{\limsup_{M\to+\infty}\left(\mathcal{L}(f_M^*, g_M^*, \tau) + 2I_M^*(\mathcal{H})\right)\right\}$$
$$= \lim_{\tau\to 0+}\left\{\limsup_{M\to+\infty}\left(2I_M^*(\mathcal{H}) - 2I(f_M^*(X); g_M^*(Y)) + 2\Delta(P; Q_{f_M^*(X),g_M^*(Y),\tau}, \widetilde{Q}_{f_M^*(X),g_M^*(Y),\tau})\right)\right\}.$$

Without loss of generality, assume $\mathbb{E}\|f^*(X)\| = \mathbb{E}\|g^*(Y)\| = 1$. By definition, since $(f^*,g^*) \in \mathcal{V}(\mathcal{H})$, we have

$$\liminf_{M\to+\infty}\left(I(f_M^*(X); g_M^*(Y)) - I_M^*(\mathcal{H})\right) = \limsup_{M\to+\infty}\left(I_M^*(\mathcal{H}) - I(f_M^*(X); g_M^*(Y))\right) = 0,$$

which implies that, for any $\tau \geq 0$,

$$\limsup_{M\to+\infty}\left(\mathcal{L}(f_M^*, g_M^*, \tau) + 2I_M^*(\mathcal{H})\right) \leq 2\limsup_{M\to+\infty}\Delta(P; Q_{f_M^*(X),g_M^*(Y),\tau}, \widetilde{Q}_{f_M^*(X),g_M^*(Y),\tau}).$$

As $(f^*,g^*) \in \mathcal{A}(\mathcal{H})$, i.e., $f^*(X) = g^*(Y)$ almost surely, we further have $p_{f^*(X)}(u) = p_{g^*(Y)}(u)$ and

$$p_{f^*(X),g^*(Y)}(u,v) = p_{f^*(X)}(u)\mathbf{1}_{u=v},$$

which is also true after discretization, i.e., $f_M^*(X) = g_M^*(Y)$ almost surely and

$$p_{f_M^*(X),g_M^*(Y)}(u,v) = p_{f_M^*(X)}(u)\mathbf{1}_{u=v}.$$

In addition, for any $M \in \mathcal{M}$,

$$\begin{aligned}
&D_{\mathrm{KL}}(P_{f_M^*(X),g_M^*(Y)} \,||\, Q_{f_M^*(X),g_M^*(Y),\tau}) \\
&= \iint_{\mathbb{R}^d\times\mathbb{R}^d} p_{f_M^*(X),g_M^*(Y)}(u,v)\log p_{f_M^*(X),g_M^*(Y)}(u,v)\mathsf{d}u\mathsf{d}v \\
&\qquad - \iint_{\mathbb{R}^d\times\mathbb{R}^d} p_{f_M^*(X),g_M^*(Y)}(u,v)\log\frac{e^{1/\tau}(p_{f_M^*(X)}(u))^2}{\int_{\mathbb{R}^d} p_{f_M^*(X)}(\widetilde{v})e^{\langle u,\widetilde{v}\rangle/\tau}\mathsf{d}\widetilde{v}}\mathsf{d}u\mathsf{d}v \\
&= -\frac{1}{\tau} + H(f_M^*(X)) + \int_{\mathbb{R}^d} p_{f_M^*(X)}(u)\log\left(\int_{\mathbb{R}^d} p_{f_M^*(X)}(\widetilde{v})e^{\langle u,\widetilde{v}\rangle/\tau}\mathsf{d}\widetilde{v}\right)\mathsf{d}u \\
&= \int_{\mathbb{R}^d} p_{f_M^*(X)}(u)\log\frac{\int_{\mathbb{R}^d} p_{f_M^*(X)}(\widetilde{v})e^{\langle u,\widetilde{v}\rangle/\tau}\mathsf{d}\widetilde{v}}{p_{f_M^*(X)}(u)e^{1/\tau}}\mathsf{d}u.
\end{aligned}$$

Then, by Lemma 2, we have

$$\lim_{M\to+\infty} D_{\mathrm{KL}}(P_{f_M^*(X),g_M^*(Y)} \,||\, Q_{f_M^*(X),g_M^*(Y),\tau}) = D_{\mathrm{KL}}(P_{f^*(X),g^*(Y)} \,||\, Q_{f^*(X),g^*(Y),\tau}),$$

for which, it holds that

$$\lim_{\tau\to 0+} D_{\mathrm{KL}}(P_{f^*(X),g^*(Y)} \,||\, Q_{f^*(X),g^*(Y),\tau}) = \lim_{\tau\to 0+}\left\{\int_{\mathbb{R}^d} p_{f^*(X)}(u)\log\frac{\int_{\mathbb{R}^d} p_{f^*(X)}(\widetilde{v})e^{\langle u,\widetilde{v}\rangle/\tau}\mathsf{d}\widetilde{v}}{p_{f^*(X)}(u)e^{1/\tau}}\mathsf{d}u\right\}$$
$$= \int_{\mathbb{R}^d} p_{f^*(X)}(u)\log\frac{p_{f^*(X)}(u)}{p_{f^*(X)}(u)}\mathsf{d}u = 0.$$

Hence, by the symmetry of $\Delta(P; Q_{f_M^*(X),g_M^*(Y),\tau}, \widetilde{Q}_{f_M^*(X),g_M^*(Y),\tau})$ in $f$ and $g$, we obtain

$$\lim_{\tau\to 0+} \left\{ \limsup_{M\to+\infty} \Delta(P; Q_{f_M^*(X),g_M^*(Y),\tau}, \widetilde{Q}_{f_M^*(X),g_M^*(Y),\tau}) \right\} = 0.$$

Then, the constant function $\omega(\eta) = 0$ is in the set $\mathcal{C}^*$ defined in (8), indicating that $\mathcal{C}^* \neq \varnothing$, thus $\cap_{\eta\geq 0}\mathcal{O}_{\mathcal{L},\eta}(\mathcal{H}) \neq \varnothing$.

**Zero limit infimum.** Suppose $\liminf_{\eta\to 0}\ \varepsilon(\eta) = \underline{\tau} > 0$. Then, for any $(f,g) \in \mathcal{H}$, it holds that

$$\limsup_{M\to+\infty} (\mathcal{L}(f_M, g_M, \underline{\tau}) + 2I_M^*(\mathcal{H})) \leq 0.$$

In addition, as we have shown in Lemma 8, it holds that

$$\mathcal{L}(f_M, g_M, \tau) + 2I_M^*(\mathcal{H}) \geq -2I(f_M(X); g_M(Y)) + 2I_M^*(\mathcal{H}) \geq 0,$$

which further implies that

$$0 \leq \liminf_{M\to+\infty} (\mathcal{L}(f_M, g_M, \underline{\tau}) + 2I_M^*(\mathcal{H})) \leq \limsup_{M\to+\infty} (\mathcal{L}(f_M, g_M, \underline{\tau}) + 2I_M^*(\mathcal{H})) \leq 0.$$

Hence, $\lim_{M\to+\infty}(\mathcal{L}(f_M^*, g_M^*, \underline{\tau}) + 2I_M^*(\mathcal{H})) = 0$. Moreover, as

$$\mathcal{L}(f_M, g_M, \underline{\tau}) + 2I_M^*(\mathcal{H}) \geq \Delta(P; Q_{f_M(X),g_M(Y),\underline{\tau}}, \widetilde{Q}_{f_M(X),g_M(Y),\underline{\tau}}) \geq 0,$$

we further obtain

$$\lim_{M\to+\infty} \Delta(P; Q_{f_M(X),g_M(Y),\underline{\tau}}, \widetilde{Q}_{f_M(X),g_M(Y),\underline{\tau}}) = 0.$$

In this case, the joint distribution of $(f(X), g(Y))$ takes the form

$$p_{f(X),g(Y)}(u,v) = \frac{\exp\left(\frac{\sigma(u,v)}{\underline{\tau}}\right) p_{f(X)}(u) p_{g(Y)}(v)}{\mathbb{E}_{P_{f(X)}\otimes P_{g(Y)}} \exp\left(\frac{\sigma(f(X),g(Y))}{\underline{\tau}}\right)},$$

thus, the mutual information satisfies

$$\begin{aligned} I(f(X); g(Y)) &= \iint p_{f(X),g(Y)}(u,v) \log \frac{p_{f(X),g(Y)}(u,v)}{p_{f(X)}(u)p_{g(Y)}(v)} \mathsf{d}u\mathsf{d}v \\ &\leq \iint \frac{\sigma(u,v)}{\underline{\tau}} \cdot \frac{\exp\left(\frac{\sigma(u,v)}{\underline{\tau}}\right) p_{f(X)}(u) p_{g(Y)}(v)}{\mathbb{E}_{P_{f(X)}\otimes P_{g(Y)}} \exp\left(\frac{\sigma(f(X),g(Y))}{\underline{\tau}}\right)} \mathsf{d}u\mathsf{d}v \\ &\qquad - \log \mathbb{E}_{P_{f(X)}\otimes P_{g(Y)}} \exp\left(\frac{\sigma(f(X),g(Y))}{\underline{\tau}}\right) \\ &\leq \frac{\Omega}{\underline{\tau}} < \infty. \end{aligned}$$

Hence, $\limsup_{M\to+\infty} I(f_M(X); g_M(Y)) < \infty$ for any $(f,g) \in \mathcal{H}$. Then, we obtain

$$\limsup_{M\to+\infty}\ I_M^*(\mathcal{H}) = \limsup_{M\to+\infty}\ \sup_{(f,g)\in\mathcal{H}} I(f_M(X); g_M(Y)) \leq \frac{\Omega}{\underline{\tau}} < \infty. \tag{10}$$

In addition, since $\mathcal{V}(\mathcal{H}) \neq \varnothing$, there exist $(f^*, g^*)$ such that $f^*(X)/\mathbb{E}\|f^*(X)\| = g^*(Y)/\mathbb{E}\|g^*(Y)\|$ almost surely, thus

$$\limsup_{M\to+\infty} I_M^*(\mathcal{H}) \geq \limsup_{M\to+\infty} I(f_M^*(X); g_M^*(X)) = +\infty,$$

which draws the contradiction to (10).

### C.2.2 Proof of Lemma 10 (1) and (2)

Assume $(f,g) \in \cap_{\eta\geq 0}\mathcal{O}_{\mathcal{L},\eta}(\mathcal{H})$. There exists a decreasing sequence $\{\eta_j\}_{j\in\mathbb{N}}$ such that $\eta_j \to 0$ as $j \to +\infty$. Accordingly, there is an associated sequence $\tau_j = \varepsilon(\eta_j)$, which satisfies $\liminf_{j\to+\infty} \tau_j = 0$ as is shown in Lemma 9. By the decomposition in Lemma 8 and the definition of $\mathcal{O}_{\mathcal{L},\eta}(\mathcal{H})$, for any $j \in \mathbb{N}$, it holds that

$$\limsup_{M\to+\infty}\left\{I_M^*(\mathcal{H}) - I(f_M(X); g_M(Y)) + \Delta(P; Q_{f_M(X),g_M(Y),\tau_j}, \widetilde{Q}_{f_M(X),g_M(Y),\tau_j})\right\} \leq \eta_j.$$

Since $\liminf_{j\to+\infty} \tau_j = 0$, there is a convergent subsequence $\{\tau_{j_l}\}_{l\in\mathbb{N}}$ with $\lim_{l\to+\infty}\tau_{j_l} = 0$. Without loss of generality, we assume $\{\tau_j\}_{j\in\mathbb{N}}$ is convergent such that $\lim_{j\to+\infty}\tau_j = 0$.

In addition, we have, for any $M \in \mathcal{M}$ and $j \in \mathbb{N}$,

$$I_M^*(\mathcal{H}) - I(f_M(X); g_M(Y)) \geq 0, \qquad \Delta(P; Q_{f_M(X),g_M(Y),\tau_j}, \widetilde{Q}_{f_M(X),g_M(Y),\tau_j}) \geq 0,$$

which implies that

$$\max\left\{\limsup_{M\to+\infty}\{I_M^*(\mathcal{H}) - I(f_M(X); g_M(Y))\}, \limsup_{M\to+\infty}\left\{\Delta(P; Q_{f_M(X),g_M(Y),\tau_j}, \widetilde{Q}_{f_M(X),g_M(Y),\tau_j})\right\}\right\}$$

$$\leq \limsup_{M\to+\infty}\left\{I_M^*(\mathcal{H}) - I(f_M(X); g_M(Y)) + \Delta(P; Q_{f_M(X),g_M(Y),\tau_j}, \widetilde{Q}_{f_M(X),g_M(Y),\tau_j})\right\} \leq \eta_j.$$

Then, by Moore-Osgood theorem [54, 19], we can exchange the order of limit operators and obtain

$$\begin{cases} \limsup_{M\to+\infty}\ \Delta(P; Q_{f_M(X),g_M(Y),0}, \widetilde{Q}_{f_M(X),g_M(Y),0}) = 0, \\ \liminf_{M\to+\infty}\ \big(I(f_M(X); g_M(Y)) - I_M^*(\mathcal{H})\big) = 0. \end{cases}$$

Denote $m = m_\sigma(f,g) = \operatorname{ess\,sup}\ \sigma(f(X), g(\widetilde{Y}))$, and $\mathcal{E}_m = \{(u,v) \in R(f)\times R(g):\ \sigma(u,v) = m\}$, $\mathcal{A}^m(u) = \{v \in R(g):\ \sigma(u,v) = m\}$, $\mathcal{B}^m(v) = \{u \in R(f):\ \sigma(u,v) = m\}$. As $\varepsilon \to 0$, to ensure $\limsup_{M\to+\infty}\ \Delta(P^M; Q^M_{\widetilde{\varepsilon}}, \widetilde{Q}^M_{\widetilde{\varepsilon}}) = 0$, we have

$$\begin{aligned} p_{f(X),g(Y)}(u,v) &= p_{f(X)}(u)p_{g(Y)}(v)\frac{\mathbf{1}_{\sigma(u,v)=m}}{\mathbb{E}_{g(Y)}[\mathbf{1}_{\{V:\sigma(u,V)=m\}}]} \\ &= p_{f(X)}(u)p_{g(Y)}(v)\frac{\mathbf{1}_{\sigma(u,v)=m}}{\mathbb{E}_{f(X)}[\mathbf{1}_{\{U:\sigma(U,v)=m\}}]}, \end{aligned} \tag{11}$$

which indicates that

$$\begin{aligned} \mathbb{E}_{g(Y)}[\mathbf{1}_{\{V:\sigma(u,V)=m\}}] &= \mathbb{E}_{f(X)}[\mathbf{1}_{\{U:\sigma(U,v)=m\}}] \\ &= \iint_{\mathcal{E}_m} p_{f(X)}(u)p_{g(Y)}(v)\mathsf{d}u\mathsf{d}v := A_m. \end{aligned}$$

Here $\mathbf{1}_A$ is the indicator function with respect to the set $A$ and if $A = \{a\}$, we define $\mathbb{E}_{f(X)}[\mathbf{1}_A] = p_{f(X)}(a)$. In this case, $\sigma(f(X), g(Y)) = m$ almost surely, which completes the proof.

### C.2.3 Proof of Lemma 10 (3)

Recall the definition

$$\begin{aligned} &\Delta(P; Q_{f(X),g(Y),\tau}, \widetilde{Q}_{f(X),g(Y),\tau}) \\ &= \frac{1}{2}D_{\mathrm{KL}}\left(P_{f(X),g(Y)} \,||\, Q_{f(X),g(Y),\tau}\right) + \frac{1}{2}D_{\mathrm{KL}}\left(P_{f(X),g(Y)} \,||\, \widetilde{Q}_{f(X),g(Y),\tau}\right). \end{aligned}$$

It suffices to show the monotonicity of the first term in $\tau$. Since infoNCE loss is scale-invariant with respect to both $f$ and $g$, without loss of generality, we can assume $\mathbb{E}\|f(X)\| = \mathbb{E}\|g(Y)\| = 1$.

When the distribution $P_{f(X),g(Y)}$ degenerates on the support $\{\sigma(f(X), g(Y)) = m_\sigma(f,g)\}$, $P_{f(X),g(Y)}$ is not absolutely continuous with respect to $Q_{f(X),g(Y),\tau}$ and $\widetilde{Q}_{f(X),g(Y),\tau}$)), thus

$\Delta(P; Q_{f(X),g(Y),\tau}, \widetilde{Q}_{f(X),g(Y),\tau}) = +\infty$ for any $\tau > 0$ and $\Delta(P; Q_{f(X),g(Y),\tau}, \widetilde{Q}_{f(X),g(Y),\tau}) = 0$ if and only if $\tau = 0$.

If the distribution $P_{f(X),g(Y)}$ is not degenerate, by the joint distribution in (11), we further have

$$\begin{aligned}
&D_{\mathrm{KL}}\left(P_{f(X),g(Y)} \,||\, Q_{f(X),g(Y),\tau}\right) \\
&= \iint_{\mathbb{R}^d\times\mathbb{R}^d} p_{f(X),g(Y)}(u,v)\log\frac{p_{f(X),g(Y)}(u,v)}{q_{f(X),g(Y),\tau}(u,v)}\mathrm{d}u\mathrm{d}v \\
&= \iint_{\mathbb{R}^d\times\mathbb{R}^d} p_{f(X),g(Y)}(u,v)\log\frac{p_{f(X),g(Y)}(u,v)}{p_{f(X)}(u)p_{g(Y)}(v)}\mathrm{d}u\mathrm{d}v \\
&\qquad - \iint_{\mathbb{R}^d\times\mathbb{R}^d} p_{f(X),g(Y)}(u,v)\log\frac{e^{m/\tau}}{\int_{\mathbb{R}^d} p_{g(Y)}(\widetilde{v})e^{\sigma(u,\widetilde{v})/\tau}\mathrm{d}\widetilde{v}}\mathrm{d}u\mathrm{d}v.
\end{aligned}$$

Then, since the first term is free of $\tau$, we have

$$\begin{aligned}
&\frac{\partial}{\partial\tau}D_{\mathrm{KL}}\left(P_{f(X),g(Y)} \,||\, Q_{f(X),g(Y),\tau}\right) \\
&= \frac{\partial}{\partial\tau}\left\{-\iint_{\mathbb{R}^d\times\mathbb{R}^d} p_{f(X),g(Y)}(u,v)\log\frac{e^{m/\tau}}{\int_{\mathbb{R}^d} p_{g(Y)}(\widetilde{v})e^{\sigma(u,\widetilde{v})/\tau}\mathrm{d}\widetilde{v}}\mathrm{d}u\mathrm{d}v\right\} \\
&= \frac{\partial}{\partial\tau}\left\{-\frac{m}{\tau} + \iint_{\mathbb{R}^d\times\mathbb{R}^d} p_{f(X),g(Y)}(u,v)\left[\log\int_{\mathbb{R}^d} p_{g(Y)}(\widetilde{v})e^{\sigma(u,\widetilde{v})/\tau}\mathrm{d}\widetilde{v}\right]\mathrm{d}u\mathrm{d}v\right\} \\
&= \frac{1}{\tau^2}\left\{m - \iint_{\mathbb{R}^d\times\mathbb{R}^d} p_{f(X),g(Y)}(u,v)\left[\frac{\int_{\mathbb{R}^d}\sigma(u,\widetilde{v})p_{g(Y)}(\widetilde{v})e^{\sigma(u,\widetilde{v})/\tau}\mathrm{d}\widetilde{v}}{\int_{\mathbb{R}^d} p_{g(Y)}(\widetilde{v})e^{\sigma(u,\widetilde{v})/\tau}\mathrm{d}\widetilde{v}}\right]\mathrm{d}u\mathrm{d}v\right\}.
\end{aligned}$$

Since $\sigma(u,\widetilde{v}) \le m$, the derivative is nonnegative, thus

$$\frac{\partial}{\partial\tau}D_{\mathrm{KL}}\left(P_{f(X),g(Y)} \,||\, Q_{f(X),g(Y),\tau}\right) \ge 0.$$

The equality holds if and only if $\sigma(f(X), g(\widetilde{Y})) = m_\sigma(f,g)$ almost surely with respect to the product distribution $P_{f(X)} \otimes P_{g(Y)}$. In this case, for any $\tau \ge 0$, the infoNCE loss $\mathcal{L}(f,g,\tau) = 0$, which contradicts the fact that $(f,g) \in \cap_{\eta\ge 0}\mathcal{O}_{\mathcal{L},\eta}(\mathcal{H})$, thus, we have

$$\frac{\partial}{\partial\tau}D_{\mathrm{KL}}\left(P_{f(X),g(Y)} \,||\, Q_{f(X),g(Y),\tau}\right) > 0.$$

By the symmetry of $\Delta(P; Q_{f(X),g(Y),\tau}, \widetilde{Q}_{f(X),g(Y),\tau})$ in $f$ and $g$, we conclude that $\Delta(P; Q_{f(X),g(Y),\tau}, \widetilde{Q}_{f(X),g(Y),\tau})$ is increasing in $\tau$.

# D Proof of Theorem 2

We present the proof of Theorem 2 in this section. If $\mathcal{V}(\mathcal{H}) \ne \varnothing$, we have $\mathcal{V}(\mathcal{H}) = \mathcal{V}(\mathcal{H}^{k^*})$ as a result of Lemma 7 and $\cap_{\eta\ge 0}\mathcal{O}_{\mathcal{L},\eta}(\mathcal{H}) \ne \varnothing$ as a result of Lemma 9, respectively.

In addition, Theorem 2 (1)(3)(4) are directly implied by Lemma 10, thus it suffices to prove Theorem 2 (2).

Based on the previous notations, we will prove the result with the following steps.

- **Step 1**: for any $(f,g) \in \cap_{\eta\ge 0}\mathcal{O}_{\mathcal{L},\eta}(\mathcal{H})$, it holds that $\min\{\mathtt{ID}(f), \mathtt{ID}(g)\} \ge k^*$;
- **Step 2**: for any $(f,g) \in \cap_{\eta\ge 0}\mathcal{O}_{\mathcal{L},\eta}(\mathcal{H})$, $\max\{\mathtt{ID}(f), \mathtt{ID}(g)\} \le k^*$.

Combining the two steps, we conclude that for any $(f,g) \in \cap_{\eta\ge 0}\mathcal{O}_{\mathcal{L},\eta}(\mathcal{H})$, it holds that $\mathtt{ID}(f) = \mathtt{ID}(g) = k^*$.

## D.1 Proof of Step 1

If $\mathtt{ID}(f) = k < k^* \le d$, there exists a random vector $Z \in \mathcal{B}_k$ such that $f(X) = \phi(Z)$ almost surely for some injective measurable function $\phi : \mathbb{R}^k \to \mathbb{R}^d$. In addition, we have $\mathtt{ID}(g) < d$. To see this, suppose $\mathtt{ID}(g) = d$. Consider $(u, v) \in R(f) \times R(g)$ such that $\langle u, v\rangle = m_\sigma(f, g)\mathbb{E}\|f(X)\|\mathbb{E}\|g(Y)\|$ and there exists an open set $\mathcal{B}_v$ satisfying $v \in \mathcal{B}_v \subseteq R(g)$. Then, consider the linear function $\zeta(z) = \langle u, z\rangle$, which is continuous in $z \in \mathcal{B}_v$. There exists $\widetilde{v} \in \mathcal{B}_v \subseteq R(g)$ such that $\zeta(\widetilde{v}) = \langle u, \widetilde{v}\rangle > \zeta(v) = m_\sigma(f, g)\mathbb{E}\|f(X)\|\mathbb{E}\|g(Y)\|$, which draws the contradiction.

Then, there exist $F(X), G(Y) \in \mathbb{R}^k$ and injective measurable functions $\phi, \psi : \mathbb{R}^k \to \mathbb{R}^d$ such that

$$f(X) = \phi\left(F(X)\right) \qquad g(Y) = \psi\left(G(Y)\right).$$

Accordingly, for any $(f^*, g^*) \in \mathcal{V}(\mathcal{H}) = \mathcal{V}(\mathcal{H}^{k^*})$, there exist $\Phi_1, \Phi_2 : \mathbb{R}^d \times \mathbb{R}^d \to \mathbb{R}$ such that

$$0 < I(\Phi_1(f(X), f^*(X)); \Phi_2(g(Y), g^*(Y)) \mid f(X)).$$

To see this, otherwise, for any $\Phi_1, \Phi_2 : \mathbb{R}^d \times \mathbb{R}^d \to \mathbb{R}$, it holds that

$$I(\Phi_1(f(X), f^*(X)); \Phi_2(g(Y), g^*(Y)) \mid f(X)) = 0,$$

which further indicates that

$$\Phi_1(f(X), f^*(X)) \perp\!\!\!\perp \Phi_2(g(Y), g^*(Y)) \mid f(X).$$

Thus, specifically, $f^*(X) \perp\!\!\!\perp g^*(Y) \mid f(X)$, which holds if and only if there exists a measurable function $h$ such that $f^*(X) = (h \circ f)(X)$ almost surely, indicating that $\mathtt{ID}(f^*) \le \mathtt{ID}(f) = k < k^*$, which draws the contradiction. By Lemma 4, we further have

$$0 \prec_{\mathcal{M}} I(\Phi_1(f(X), f^*(X)); \Phi_2(g(Y), g^*(Y)) \mid f(X)).$$

Hence, we can define

$$\widetilde{F}(X) = \begin{pmatrix} F(X) \\ \mathbf{0}_{d-k-1} \\ \Phi_1(f(X), f^*(X)) \end{pmatrix} \qquad \text{and} \qquad \widetilde{G}(Y) = \begin{pmatrix} G(Y) \\ \mathbf{0}_{d-k-1} \\ \Phi_2(g(Y), g^*(Y)) \end{pmatrix}.$$

By the Data Processing Inequality [58, Theorem 2.17], we have

$$\begin{aligned} -I(\widetilde{F}(X); \widetilde{G}(Y)) &\preceq_{\mathcal{M}} -I(F(X); \widetilde{G}(Y)) \\ &\preceq_{\mathcal{M}} -I(F(X); G(Y)). \end{aligned}$$

In the first inequality, $=_{\mathcal{M}}$ holds if and only if $\widetilde{G}(Y) \perp\!\!\!\perp \Phi_1(f(X), f^*(X)) \mid F(X)$, i.e.

$$\Phi_1(f(X), f^*(X)) \perp\!\!\!\perp \Phi_2(g(Y), g^*(Y)) \mid f(X),$$

drawing the contradiction to the construction of $\Phi$, thus the first inequality is strict, i.e.

$$-I(\widetilde{F}(X); \widetilde{G}(Y)) \prec_{\mathcal{M}} -I(F(X); G(Y)).$$

Since $\phi$ and $\psi$ are injective, we further have $-I(\widetilde{F}(X); \widetilde{G}(Y)) \prec_{\mathcal{M}} -I(F(X); G(Y)) =_{\mathcal{M}} -I(f(X); g(Y))$, which draws the contradiction to the fact that $(f, g) \in \mathcal{W}(\mathcal{H})$, thus $\mathtt{ID}(f) \ge k^*$.

## D.2 Proof of Step 2

As we already show that $\mathtt{ID}(f), \mathtt{ID}(g) \ge k^*$, let $\mathtt{ID}(f) = k \ge k^*$. If $k^* = d$, we naturally have $k = k^*$, thus we consider the case where $k^* < d$ and there exist injective measurable functions $\phi^*, \psi^* : \mathbb{R}^{k^*} \to \mathbb{R}^d$ and $F^*(X), G^*(Y) \in \mathbb{R}^{k^*}$ such that

$$f^*(X) = \phi^*\left(F^*(X)\right) \qquad g^*(Y) = \psi^*\left(G^*(Y)\right).$$

By Lemma 7, for any $(f^*, g^*) \in \mathcal{V}(\mathcal{H})$, it holds that $\mathtt{ID}(f^*), \mathtt{ID}(g^*) \le k^*$ and

$$f(X) \perp\!\!\!\perp g(Y) \mid f^*(X). \tag{12}$$

Hence, we can define

$$f_*(X) = \mathbb{E}[f(X) \mid f^*(X)].$$

Then, for any $\tau > 0$, consider the infoNCE loss

$$\mathcal{L}(f, g, \tau) = -\frac{2}{\tau} \frac{\mathbb{E}\left[\langle f(X), g(Y)\rangle\right]}{\mathbb{E}\|f(X)\|\mathbb{E}\|g(Y)\|} + \mathbb{E}_X \left\{ \log \mathbb{E}_{\widetilde{Y}} \left[ \exp\left( \frac{\langle f(X), g(\widetilde{Y})\rangle}{\tau\mathbb{E}\|f(X)\|\mathbb{E}\|g(\widetilde{Y})\|} \right) \right] \right\}$$
$$+ \mathbb{E}_{\widetilde{Y}} \left\{ \log \mathbb{E}_X \left[ \exp\left( \frac{\langle f(X), g(\widetilde{Y})\rangle}{\tau\mathbb{E}\|f(X)\|\mathbb{E}\|g(\widetilde{Y})\|} \right) \right] \right\}.$$

In addition, with the defined representation maps $(f_*, g)$, we further have

$$\mathcal{L}(f_*, g, \tau) = -\frac{2}{\tau} \frac{\mathbb{E}\left[\langle f_*(X), g(Y)\rangle\right]}{\mathbb{E}\|f_*(X)\|\mathbb{E}\|g(Y)\|} + \mathbb{E}_X \left\{ \log \mathbb{E}_{\widetilde{Y}} \left[ \exp\left( \frac{\langle f_*(X), g(\widetilde{Y})\rangle}{\tau\mathbb{E}\|f_*(X)\|\mathbb{E}\|g(\widetilde{Y})\|} \right) \right] \right\}$$
$$+ \mathbb{E}_{\widetilde{Y}} \left\{ \log \mathbb{E}_X \left[ \exp\left( \frac{\langle f_*(X), g(\widetilde{Y})\rangle}{\tau\mathbb{E}\|f_*(X)\|\mathbb{E}\|g(\widetilde{Y})\|} \right) \right] \right\},$$

where, by Jensen's inequality, $\|f_*(X)\| = \|\mathbb{E}[f(X) \mid f^*(X)]\| \le \mathbb{E}[\|f(X)\| \mid f^*(X)]$ almost surely.

We note that, according to Lemma 10, it holds that

$$\langle f(X), g(Y)\rangle = m_\sigma(f, g) \cdot \mathbb{E}\|f(X)\|\mathbb{E}\|g(Y)\| \qquad \text{almost surely.}$$

In addition, by the conditional independence (12), we further obtain

$$\mathbb{E}\left\{\langle \mathbb{E}[f(X) \mid f^*(X)], g(Y)\rangle\right\} = \mathbb{E}\left\{\mathbb{E}[\langle f(X), g(Y)\rangle \mid f^*(X)]\right\} = m_\sigma(f, g) \cdot \mathbb{E}\|f(X)\|\mathbb{E}\|g(Y)\|$$

almost surely and, in the meantime,

$$\mathbb{E}_X \left\{ \log \mathbb{E}_{\widetilde{Y}} \left[ \exp\left( \frac{\langle f(X), g(\widetilde{Y})\rangle}{\tau\mathbb{E}\|f(X)\|\mathbb{E}\|g(\widetilde{Y})\|} \right) \right] \right\}$$
$$= \mathbb{E}_{f^*(X)} \left\{ \mathbb{E} \left[ \log \mathbb{E}_{g(\widetilde{Y})} \left\{ \exp\left( \frac{\langle f(X), g(\widetilde{Y})\rangle}{\tau\mathbb{E}\|f(X)\|\mathbb{E}\|g(\widetilde{Y})\|} \right) \right\} \;\middle|\; f^*(X) \right] \right\}.$$

Since $\log \mathbb{E}_U \exp(x^\top U)$ is convex in $x$, by Jensen's inequality, we obtain

$$\mathbb{E}_X \left\{ \log \mathbb{E}_{\widetilde{Y}} \left[ \exp\left( \frac{\langle f(X), g(\widetilde{Y})\rangle}{\tau\mathbb{E}\|f(X)\|\mathbb{E}\|g(\widetilde{Y})\|} \right) \right] \right\}$$
$$\ge \mathbb{E}_{f^*(X)} \left\{ \log \mathbb{E}_{g(\widetilde{Y})} \left\{ \exp\left( \frac{\langle \mathbb{E}[f(X) \mid f^*(X)], g(\widetilde{Y})\rangle}{\tau\mathbb{E}\|f(X)\|\mathbb{E}\|g(\widetilde{Y})\|} \right) \right\} \right\}$$
$$= \mathbb{E}_X \left\{ \log \mathbb{E}_{\widetilde{Y}} \left[ \exp\left( \frac{\langle f_*(X), g(\widetilde{Y})\rangle}{\tau\mathbb{E}\|f(X)\|\mathbb{E}\|g(\widetilde{Y})\|} \right) \right] \right\}.$$

In addition, since $\exp(x^\top U)$ is convex in $x$, we also obtain

$$\mathbb{E}_X \left[ \exp\left( \frac{\langle f(X), g(\widetilde{Y})\rangle}{\tau\mathbb{E}\|f(X)\|\mathbb{E}\|g(\widetilde{Y})\|} \right) \right] \ge \mathbb{E}_{f^*(X)} \left[ \exp\left( \frac{\langle \mathbb{E}\left[f(X) \mid f^*(X)\right], g(\widetilde{Y})\rangle}{\tau\mathbb{E}\|f(X)\|\mathbb{E}\|g(\widetilde{Y})\|} \right) \right]$$

Combining the pieces above, if we define

$$\widetilde{\tau} = \tau \cdot \frac{\mathbb{E}\|f_*(X)\|}{\mathbb{E}\|f(X)\|} \le \tau,$$

it holds that

$$\mathcal{L}(f_*, g, \tau) \leq \mathcal{L}(f, g, \widetilde{\tau}),$$

where the equality holds if and only if $f(X)$ can be expressed as a measurable function of $f^*(X)$ almost surely. Thus, if $f(X)$ cannot be expressed as functions of $f^*(X)$, we have

$$\mathcal{L}(f, g, \widetilde{\tau}) > \mathcal{L}(f_*, g, \tau). \tag{13}$$

Then, according to Lemma 10 (3), since $\tau \geq \widetilde{\tau} \geq 0$, it holds that

$$\Delta(f, g, \tau) \geq \Delta(f, g, \widetilde{\tau}),$$

which, by Lemma 2 further implies that

$$\limsup_{M\to+\infty} \big(\mathcal{L}(f_M, g_M, \tau) + 2I_M^*(\mathcal{H})\big) \geq \limsup_{M\to+\infty} \big(\mathcal{L}(f_M, g_M, \widetilde{\tau}) + 2I_M^*(\mathcal{H})\big).$$

According to (13) and Lemma 5, we further obtain

$$\begin{aligned}\liminf_{M\to+\infty} \big(\mathcal{L}(f_M, g_M, \widetilde{\tau}) - \mathcal{L}((f_*)_M, g_M, \tau)\big) &\geq \liminf_{M\to+\infty} \mathcal{L}(f_M, g_M, \widetilde{\tau}) - \limsup_{M\to+\infty} \mathcal{L}((f_*)_M, g_M, \tau) \\ &= \lim_{M\to+\infty} \mathcal{L}(f_M, g_M, \widetilde{\tau}) - \lim_{M\to+\infty} \mathcal{L}((f_*)_M, g_M, \tau) > 0.\end{aligned}$$

Hence, it holds that

$$\begin{aligned}\limsup_{M\to+\infty} \big(\mathcal{L}(f_M, g_M, \tau) + 2I_M^*(\mathcal{H})\big) &\geq \limsup_{M\to+\infty} \big(\mathcal{L}(f_M, g_M, \widetilde{\tau}) + 2I_M^*(\mathcal{H})\big) \\ &\geq \limsup_{M\to+\infty} \big(\mathcal{L}((f_*)_M, g_M, \tau) + 2I_M^*(\mathcal{H})\big) \\ &\qquad + \liminf_{M\to+\infty} \big(\mathcal{L}(f_M, g_M, \widetilde{\tau}) - \mathcal{L}((f_*)_M, g_M, \tau)\big) \\ &> \limsup_{M\to+\infty} \big(\mathcal{L}((f_*)_M, g_M, \tau) + 2I_M^*(\mathcal{H})\big).\end{aligned}$$

Particularly, for any $\eta > 0$, with $\tau = \varepsilon(\eta)$, it holds that

$$\begin{aligned}\limsup_{M\to+\infty} \big(\mathcal{L}(f_M, g_M, \varepsilon(\eta)) + 2I_M^*(\mathcal{H})\big) &> \limsup_{M\to+\infty} \big(\mathcal{L}((f_*)_M, g_M, \varepsilon(\eta)) + 2I_M^*(\mathcal{H})\big) \\ &\geq 2\eta, \qquad \text{for all } \eta > 0.\end{aligned}$$

Then, by the definition of $\mathcal{O}_{\mathcal{L},\eta}(\mathcal{H})$, we can conclude that $(f, g) \notin \mathcal{O}_{\mathcal{L},\eta}(\mathcal{H})$, which draws the contradiction. Hence, there exist measurable functions $\widetilde{h}$ and $\widetilde{\ell}$ such that

$$f(X) = (h \circ f^*)(X) \qquad g(Y) = (\ell \circ g^*)(Y) \qquad \text{almost surely.}$$

By Lemma 6, it holds that $\texttt{ID}(f), \texttt{ID}(g) \leq k^*$, and combining with the result in Step 1, we obtain $\texttt{ID}(f) = \texttt{ID}(g) = k^*$, which concludes the proof.

# E Extensions and additional theoretical results

## E.1 Connection with sufficient dimension reduction

Sufficient dimension reduction (SDR) [38, 10, 11] is an important topic in statistics and machine learning, in which, the goal is to find a low-dimensional representation $f(X)$ such that $f(X)$ is sufficient for the conditional distribution of $Y \mid X$ in the sense that $Y \perp\!\!\!\perp X \mid f(X)$. In the following proposition, we will show that the representations learned by CLIP are also sufficient if $\mathcal{V}(\mathcal{H}) \neq \varnothing$. To simplify notation, we focus on the case where $X, Y \in \mathbb{R}^d$. Extensions to a more general setting, where $X \in \mathbb{R}^{d_1}$ and $Y \in \mathbb{R}^{d_2}$, are provided in Appendix E.1.2. Here we denote $\texttt{id} : \mathbb{R}^d \to \mathbb{R}^d$ as the identical map and $X_M = \texttt{id}_M(X)$. Then, we have the following result.

**Proposition 2.** *Assume $\mathcal{V}(\mathcal{H}) \neq \varnothing$ and there exists $(f,g) \in \mathcal{H}$ such that $\liminf_{M\to+\infty}(I(f_M(X); g_M(Y)) - I(X_M; Y_M)) = 0$. For any $(f^*, g^*) \in \mathcal{V}(\mathcal{H}^{k^*})$, the dimension reductions $f^*(X)$ and $g^*(Y)$ are sufficient, i.e. $Y \perp\!\!\!\perp X \mid f^*(X)$ and $Y \perp\!\!\!\perp X \mid g^*(Y)$.*

We note that compared with Theorem 2, Proposition 2 has an additional condition that there exists $(f,g) \in \mathcal{H}$ such that $\liminf_{M\to+\infty}(I(f_M(X); g_M(Y)) - I(X_M; Y_M)) = 0$. One specific example that satisfies this condition is the scenario where we set $f^*(X) = (h_1(X), \cdots, h_k(X))$ almost surely such that $Y \perp\!\!\!\perp X \mid (h_1(X), \cdots, h_k(X))$, which is commonly considered in the literature of (nonlinear) SDR [15, 16, 36, 8] and the tuple of nonlinear functions can be further extended to $\sigma$-field [37, 8].

### E.1.1 Proof of Proposition 2

We prove by showing that $Y \perp\!\!\!\perp X \mid f^*(X)$. By the assumption that there exists $(\widetilde{f}, \widetilde{g}) \in \mathcal{H}$ such that $I(Y; X) =_{\mathcal{M}} I(\widetilde{f}(X); \widetilde{g}(Y))$ and the definition of $\mathcal{V}(\mathcal{H}^{k^*})$, it holds that $I(Y;X) =_{\mathcal{M}} I(f^*(X); g^*(Y))$ for all $(f^*, g^*) \in \mathcal{V}(\mathcal{H}^{k^*})$. Then, by property of mutual information ([58] Theorem 2.3), we have

$$I(g^*(Y); f^*(X)) \preceq_{\mathcal{M}} I(Y; f^*(X)) \preceq_{\mathcal{M}} I(Y;X) =_{\mathcal{M}} I(g^*(Y); f^*(X)),$$

which implies that

$$I(Y; f^*(X)) =_{\mathcal{M}} I(Y;X). \tag{14}$$

In addition, combining (14) with the Kolmogorov identity of mutual information ([58] Theorem 2.5 (2)) that

$$I(Y;X) =_{\mathcal{M}} I(Y; X, f^*(X)) =_{\mathcal{M}} I(Y; f^*(X)) + I(Y; X \mid f^*(X)),$$

we further obtain $I(Y; X \mid f^*(X)) =_{\mathcal{M}} 0$. According to the property of the conditional mutual information ([58] Theorem 2.5 (1)), $I(Y; X \mid f^*(X)) =_{\mathcal{M}} 0$ if and only if $(X_M, f^*_M(X), Y_M)$ forms a Markov chain, i.e. $Y_M \perp\!\!\!\perp X_M \mid f^*_M(X)$ for all $M \in \mathcal{M}$, which, by continuity of $X$, $Y$, and $f^*(X)$, further indicates that $Y \perp\!\!\!\perp X \mid f^*(X)$.

### E.1.2 Extension of Proposition 2

We then turn to a general case where $X \in \widetilde{\mathcal{B}}_{d_1}$ and $Y \in \widetilde{\mathcal{B}}_{d_2}$, where $\widetilde{B}_{d_1}$ and $\widetilde{\mathcal{B}}_{d_2}$ are bounded sets in $\mathbb{R}^{d_1}$ and $\mathbb{R}^{d_2}$, respectively. If $\mathcal{V}(\mathcal{H}) \neq \varnothing$, we fix a pair $(f^*, g^*) \in \mathcal{V}(\mathcal{H}^{k^*})$. Consider discretizations $\mathcal{D}_M = \{D_{M,i}\}_{i\in[M]}$ and $\mathcal{E}_M = \{E_{M,i}\}_{i\in[M]}$ of $\widetilde{\mathcal{B}}_{d_1}$ and $\widetilde{\mathcal{B}}_{d_2}$, respectively, where $D_{M,i}$ is the preimage of $c_{M,i}$ under $f^*$ and $E_{M,i}$ is the preimage of $c_{M,i}$ under $g^*$. Denote representatives in $D_{M,i}$ and $E_{M,i}$ by $x_{M,i}$ and $y_{M,i}$, respectively. In addition, we assume $\mathcal{D}_M$ and $\mathcal{E}_M$ are nested discretizations and as $M \to +\infty$, it holds that

$$\lim_{M\to+\infty} \max_{i\in[M]} \operatorname{diam}(D_{M,i}) = 0, \qquad \lim_{M\to+\infty} \max_{i\in[M]} \operatorname{diam}(E_{M,i}) = 0.$$

Similar to notations for $\widetilde{B}_d$ before, we write $X_M$ and $Y_M$ to denote discretized random vectors where

$$\mathbb{P}(X_M = x_{M,i}) = \mathbb{P}(X \in D_{M,i}) \qquad \text{and} \qquad \mathbb{P}(Y_M = y_{M,i}) = \mathbb{P}(Y \in E_{M,i}).$$

Then, we define a new partial order $\overset{*}{\preceq}_{\mathcal{M}}$ as follows. For any random vectors $X \in \mathbb{R}^{d_1}$, $Y \in \mathbb{R}^{d_2}$, $U$ supported on $\widetilde{B}_d$, and functionals $F, G$ of random vectors, we define the order $\overset{*}{\preceq}_{\mathcal{M}}$ associated with $(f^*, g^*)$ such that

$$F(X,Y) \overset{*}{\preceq}_{\mathcal{M}} G(U) \iff \limsup_{M\to+\infty} \{F(X_M, Y_M) - G(U_M)\} \leq 0,$$

and moreover, the inequality is strict, i.e. $F(X,Y) \overset{*}{\prec}_{\mathcal{M}} G(U)$, if and only if

$$\limsup_{M\to+\infty} \left\{ F(X_M, Y_M) \overset{*}{\preceq}_{\mathcal{M}} G(U_M) \right\} < 0.$$

We also write $F(X,Y) \overset{*}{=}_{\mathcal{M}} G(U)$ if $\lim_{M\to+\infty} \{F(X_M, Y_M) - G(U_M)\} = 0$. Then, we state the extension of Proposition 2 as follows.

**Proposition 3.** *Suppose $\mathcal{V}(\mathcal{H}) \neq \varnothing$. Assume there exists $(f, g) \in \mathcal{H}$ such that $I(X;Y) \overset{*}{=}_{\mathcal{M}} I(f(X); g(Y))$ with $\overset{*}{=}_{\mathcal{M}}$ associated with $(f^*, g^*) \in \mathcal{V}(\mathcal{H}^{k^*})$. Then, the dimension reductions $f^*(X)$ and $g^*(Y)$ are sufficient, i.e.*

$$Y \perp\!\!\!\perp X \mid f^*(X) \qquad \text{and} \qquad Y \perp\!\!\!\perp X \mid g^*(Y).$$

*Proof of Proposition 3.* Similar with the proof of Proposition 2, for any $(f^*, g^*) \in \mathcal{V}(\mathcal{H}^{k^*})$, it holds that

$$I(g^*(Y); f^*(X)) \overset{*}{\preceq}_{\mathcal{M}} I(Y; f^*(X)) \overset{*}{\preceq}_{\mathcal{M}} I(Y; X).$$

In addition, by assumption, there exists $(\widetilde{f}, \widetilde{g}) \in \mathcal{H}$ such that

$$I(Y; X) \overset{*}{=}_{\mathcal{M}} I(\widetilde{f}(X); \widetilde{g}(Y)) \overset{*}{\preceq}_{\mathcal{M}} I(g^*(Y); f^*(X)).$$

Combining results above, we obtain

$$I(Y; X) \overset{*}{=}_{\mathcal{M}} I(g^*(Y); f^*(X)).$$

In addition, by the Kolmogorov identity of mutual information ([58] Theorem 2.5 (2)), we obtain

$$I(Y; X) \overset{*}{=}_{\mathcal{M}} I(Y; X, f^*(X)) \overset{*}{=}_{\mathcal{M}} I(Y; f^*(X)) + I(Y; X \mid f^*(X)),$$

which implies that $I(Y; X \mid f^*(X)) \overset{*}{=}_{\mathcal{M}} 0$. According to the property of the conditional mutual information ([58] Theorem 2.5 (1)), $I(Y; X \mid f^*(X)) \overset{*}{=}_{\mathcal{M}} 0$ if and only if $(X_M, f^*_M(X), Y_M)$ forms a Markov chain, i.e. $Y_M \perp\!\!\!\perp X_M \mid f^*_M(X)$ for all $M \in \mathcal{M}$, which, by continuity of $X$, $Y$, and $f^*(X)$, further indicates that $Y \perp\!\!\!\perp X \mid f^*(X)$. $\square$

### E.2 Alignment and uniformity with correctly specified dimension

To begin with, recall that for any $(f, g) \in \mathcal{A}(\mathcal{H})$, it holds that $f(X)/\mathbb{E}\|f(X)\|, g(Y)/\mathbb{E}\|g(Y)\| \in \mathcal{S}^{d-1}$. Then, we define the set of uniformly distributed representations by

$$\mathcal{U}_p(\mathcal{H}') = \left\{(f, g) \in \mathcal{H}' : \ f(X)/\mathbb{E}\|f(X)\|, g(Y)/\mathbb{E}\|g(Y)\| \sim \mathrm{Unif}(\mathcal{S}^{p-1})\right\}.$$

Note that for any representation map $f \in \mathcal{H}'$ with $\mathtt{ID}(f) = k$, if the function class is sufficiently expressive, the linear dimension of $R(f)$ can exceed $k$, then recalling the definition of $\mathcal{A}(\mathcal{H}^k)$ and $\mathcal{U}_p(\mathcal{H}^k)$, we define

$$D(k, \mathcal{H}) = \max\left\{p \in [d] : \ \mathcal{A}(\mathcal{H}^k) \cap \mathcal{U}_p(\mathcal{H}^k) \neq \varnothing\right\}.$$

Specifically, write $d^* = D(k^*, \mathcal{H})$. Then, we define a subset of $\mathcal{H}$ by $\mathcal{H}^* = \mathcal{H}^*_X \times \mathcal{H}^*_Y$ with

$$\mathcal{H}^*_X = \left\{f_{\mathcal{S}^{d^*-1}} : f \in \mathcal{H}_X, \ f(X)/\mathbb{E}\|f(X)\| \in \mathcal{S}^{d^*-1}\right\},$$
$$\mathcal{H}^*_Y = \left\{g_{\mathcal{S}^{d^*-1}} : g \in \mathcal{H}_Y, \ g(Y)/\mathbb{E}\|g(Y)\| \in \mathcal{S}^{d^*-1}\right\}.$$

We further denote $\mathcal{U}(\mathcal{H}^*) = \{(f, g) \in \mathcal{H}^* : \ f(X)/\mathbb{E}\|f(X)\|, g(Y)/\mathbb{E}\|g(Y)\| \sim \mathrm{Unif}(\mathcal{S}^{d^*-1})\}$.

Then, by specifying the output dimension $d = d^*$, we have the following result.

**Theorem 3.** *With the function class $\mathcal{H} = \mathcal{H}^*$ and output dimension $d = d^* = D(k^*, \mathcal{H})$, it holds that*

$$\bigcap_{\eta \geq 0} O_{\mathcal{L},\eta}(\mathcal{H}^*) = \mathcal{V}(\mathcal{H}^*) \cap \mathcal{U}(\mathcal{H}^*).$$

*Particularly, for any $(f^*, g^*) \in \mathcal{V}(\mathcal{H}^*) \cap \mathcal{U}(\mathcal{H}^*)$, the tuple $(f^*, g^*, 0^+)$ is a minimizer of $\mathcal{L}(f, g, \tau)$.*

### E.2.1 Proof of Theorem 3

By [79, Theorem 1 (2)], if we define $\widetilde{f}(X) = f(X)/\mathbb{E}\|f(X)\|$ and $\widetilde{g}(Y) = g(Y)/\mathbb{E}\|g(Y)\|$, then $(\widetilde{f}, \widetilde{g})$ is the global minimizer of $\mathcal{L}$ in $\mathcal{H}^*$ for any given $\tau > 0$. Then, based on the definition of $\mathcal{O}_\eta(\mathcal{H})$, it holds that

$$\mathcal{A}(\mathcal{H}^*) \cap \mathcal{U}(\mathcal{H}^*) = \bigcap_{\eta \geq 0} O_{\mathcal{L},\eta}(\mathcal{H}^*).$$

In addition, with the choice of $\mathcal{H}^*$, the uniformly distributed representation on $\mathcal{S}^{d^*-1}$ maximizes the entropy. Hence, for any $(f, g) \in \mathcal{A}(\mathcal{H}^*) \cap \mathcal{U}(\mathcal{H}^*)$, we also have $(f, g) \in \mathcal{W}(\mathcal{H}^*)$, thus $\mathcal{A}(\mathcal{H}^*) \cap \mathcal{U}(\mathcal{H}^*) = \mathcal{A}(\mathcal{H}^*) \cap \mathcal{U}(\mathcal{H}^*) \cap \mathcal{W}(\mathcal{H}^*) = \mathcal{V}(\mathcal{H}^*) \cap \mathcal{U}(\mathcal{H}^*)$, which completes the proof.

# F Proof of preliminary results in Section A.2

## F.1 Proof of Lemma 1

We adapt the proof in [12, Theorem 8.3.1]. The entropy of $U_M$ can be written as

$$\begin{aligned} H(U_M) &= -\sum_{i \in [M]} p_i \log p_i \\ &= -\sum_{i \in [M]} \left( \int_{c_{M,i}} p(u) \mathsf{d}u \right) \log \left( \int_{c_{M,i}} p(u) \mathsf{d}u \right) \\ &= -\sum_{i \in [M]} \left[ p(z_{M,i}) \log p(z_{M,i}) \right] \Delta_M - \log \Delta_M + o(1). \end{aligned}$$

Hence, it holds that

$$\lim_{M \to +\infty} \left( H(U_M) + \log \Delta_M \right) = \widetilde{H}(U).$$

## F.2 Proof of Lemma 3

We note that

$$H(\phi(U_M, V_M) \mid U_M) = H(\phi(U_M, V_M), U_M) - H(U_M),$$

for which, we already see in Lemma 1 that

$$\lim_{M \to +\infty} \left( H(U_M) + \log \Delta_M \right) = \widetilde{H}(U).$$

Denoting $W = \phi(U, V)$, $\zeta = \Phi(U, V) = (\phi(U, V), U)$, $w_{M,ij} = \phi(z_{M,i}, z_{M,j})$, and $\zeta_{M,ij} = (w_{M,ij}, z_{M,i})$. In addition, we write

$$q_{ij} = \mathbb{P}(U \in c_{M,i}, V \in c_{M,j}), \qquad q_{ij}^{\Phi} = \mathbb{P}(\zeta \in \Phi(c_{M,i} \times c_{M,j})),$$

and $\Delta^{\phi}(z_{M,i}, z_{M,j})$ as the Lebesgue measure of $\phi(c_{M,i}, c_{M,j})$, $\Delta_M$ as the Lebesgue measure of $c_{M,i}$. Define $q^{\Phi}(w, u)$ as the joint density of $\Phi(U, V) = (\phi(U, V), U)$.

The joint entropy takes the form

$$
\begin{aligned}
&H(\phi(U_M,V_M),U_M)\\
&\quad= -\sum_{i,j\in[M]} q_{ij}^{\Phi}\log q_{ij}^{\Phi}\\
&\quad= -\sum_{i,j\in[M]} q^{\Phi}(w_{M,ij},z_{M,i})\Delta_M^{\phi}(z_{M,i},z_{M,j})\Delta_M\\
&\qquad\quad \times\log\left(q^{\Phi}(w_{M,ij},z_{M,i})\Delta_M^{\phi}(z_{M,i},z_{M,j})\Delta_M\right)+o(1)\\
&\quad= -\sum_{i,j\in[M]} q^{\Phi}(w_{M,ij},z_{M,i})\Delta_M^{\phi}(z_{M,i},z_{M,j})\Delta_M\log\left(q^{\Phi}(w_{M,ij},z_{M,i})\right)\\
&\qquad\quad -\sum_{i,j\in[M]} q^{\Phi}(w_{M,ij},z_{M,i})\Delta_M^{\phi}(z_{M,i},z_{M,j})\Delta_M\log\left(\Delta_M^{\phi}(z_{M,i},z_{M,j})\Delta_M\right)+o(1)\\
&\quad= -\sum_{i,j\in[M]} q^{\Phi}(w_{M,ij},z_{M,i})\Delta_M^{\phi}(z_{M,i},z_{M,j})\Delta_M\log\left(q^{\Phi}(w_{M,ij},z_{M,i})\right)\\
&\qquad\quad -\sum_{i,j\in[M]} q^{\Phi}(w_{M,ij},z_{M,i})\Delta_M^{\phi}(z_{M,i},z_{M,j})\Delta_M\log\Delta_M^{\phi}(z_{M,i},z_{M,j})\\
&\qquad\quad -\log\Delta_M+o(1).
\end{aligned}
$$

Then, by the results in Lemma 1, we obtain

$$
\begin{aligned}
&H(\phi(U_M,V_M),U_M)-H(U_M)\\
&\quad= \underbrace{-\sum_{i,j\in[M]} q^{\Phi}(w_{M,ij},z_{M,i})\Delta_M^{\phi}(z_{M,i},z_{M,j})\Delta_M\log\left(q^{\Phi}(w_{M,ij},z_{M,i})\right)}_{(a)}\\
&\quad+\underbrace{\sum_{i\in[M]}\left[p(z_{M,i})\Delta_M\log p(z_{M,i})\right]}_{(b)}\\
&\qquad -\sum_{i,j\in[M]} q^{\Phi}(w_{M,ij},z_{M,i})\Delta_M^{\phi}(z_{M,i},z_{M,j})\Delta_M\log\Delta_M^{\phi}(z_{M,i},z_{M,j})+o(1),
\end{aligned}
$$

where, as $M\to+\infty$ and $\operatorname{diam}(c_{M,i})\to 0$, we have

$$
\begin{cases}
\text{(a)}\to -\widetilde{H}(\phi(U,V),U),\\
\text{(b)}\to -\widetilde{H}(U).
\end{cases}
$$

In addition, as $\Delta_M^{\phi}(z_{M,i},z_{M,j})<1$,

$$
\limsup_{M\to+\infty}\sum_{i,j\in[M]} q^{\Phi}(w_{M,ij},z_{M,i})\Delta_M^{\phi}(z_{M,i},z_{M,j})\Delta_M\log\Delta_M^{\phi}(z_{M,i},z_{M,j})<0
$$

Hence, we obtain

$$
\begin{aligned}
&\liminf_{M\to+\infty}\left(H(\phi(U_M,V_M),U_M)-H(U_M)\right)\\
&\quad\geq \widetilde{H}(\phi(U,V),U)-\widetilde{H}(U)\\
&\qquad -\limsup_{M\to+\infty}\sum_{i,j\in[M]} q^{\Phi}(w_{M,ij},z_{M,i})\Delta_M^{\phi}(z_{M,i},z_{M,j})\Delta_M\log\Delta_M^{\phi}(z_{M,i},z_{M,j})\\
&\quad> \widetilde{H}(\phi(U,V),U)-\widetilde{H}(U)=\widetilde{H}(\phi(U,V)\mid U)>0.
\end{aligned}
$$

### F.3 Proof of Lemma 4

Denoting $R = \phi(U,V)$, $W = \psi(U,V)$, and $r_{M,ij} = \phi(z_{M,i}, z_{M,j})$, $w_{M,ij} = \psi(z_{M,i}, z_{M,j})$, we note that

$$I(\phi(U,V);\psi(U,V)) = \iint p_{R,W}(z,w)\frac{p_{R,W}(r,w)}{p_R(r)p_W(w)}\mathsf{d}z\mathsf{d}w.$$

Denote $\Phi(U,V) = (\phi(U,V),\psi(U,V))$ and $q_{ij}^{\phi} = \mathbb{P}(\phi(U,V) \in \phi(c_{M,i} \times c_{M,j}))$, $q_{ij}^{\psi} = \mathbb{P}(\psi(U,V) \in \psi(c_{M,i} \times c_{M,j}))$, $q_{ij}^{\Phi} = \mathbb{P}(\Phi(U,V) \in \Phi(c_{M,i} \times c_{M,j}))$. In addition, denote $\Delta_M^{\Phi}(r_{M,ij}, w_{M,ij})$ as the Lebesgue measure of $\Phi(c_{M,i} \times c_{M,j})$, $\Delta_M^{\phi}(r_{M,ij}, w_{M,ij})$ as the Lebesgue measure of $\phi(c_{M,i} \times c_{M,j})$, and $\Delta_M^{\psi}(r_{M,ij}, w_{M,ij})$ as the Lebesgue measure of $\psi(c_{M,i} \times c_{M,j})$. Then, for the discretized mutual information, we have

$$\begin{aligned} &I(\phi(U_M,V_M);\psi(U_M,V_M)) \qquad (15)\\ &= \sum_{i,j\in[M]} p_{ij}^{\Phi}\Delta_M^{\Phi}(r_{M,ij},w_{M,ij})\log\frac{p_{ij}^{\Phi}\Delta_M^{\Phi}(r_{M,ij},w_{M,ij})}{p_{ij}^{\phi}p_{ij}^{\psi}\Delta_M^{\phi}(r_{M,ij})\Delta_M^{\psi}(w_{M,ij})} + o(1). \end{aligned}$$

Since $\Delta_M^{\phi,\psi}(r_{M,i}, w_{M,j}) = \Delta_M^{\phi}(r_{M,i})\Delta_M^{\psi}(w_{M,j}) + o(1)$, (15) can be further written as

$$\begin{aligned} &I(\phi(U_M,V_M);\psi(U_M,V_M))\\ &= \sum_{i,j\in[M]} p_{ij}^{\Phi}\Delta_M^{\Phi}(r_{M,ij},w_{M,ij})\log\frac{p_{ij}^{\Phi}}{p_{ij}^{\phi}p_{ij}^{\psi}} + o(1)\\ &= \underbrace{\sum_{i,j\in[M]}\left(p_{ij}^{\Phi}\log p_{ij}^{\Phi}\right)\Delta_M^{\phi}(r_{M,ij})\Delta_M^{\psi}(w_{M,ij})}_{(a)}\\ &\qquad - \underbrace{\sum_{i,j\in[M]}\left(p_{ij}^{\Phi}\log\left(p_{ij}^{\phi}p_{ij}^{\psi}\right)\right)\Delta_M^{\phi}(r_{M,ij})\Delta_M^{\psi}(w_{M,ij})}_{(b)} + o(1). \end{aligned}$$

As $M \to +\infty$ and $\max_{i\in[M]} \operatorname{diam}(c_{M,i}) \to 0$, it holds that

$$\begin{aligned} &\text{(a)} \to -\widetilde{H}(R,W),\\ &\text{(b)} \to -H_C(P_{R,W}, P_R \otimes P_W), \end{aligned}$$

where $H_C(P,Q)$ denotes the cross-entropy of $Q$ relative to $P$. Hence, as $M \to +\infty$ and $\max_{i\in[M]} \operatorname{diam}(c_{M,i}) \to 0$,

$$\text{(a)} - \text{(b)} \to D_{\mathrm{KL}}(P_{R,W} \,||\, P_R \otimes P_W) = I(R;W).$$

### F.4 Proof of Lemma 5

Since infoNCE loss is scale-invariant with respect to both $f$ and $g$, without loss of generality, we assume $\mathbb{E}\|f(X)\| = \mathbb{E}\|g(Y)\| = 1$. For any fineness $M \in \mathcal{M} \subseteq \mathbb{N}$, recall the infoNCE loss

$$\begin{aligned} \mathcal{L}(f_M, g_M, \tau) = &-\frac{2}{\tau}\mathbb{E}\left\{\langle f_M(X), g_M(Y)\rangle\right\}\\ &+ \mathbb{E}_X\left\{\log\mathbb{E}_{\widetilde{Y}}\left[\exp\left(\frac{\langle f_M(X), g_M(\widetilde{Y})\rangle}{\tau}\right)\right]\right\}\\ &+ \mathbb{E}_{\widetilde{Y}}\left\{\log\mathbb{E}_X\left[\exp\left(\frac{\langle f_M(X), g_M(\widetilde{Y})\rangle}{\tau}\right)\right]\right\}. \end{aligned}$$

Similar to the proof of [79, Theorem 1], we have

$$\lim_{M\to+\infty}\mathbb{E}\left\{\langle f_M(X), g_M(Y)\rangle\right\} = \mathbb{E}\left\{\langle f(X), g(Y)\rangle\right\},$$

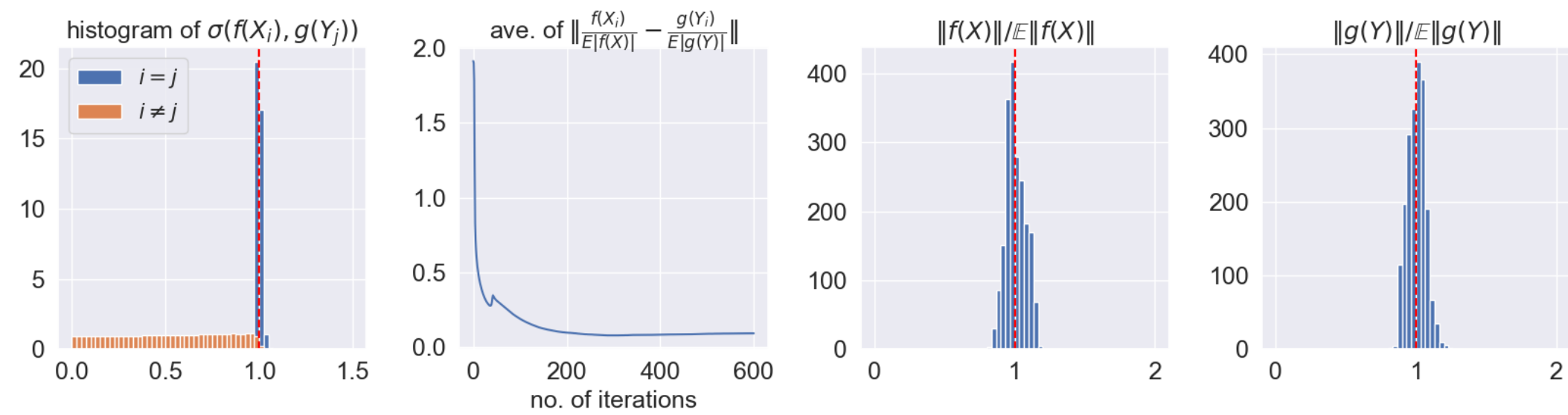

Figure 7: Histogram of similarities and norms of representations: $(d, k^*) = (3, 2)$.

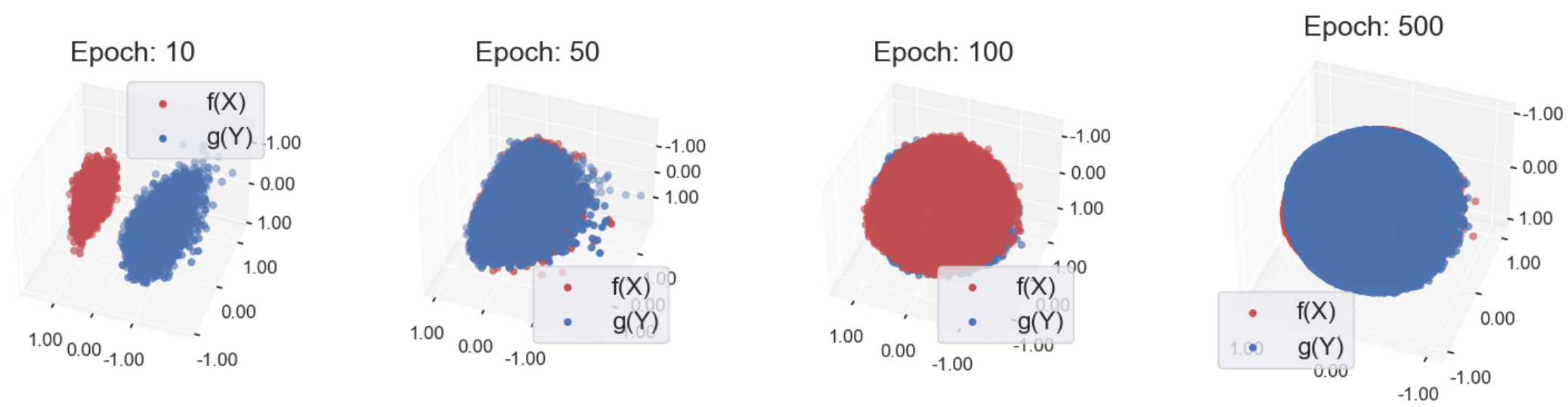

Figure 8: Scatterplots of out-of-sample representations: $(d, k^*) = (3, 2)$.

and, by the boundedness of $\exp(\langle f(X), g(\widetilde{Y})\rangle) \in [0, e^{\Omega}]$ and the dominated convergence theorem,

$$\begin{aligned}
&\lim_{M\to+\infty} \mathbb{E}_X \left\{ \log \mathbb{E}_{\widetilde{Y}} \left[ \exp\left( \frac{\langle f_M(X), g_M(\widetilde{Y})\rangle}{\tau} \right) \right] \right\} \\
&= \lim_{M\to+\infty} \mathbb{E}_X \left\{ \log \mathbb{E}_{\widetilde{Y}} \left[ \exp\left( \frac{\langle f(X), g_M(\widetilde{Y})\rangle}{\tau} \right) \right] \right\} \\
&= \mathbb{E}_X \left\{ \log \mathbb{E}_{\widetilde{Y}} \left[ \exp\left( \frac{\langle f(X), g(\widetilde{Y})\rangle}{\tau} \right) \right] \right\}.
\end{aligned}$$

Hence, it holds that

$$\lim_{M\to+\infty} \mathcal{L}(f_M, g_M, \tau) = \mathcal{L}(f, g, \tau).$$

# G Additional experimental results

In this section, we present additional results to Section 4 and we follow the same setting for each dataset in Section 4.

## G.1 Norm concentration

Following the discussion in Section 1.1, we consider the same setting with $(d, k^*) = (3, 2)$ (Figure 7) and $(d, k^*) = (20, 5)$ (Figure 9), respectively. In addition, for the setting with $d = 3$, we also present the scatterplot for out-of-sample representations along the training process in Figure 8.

From Figure 7 and Figure 9, it is clear that $m_\sigma(f, g) = 1$ and $\|f(X)\|/\mathbb{E}\|f(X)\|, \|g(Y)\|/\mathbb{E}\|g(Y)\| \approx 1$, which indicates that $(f, g) \in \mathcal{A}(\mathcal{H})$. This empirical finding is nontrivial in the sense that in our training, there is no constraint on $\|f(X)\|/\mathbb{E}\|f(X)\|$ and $\|g(Y)\|/\mathbb{E}\|g(Y)\|$, but, when the function class $\mathcal{H}$ is sufficiently expressive, the representations after population-level normalization tend to concentrate on the unit hypersphere automatically.

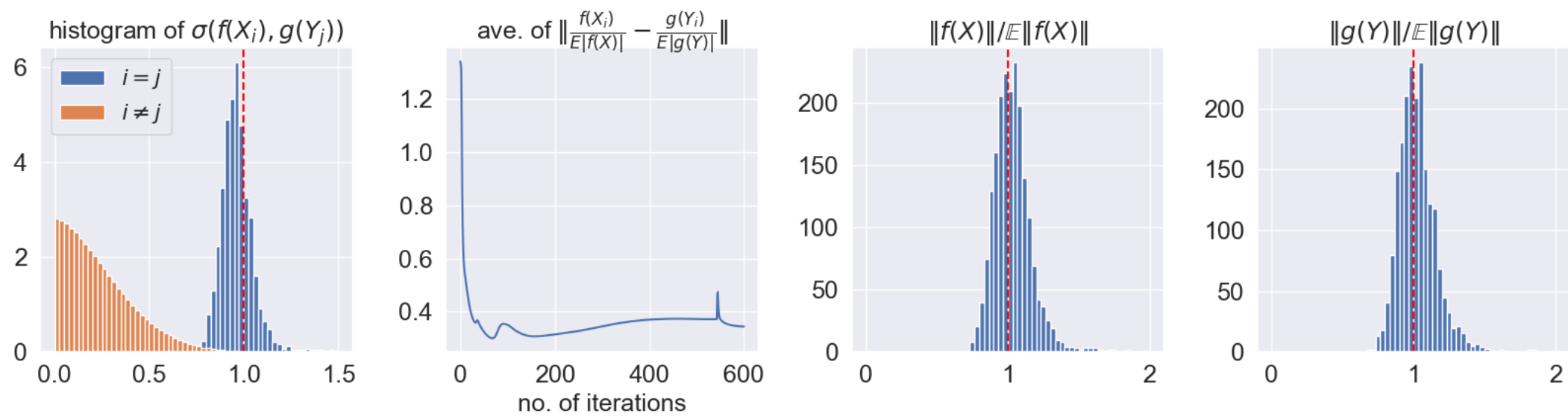

Figure 9: Histogram of similarities and norms of representations: $(d, k^*) = (20, 5)$.

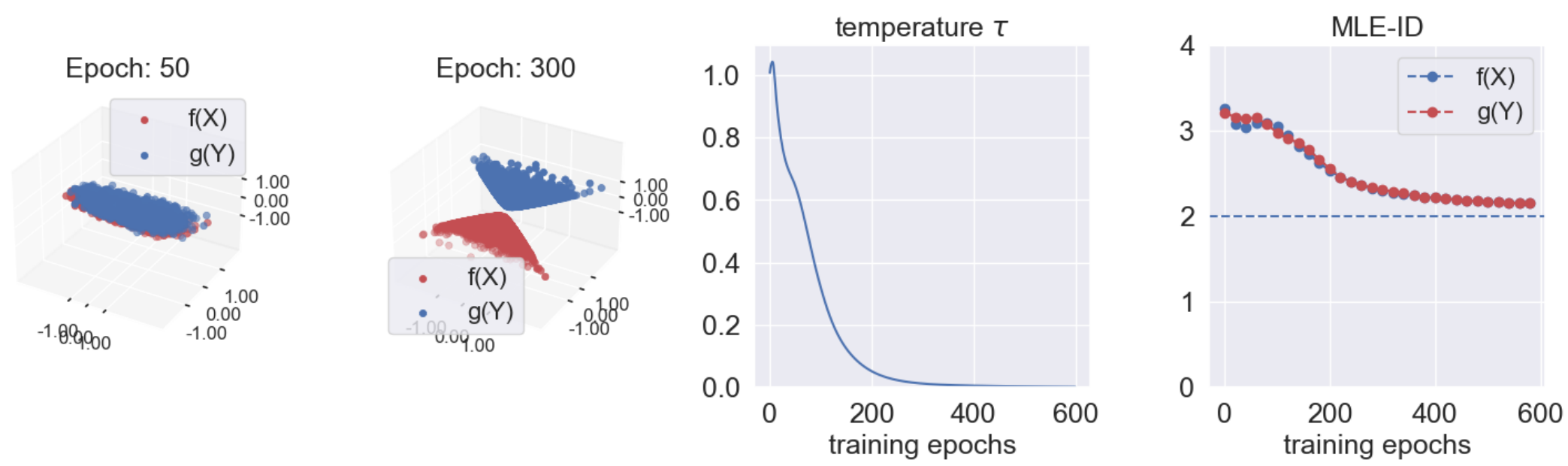

Figure 10: Results with two-layer ReLU: $m_\sigma(f, g) = 0$.

### G.2 Example with $\mathcal{V}(\mathcal{H}) = \varnothing$: alignment versus simply similarity maximization

To illustrate why the assumption $\mathcal{V}(\mathcal{H}) \neq \varnothing$ is important, in this section, we use empirical examples to show that CLIP minimizers may not be semantically meaningful if $\mathcal{V}(\mathcal{H}) = \varnothing$. We consider the same setting in Section 1.1 but instead adopt the function class with 2-layer ReLU neural networks with the width of the middle layer as 50, in which case $\mathcal{A}(\mathcal{H}) = \varnothing$. It is illustrated in Figure 10 that CLIP can lead to minimizers $(f, g)$ with $m_\sigma(f, g) = 0$, i.e., orthogonal representations. However, even in this undesired setting, the intrinsic dimension $k^* = 2$ is correctly specified.

### G.3 Experiment setup

We start with details of the experiment setup for real datasets. The same architecture is used in experiments: a 5-layer ReLU neural network with a width of middle-layer set to be 50 and varying input and output dimensions. The neural network is trained for 800 epochs with learning rate $10^{-4}$ and weight decay $10^{-4}$, and a slightly faster rate is used for temperature $\tau$: $10^{-3}$ in synthetic experiments and $2 \times 10^{-4}$ for real data.

**`CITE-seq` dataset.** We follow the preprocessing in https://satijalab.org/seurat/articles/weighted_nearest_neighbor_analysis, where we normalize ADT data with centering to produce a 24-dimension input, and normalize the extremely high-dimensional RNA data and extract the first 200 principal components as the inputs for CLIP.

For the downstream classification tasks, we consider two set of labels. `CITE-seq` provides annotations of cells at two levels of granularity [21]. For BMCs data, at level (1), 5 major cell populations include CD4 T cells, CD8 T cells, B cells, classical monocytes (CM), and natural killer (NK) cells. Cell populations are further divided into 27 finer subpopulations at level (2).

**`ImageNetV2` and `YFCC` dataset.** For the image-text datasets, we first use pretrained image and text encoders to transform images and texts to numerical inputs. Concretely, we adopt a pretrained `ViT-14L` from `openai/clip-vit-large-patch14`[9] (without projection) as the image encoder and a masked self-attention Transformer as the text encoder from the same pretrained model. Then,

[9] https://huggingface.co/openai/clip-vit-large-patch14.

with the pretrained encoders, we obtain 1024-dimensional image inputs and 768-dimensional text inputs.

### G.4 Convergence of temperature

We vary the output dimension and present the convergence of temperature for the following three datasets.

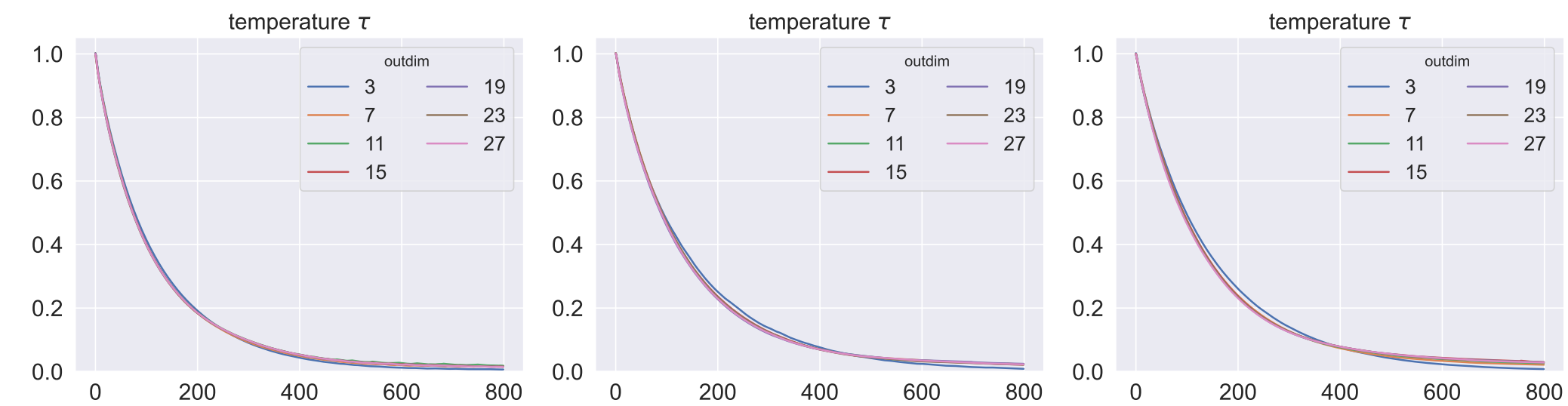

(a) CITE-seq dataset. (b) YFCC dataset. (c) ImageNetV2 dataset.

Figure 11: Convergence of temperature.

From Figure 11, we can see that with each real dataset and each choice of the output dimension, we have the temperature $\tau$ converging to zero, which is in line with our theory, and partially justifies that there are representations that can simultaneously maximize similarity and mutual information.

### G.5 Comparison with cosine similarity

In this section, we present the experiment results with cosine similarity. Results for the synthetic (linear and nonlinear) and CITE-seq datasets are shown in Figure 12a, Figure 12b, and Figure 13, respectively.

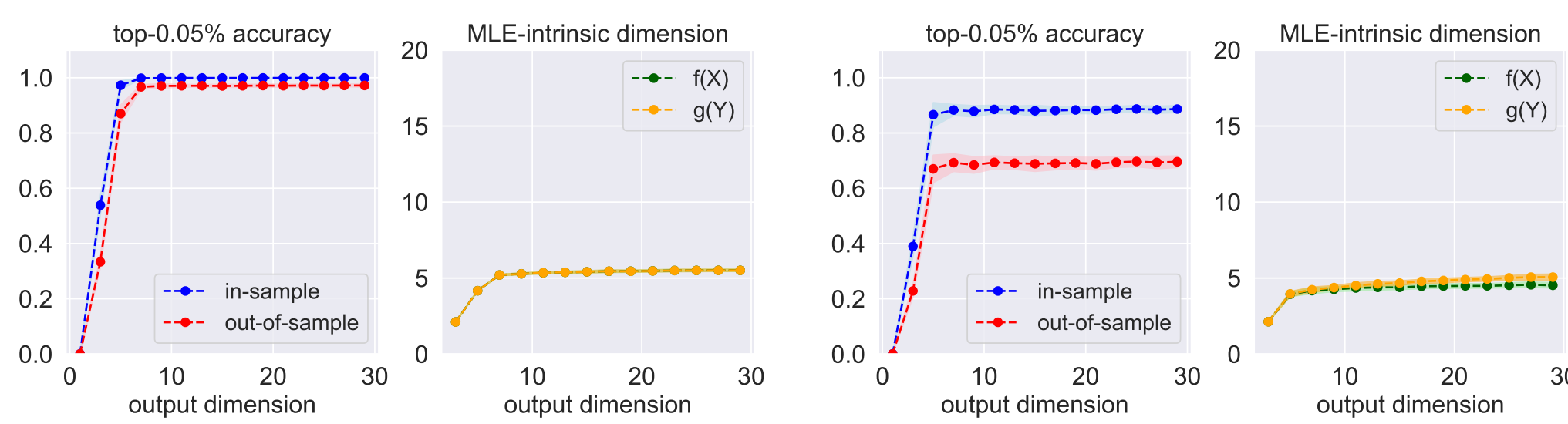

(a) Linear setting. (b) Nonlinear setting.

Figure 12: Results with synthetic dataset.

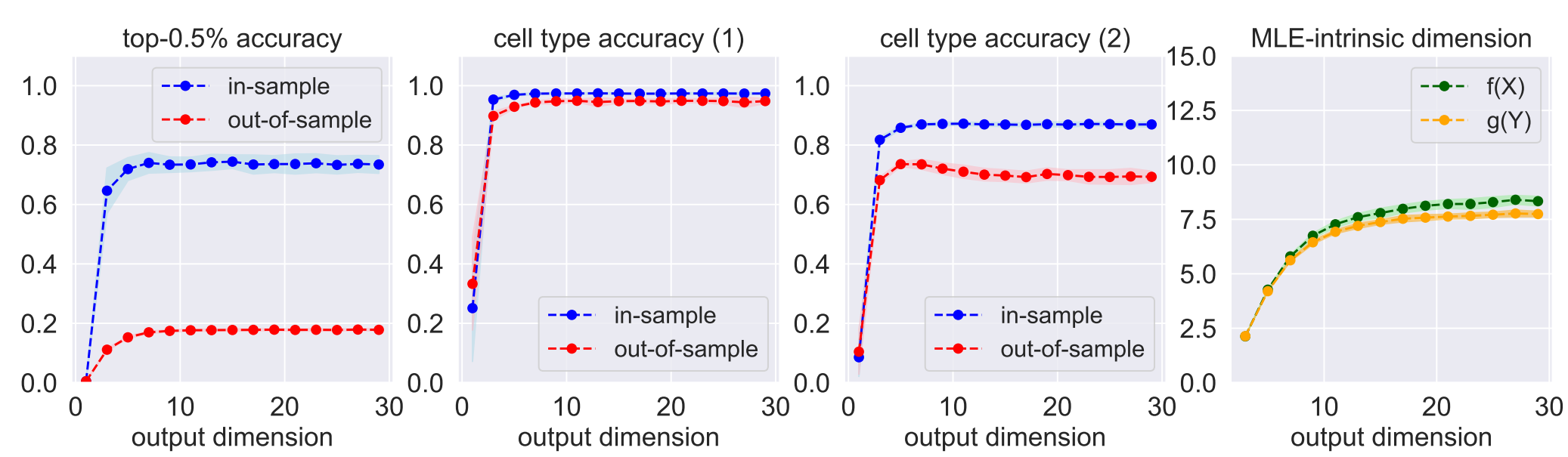

Figure 13: Results with CITE-seq dataset.

We can see that with two kinds of similarity measures, the change of estimated intrinsic dimensions with varying output dimensions is nearly the same. With the similarity measure $\sigma(\cdot,\cdot)$ adopted in

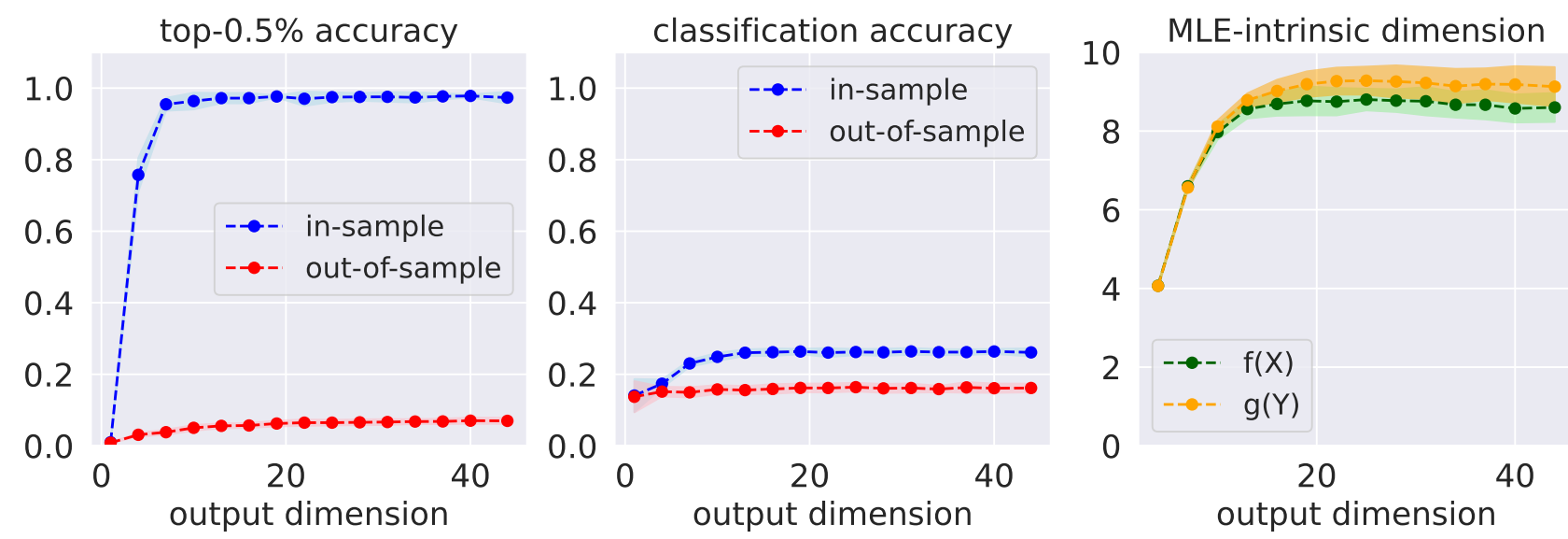

Figure 14: Results with YFCC dataset.

the paper, the top-$\alpha\%$ accuracy is even higher for synthetic data, and in the meantime, the top-$\alpha\%$ accuracy for CITE-seq dataset is higher with cosine similarity, which is partially due to the lower signal-to-noise ratio in the dataset and cosine similarity normalizes representations more strictly. In addition, for the downstream accuracy with respect to two levels of cell types, the results with the two similarity measures are comparable.

### G.6 Ablation study with a Transformer encoder

Although our theory treats the function class as fixed and is agnostic to the specification of $\mathcal{H}$, we have additionally implemented a small-scale Transformer with 2 layers, 2 heads, a feedforward dimension of 128, and a embedding dimension of $\max\{d_x, d_y\}$, topped with a final linear layer with user-specified output dimension, on CITE-seq dataset (same setting with Figure 5), the results of which are presented in the following table. From Table 2, we can see that both level-1 and level-2 accuracies averaged over 5 repetitions tend to saturate as the intrinsic dimension exceeds 15, which is similar to the results presented in Figure 5. These results indicate that our theory is agnostic to network architecture as long as the underlying architecture is sufficiently "expressive" so that the ideal representation maps can be well-approximated.

Table 2: Intrinsic dimension and cell type accuracy across output dimensions.

| Output dimension | 1 | 7 | 13 | 19 | 25 | 31 | 37 |
|---|---|---|---|---|---|---|---|
| ID of $f(X)$ | 1.00 | 5.67 | 9.79 | 12.19 | 14.15 | 15.67 | 15.97 |
| ID of $g(Y)$ | 1.00 | 5.15 | 8.63 | 10.39 | 11.75 | 12.64 | 12.96 |
| Accuracy (level-1) | 60.00% | 93.00% | 94.00% | 94.00% | 95.00% | 95.00% | 95.00% |
| Accuracy (level-2) | 22.00% | 78.00% | 79.00% | 80.00% | 81.00% | 79.00% | 81.00% |

### G.7 Results with YFCC dataset

YFCC (or YFCC100M) is a multimedia dataset consisting of images and videos as media objects with metadata including title, tags, etc [72]. We adopt a subset used by OpenAI[10], and focus on two modalities: images and text data with descriptions of images. In addition, each image in YFCC dataset is assigned a 9-level class label (farmid), which will be adopted in a downstream classification task. We follow the same procedure as that in YFCC experiments. Similar to the CITE-seq experiments, we randomly sample 10000 rows without replacement from the preprocessed dataset and randomly split the subset into a training set $\mathcal{D}_{\text{train}}$ with $|\mathcal{D}_{\text{train}}| = 8000$, a test set $\mathcal{D}_{\text{test}}$ with $|\mathcal{D}_{\text{test}}| = 1000$, and a separate dataset with size 1000 to estimate the expected norms. With learned representation maps $\widehat{f} \in \mathcal{F}_{\text{NN}}^{d_1,d}, \widehat{g} \in \mathcal{F}_{\text{NN}}^{d_2,d}$, we consider image classification and the top-$\alpha\%$ matching ($\alpha\% = 0.5$) as the downstream tasks on $\mathcal{D}_{\text{test}}$. We vary the output dimension $d$ from 1 to 29, and the results averaged after 50 repetitions are presented in Figure 14. We can see that both accuracies tend to saturate when $d$ is around 20, and the MLE-based estimation of the intrinsic dimension for the image embeddings is approximately 9 when $d$ keeps increasing. Similarly, we use a 5-layer ReLU network for CLIP training only for illustration purposes, and better results can potentially be obtained with more sophisticated network architectures.

[10] https://huggingface.co/datasets/dalle-mini/YFCC100M_OpenAI_subset.

