# OpenReview forum: "Multi-modal contrastive learning adapts to intrinsic dimensions of shared latent variables"
_NeurIPS.cc/2025/Conference — NeurIPS 2025 poster_

### Official Review · Reviewer_qwpm · 2025-07-01

**Clarity:** 3
**Significance:** 3
**Originality:** 3
**Rating:** 4
**Confidence:** 2

**Summary:**

This paper presents a thorough theoretical analysis of multi-modal contrastive learning, focusing on the properties of the learned representations beyond linear representations and specific data distributions. The authors provide a detailed investigation into how the intrinsic dimensions of shared latent variables influence the learning process and the resulting representations.

**Questions:**

- Although the author has conducted experimental verification on multiple datasets, the types and scales of these datasets are relatively limited. Could more complex models, such as those for video or audio, be tried for training?
- However, in practical applications, how can we choose appropriate model parameters (such as output dimension, temperature parameter, etc.) based on the theoretical results? Can the author provide some practical guidelines based on the theoretical analysis to help researchers and developers better apply multimodal contrastive learning methods? For example, in practical tasks, how can we adjust the output dimension according to the intrinsic dimension of the data to achieve optimal performance?

**Ethical Concerns:**

["NO or VERY MINOR ethics concerns only"]

**Limitations:**

yes

**Quality:**

3

**Strengths And Weaknesses:**

Strengths

- This paper is well written and easy to follow.
- The paper offers significant theoretical contributions by analyzing the global minimizers of the InfoNCE loss and demonstrating how multi-modal contrastive learning adapts to the intrinsic dimension of data. The proof techniques are well-presented and provide a deeper understanding of the underlying mechanisms.
- The experimental results on both synthetic and real-world datasets effectively support the theoretical findings.
- The findings have the potential to impact the design and understanding of multi-modal models. I think this paper can give insights to people.

Weakness

- In writing, clearer notation should be provided. Many symbols are not explained in detail, which reduces readability.
- The experimental part used a 5-layer ReLU network, but larger models should be tried to further illustrate the conclusions.

---

> ### Author Rebuttal · Authors · 2025-07-30
>
> We thank the reviewer for the valuable feedback, insightful questions, and constructive suggestions for improvement, and for appreciating our contribution to the understanding of multi-modal learning.
> We address weaknesses and questions as follows.
>
> - [W1: notation] Thank you for the helpful feedback! In the revision, we will (1) add a notation table clarifying all symbols and variables; (2) ensure that important notations are adequately explained.
>
> - [W2 and Q1: architectures and datasets] We appreciate the reviewer’s questions on alternative architectures and datasets, and we will address them as follows.
>     - [architectures] Although our theory treats the function class $\mathcal{H}$ as fixed and is agnostic to the specification of $\mathcal{H}$, we may expect $\mathcal{H}$ to have “expressive power” such that the set of “ideal” representation maps is not empty. We would like to clarify that the choice in experiments (i.e. the class of 5-layer ReLU MLPs) is only for illustrative purposes to justify our theory, and it turns out to be adequately powerful for datasets such as CITE-seq. In this sense, it is very meaningful to implement more powerful architectures in simulations and applications. We have additionally tried a small-scale Transformer (2 layers, 2 heads, feedforward dimension 64, and embedding dimension $\max(100, d_x, d_y)$), topped with a final linear layer with user-specified output dimension, on CITE-seq dataset (same setting with Figure 5), the results of which are presented in the following table. From the table, we can see that both level-1 and level-2 accuracies tend to saturate as the intrinsic dimension exceeds 15, which is similar to the results presented in Figure 5.
> | output dimension    | 1    | 7    | 13   | 19   | 25   | 31   | 37   |
> |---------------------|------|------|------|------|------|------|------|
> | ID of f(X)          | 1.00 | 5.67 | 9.79 | 12.19| 14.15| 15.67| 15.97|
> | ID of g(Y)          | 1.00 | 5.15 | 8.63 | 10.39| 11.75| 12.64| 12.96|
> | cell type acc (1)   | 0.60 | 0.93 | 0.94 | 0.94 | 0.95 | 0.95 | 0.95 |
> | cell type acc (2)   | 0.22 | 0.78 | 0.79 | 0.80 | 0.81 | 0.79 | 0.81 |
>
>     - [datasets] It is meaningful to evaluate our theory on more real-world, especially heterogeneous, datasets. In addition to the CITE-seq dataset presented in the main paper, we would like to note that results on image-text datasets such as ImageNetv2 and YFCC are presented in Appendix G.4, G.6, and G.7. For these datasets, we also observe the convergence of optimized temperature to zero as is shown in Appendix G.4. In our updated version, we will present more experimental results, and it would also be interesting to quantify and compare the “intrinsic dimensions” across datasets.
>
>
> - [Q2: practical guidance] Thank you for raising this insightful question. Our theoretical and experimental findings suggest that when contrastive learning is combined with temperature optimization, the learned features naturally adapt to the intrinsic dimension (ID) of the dataset. In practice, we recommend the following guideline:
> (1) Optimize the temperature parameter during training (as done in CLIP) to ensure that feature learning aligns with the data’s ID.
> (2) Start with a large output dimension and pretrain the model for a few hundred epochs to estimate the ID of the shared features. Then, iteratively (e.g., via binary search) reduce the output dimension until it is close to the estimated ID (as illustrated in Panel 4, Figure 5). This approach maintains accuracy while reducing inference cost in downstream tasks.

---

### Official Review · Reviewer_dKqv · 2025-07-01

**Clarity:** 3
**Significance:** 2
**Originality:** 2
**Rating:** 4
**Confidence:** 3

**Summary:**

This paper develops a theory of CLIP loss to analyze the efficiency of data alignment in the multi-modalities through optimizing the temperature. The main theory (theorem 2) claims that the CLIP automatically adapts to intrinsic dimensions near global minimizers, which contains four merits including the similarity maximization, the intrinsic dimension adaptation, monotonicity in temperature, and mutual information maximization. Empirical results on both synthetic and CITE-seq single-cell datasets confirm that matching accuracy and estimated intrinsic dimensions plateau once the output dimension exceeds latent dimension, validating the theory.

**Questions:**

Q1: Why the theory you care about intrinsic-dimension and temperature-tuned InfoNCE is important, and how does understanding it lead to the improvements (e.g. in embedding size or temperature scheduling) in real-world contrastive training?



Q2: Theorem 2 crucially assumes that H contains all measurable functions, yet in experiments you restrict H to finite-width 5-layer ReLU networks. How do these capacity limitations affect the intrinsic-dimension adaptation guarantee, and can you clarify the robustness of the MLE-based dimension estimator under finite samples and nonlinear mappings?


Q3: One important difference between the CLIP and the infoNCE loss is that the CLIP usually aligns two very different modalities.  How do the inherent differences between modalities (e.g. image vs. text)—impact your InfoNCE minimizer analysis and the resulting intrinsic-dimension adaptation? Could modality‐specific properties invalidate or alter the guarantees of Theorem 2?



Q4: The experiments are generally good to me, including synthetic data, a small CITE-seq protein/RNA pairing, and ImageNetV2—can the authors demonstrate intrinsic-dimension adaptation on truly heterogeneous modalities (e.g. image–text CLIP datasets)?

**Ethical Concerns:**

["NO or VERY MINOR ethics concerns only"]

**Final Justification:**

This is a good paper in multi-model.

**Limitations:**

Yes

**Quality:**

2

**Strengths And Weaknesses:**

For strengths across the dimensions of originality, quality, clarity, and significance:


Quality:


Overall, this paper contains a well setup framework to support the intrinsic-dimension adaptation in CLIP loss. Even previous papers discussed the similarity maximization, monotonicity in temperature, or mutual information maximization, this paper provides a serious proof of those conclusions via the variational decomposition of InfoNCE loss.  The authors validate their theory with experiments on both synthetic and real single-cell datasets, demonstrating consistency between empirical findings and theoretical predictions.


Clarity:


The writing is generally clear and well-structured, with helpful motivating examples and detailed appendices. But still, the structure could be further improved.


Significance:


As this paper mentioned, the purpose here is to understand “How can one efficiently integrate data from multi-modalities?” While the paper sets out to explain how to integrate truly diverse modalities, its core contribution is a formal proof that any near-minimizer of the InfoNCE loss must automatically select the correct latent dimension and drive the temperature to zero—phenomena that earlier works had observed empirically but may not rigorously established—rather than offering new training algorithms or modality-specific insights. The practical impact on real-world multi-modal training awaits further demonstration.


Originality:


The paper’s main originality lies in its variational decomposition of the InfoNCE loss that explicitly integrates temperature optimization and a novel discretization framework to define near-minimizers. This unified approach lets the authors rigorously prove that contrastive learning automatically adapts to the true latent dimension and drives the temperature to zero—results previously only observed empirically and not shown for general (nonlinear, deep‐net) encoders.


Weakness:


W1: Theorem 2 gives a careful, rigorous proof that InfoNCE minimizers must maximize alignment, adapt to the true latent dimension, and drive temperature to zero, each of these behaviors has already been empirically noted in earlier contrastive-learning work.  The paper’s main novelty is the rigor of its discretization-based argument rather than fundamentally new phenomena.


W2:  All theoretical results hinge on properties of the InfoNCE loss itself and make no use of any structure specific to language, vision, single-cell data, or other modalities. As a result, it’s unclear why this should be called a “multimodal” analysis rather than simply a general InfoNCE study.

W3: The structure could be further improved, i. e., the Lemma 8’ s variational decomposition—are referenced repeatedly across sections even before the definition 1.  Meanwhile Theorem 1 is stated informally without proof, making it feel more like an unsubstantiated observation than a formal result.

---

> ### Author Rebuttal · Authors · 2025-07-30
>
> We thank the reviewer for the valuable feedback, insightful questions, constructive suggestions for improvement, and for appreciating the novelty and quality of our theoretical analysis that incorporates temperature optimization.
> We address weaknesses and questions as follows.
>
> - [W1: novelty] We acknowledge that some of the studied empirical phenomena have been observed in prior works. Our contribution is to rigorously formalize and unify these observations through a statistical lens, showing how they arise naturally from contrastive learning via mutual information maximization and intrinsic dimension adaptation. Prior works reported isolated effects without fully clarifying the underlying mechanisms or their connections. Our theory provides the first principled justification and explanation of these links, bridging empirical findings with fundamental properties of contrastive methods. We welcome additional references on empirical findings to further strengthen the discussion in our revision.
>
> - [W2: role of multi-modality] We appreciate this insightful comment. Our framework is motivated by multimodal settings because shared latent features and their intrinsic dimension are most naturally and meaningfully defined when aligning truly different modalities (e.g., RNA–protein, image–text). This is where temperature optimization and dimension adaptation are crucial, as different modalities may have heterogeneous noise levels, making it important to adaptively select the embedding dimension.
> That said, the framework is general enough to recover single-modality contrastive learning as a special case: under a fixed augmentation strategy, the augmented data acts as a second modality, and the intrinsic dimension is determined by the augmentation-induced shared information. This demonstrates that our results apply broadly but are particularly impactful in multi-modal integration, where shared structure is less obvious and the benefits of adaptive temperature/dimension are amplified. We will revise the paper to make this motivation explicit and highlight modality-specific implications of our theory.
>
> - [W3: paper structure] Thank you for pointing this out! We agree that referencing Lemma 8’s variational decomposition prior to formally introducing Definition 1 may interrupt the logic flow. In the revised version, we will adjust the structure to introduce key definitions earlier and streamline cross-references where possible. Regarding Theorem 1, our intention was to present the core insight informally up front to convey intuition before delving into technical details. That said, we understand the concern, and in an updated version, we will point to the formal theorem to avoid confusion.
>
> - [Q1: real-world practice] Thank you for raising this insightful question. Our theoretical and experimental findings suggest that when contrastive learning is combined with temperature optimization, the learned features naturally adapt to the intrinsic dimension (ID) of the dataset. In practice, we recommend the following guideline:
> (1) Optimize the temperature parameter during training (as done in CLIP) to ensure that feature learning aligns with the data’s ID.
> (2) Start with a large output dimension and pretrain the model for a few hundred epochs to estimate the ID of the shared features. Then, iteratively (e.g., via binary search) reduce the output dimension until it is close to the estimated ID (as illustrated in Panel 4, Figure 5). This approach maintains accuracy while reducing inference cost in downstream tasks.
>
> - [Q2: capacity of function class and robustness of ID estimator] Thank you for this insightful question!
>     - [MLE-based ID estimator] The estimator was proposed in [Levina and Bickle, 2004]. This estimator is not restricted to data with linear structures, indicating that it can handle data supported on nonlinear manifolds (unlike eigenvalue-based methods such as global PCA). In addition, the asymptotic properties, including bias and variance, are also presented in [Levina and Bickle, 2004, Section 3.1]. Concretely, the estimator is unbiased and the variance decreases as the number of effective “neighbors” increases, which guarantees the finite-sample performance when the sample size is sufficiently large.
>     - [architecture] Although our theory treats the function class $\mathcal{H}$ as fixed and is agnostic to the specification of $\mathcal{H}$, we may expect $\mathcal{H}$ to have “expressive power” such that the set of “ideal” representation maps is not empty. We would like to clarify that the choice in experiments (i.e., the class of 5-layer ReLU MLPs) is only for illustrative purposes to justify our theory, and it turns out to be adequately powerful for datasets such as CITE-seq. In this sense, it is very meaningful to implement more powerful architectures in simulations and applications. We have additionally tried a small-scale Transformer (2 layers, 2 heads, feedforward dimension 64, and embedding dimension $\max(100, d_x,d_y)$), topped with a final linear layer with user-specified output dimension, on CITE-seq dataset, the results of which are presented in the following table. From the table, we can see that both level-1 and level-2 accuracies tend to saturate as the intrinsic dimension exceeds 15, which is similar to the results presented in Figure 5.
>
> | Output dimension  | 1    | 7    | 13   | 19   | 25   | 31   | 37   |
> |------------------|------|------|------|------|------|------|------|
> | ID of f(X)       | 1.00 | 5.67 | 9.79 | 12.19| 14.15| 15.67| 15.97|
> | ID of g(Y)       | 1.00 | 5.15 | 8.63 | 10.39| 11.75| 12.64| 12.96|
> | cell type acc (1)| 0.60 | 0.93 | 0.94 | 0.94 | 0.95 | 0.95 | 0.95 |
> | cell type acc (2)| 0.22 | 0.78 | 0.79 | 0.80 | 0.81 | 0.79 | 0.81 |
> - [Q3: modality-specific features] Our theory focuses on shared representation adaptation under temperature optimization. Modality-specific components require specialized losses or post-processing (see [Liang et al., 2023; Wang et al., 2024]). Their presence does not contradict our results: when shared dimension is small, adapting to it is crucial, and our method facilitates subsequent extraction of unique modality features. We will clarify this point.
>
> - [Q4: heterogeneous modalities] We agree that testing on more heterogeneous datasets is valuable. In addition to CITE-seq, we report results on image–text datasets such as ImageNetv2 and YFCC  (Appendix G.4, G.6, G.7). We will expand experiments and quantify “intrinsic dimensions” across datasets in the revision.
>
> **References**
> - Levina, Elizaveta, and Peter Bickel. "Maximum likelihood estimation of intrinsic dimension." Advances in neural information processing systems 17 (2004).
> - Liang, Paul Pu, Zihao Deng, Martin Q. Ma, James Y. Zou, Louis-Philippe Morency, and Ruslan Salakhutdinov. "Factorized contrastive learning: Going beyond multi-view redundancy." Advances in Neural Information Processing Systems 36 (2023): 32971-32998.
> - Wang, Chenyu, Sharut Gupta, Xinyi Zhang, Sana Tonekaboni, Stefanie Jegelka, Tommi Jaakkola, and Caroline Uhler. "An information criterion for controlled disentanglement of multimodal data." arXiv preprint arXiv:2410.23996 (2024).

---

> > ### Comment · Reviewer_dKqv · 2025-08-04
> >
> > I thank the authors for the rebuttal. The authors have addressed my concerns.

---

### Official Review · Reviewer_qdkz · 2025-07-02

**Clarity:** 4
**Significance:** 2
**Originality:** 2
**Rating:** 4
**Confidence:** 3

**Summary:**

This paper provides a theoretical analysis of CLIP-style multi-modal contrastive learning objectives. The authors show that, when temperature is optimized (like in CLIP), the InfoNCE loss not only aligns representations across modalities but also automatically adapts to the true intrinsic dimension of the shared latent variables. They formalize ideal representations via alignment and maximal mutual information (Definitions 1–2) and intrinsic dimension (Definition 3). Consequently in Theorem 2 show that any approxmate minimizers of the InfoNCE achieves similarity maximization, exact intrinsic-dimension adaptation, monotonicity in temperature, and maximal mutual information, even without requiring representations to be uniformly distributed over the entire output space. Empirical results on synthetic datasets and on a CITE-seq single-cell dataset confirm that downstream accuracy and estimated intrinsic dimensions saturate once the model’s output dimension exceeds the true dimension

**Questions:**

* Q1. Can the authors characterize finite-sample generalization for InfoNCE minimization? In particular, how many samples are needed to guarantee, with high probability, similarity maximization and intrinsic-dimension adaptation?
* Q2. Empirical work reveals a persistent modality gap in CLIP-like models (few are cited in weakness section); can the V(H)≠∅ assumption be relaxed to account for approximate alignment?
* Q3. Theorem 2 is proven under inner-product similarity and abstract hypothesis classes—how do these guarantees translate when using cosine similarity and modern architectures?
* Q4. Many applications employ noisy pairs or one-to-many pairings, and multi-view data; can the analysis be extended to these settings, and under what conditions does intrinsic-dimension adaptation still hold in the presence of label noise or weak alignments?

I would be happy to raise my rating if the authors satisfactorily address my concerns.

**Ethical Concerns:**

["NO or VERY MINOR ethics concerns only"]

**Final Justification:**

I found the new experiments on the modality gap especially insightful. I believe these additions would strengthen the paper.  I’m happy to update my score to borderline accept

**Limitations:**

yes

**Paper Formatting Concerns:**

Not a major concern, and I’m not certain, but some lines with inline formulas seem much more closely spaced than they should be (e.g. lines 221-222, 228-229)

**Quality:**

3

**Strengths And Weaknesses:**

## Strengths
* S1. The paper extends prior theoretical work on contrastive learning by explicitly incorporating temperature τ optimization into the analysis, explaining convergence as τ→0 and its role in intrinsic-dimension adaptation.
* S2. Formal definitions of ideal representations and intrinsic dimension (Definitions 1–3) create a unified framework linking alignment, mutual information maximization, and latent representation complexity.
* S3. Theorem 2 provides rigorous, population-level guarantees, including similarity maximization, exact intrinsic-dimension adaptation, temperature monotonicity, and maximal mutual information

## Weaknesses
* W1. The analysis assumes access to population-level objectives and infinite data; finite-sample effects and generalization bounds are not addressed
* W2. The requirement V(H)≠∅ assumes the existence of perfectly aligned, maximally informative representations across modalities, yet empirical studies reveal a persistent modality gap in CLIP-like model, i.e. a systematic geometric separation between image and text embeddings that violates this assumption and undermines practical applicability [1, 2, 3]
* W3. Experimental evaluation is limited to a 5-layer ReLU network and one biological dataset; broader benchmarks (e.g., large-scale image-text like COCO or CC3M) would strengthen the claims.
* W4. The impact of different similarity measures (inner product vs. cosine) and architectural choices (e.g., transformers vs. MLPs) is only briefly mentioned.
* W5. The paper does not explore how this theory extends to multi-class or noisy pairing scenarios common in real-world contrastive pretraining.

[1] Mind the Gap: Understanding the Modality Gap in Multi-modal Contrastive Representation Learning
[2] It's Not a Modality Gap: Characterizing and Addressing the Contrastive Gap
[3] Cross the Gap: Exposing the Intra-modal Misalignment in CLIP via Modality Inversion

---

> ### Author Rebuttal · Authors · 2025-07-30
>
> We thank the reviewer for the valuable feedback and constructive suggestions, and for appreciating our analysis of temperature optimization and the rigor of our theory that links similarity maximization and mutual information distillation. We address weaknesses and questions below.
>
> - [finite-sample analysis: W1 & Q1] To understand the effect of sample size, from the empirical viewpoint, we conduct an ablation study in the same setting as Figure 3 (with a fixed output dimension 20) with varying sample size $N_{\rm train}$ from 2000 to 14000. We present the estimated IDs, in-sample top-1 accuracy, and out-of-sample top-1 accuracy in the following table, from which we can see that as $N_{\rm train} \geq 10000$, these quantities begin to saturate, which validates that there is indeed a lower bound of $N_{\rm train}$ to ensure good properties. To answer this question, it is also relevant to the convergence analysis of infoNCE loss in [Wang and Isola (2020), Theorem 1 (3)]. When we consider compact sets of maps and temperature (e.g., with a boundedness condition), we expect to derive the uniform law of large numbers for the finite-sample loss. In the updated version, we will add a detailed discussion on the convergence analysis below Lemma 8.
> | sample size       | 2000 | 4000 | 6000 | 8000 | 10000 | 12000 | 14000 |
> |-------------------|------|------|------|------|-------|--------|--------|
> | ID of f(X)        | 8.50 | 6.80 | 5.62 | 5.18 | 4.76  | 4.53   | 4.45   |
> | ID of g(Y)        | 8.54 | 6.88 | 5.81 | 5.46 | 5.07  | 4.87   | 4.71   |
> | in-sample ACC     | 0.38 | 0.75 | 0.91 | 0.95 | 0.96  | 0.96   | 0.96   |
> | out-of-sample ACC | 0.08 | 0.44 | 0.74 | 0.81 | 0.87  | 0.90   | 0.91   |
>
> - [modality gap and $\mathcal{V}(\mathcal{H})$: W2 & Q2] Thank you for pointing this out and for very helpful references! We will definitely add these references and related discussions on the modality gap to our updated version. Regarding the assumption on $\mathcal{V}(\mathcal{H})$, we agree that it is an idealized statistical setting, in which we could sharply understand the performance of multi-modal contrastive learning, and the established theory can also explain the observed phenomena, especially the convergence of temperature and the revealed intrinsic dimension, in some real-world datasets (e.g. CITE-seq and ImageNet, additional evidence is presented in Appendix G.4 and G.6). Also, the modality gap is indeed a crucial phenomenon in contrastive learning. To first see this empirically, we conduct an ablation study for the setting in Figure 2, where we add entrywise iid centered Gaussian noise to the first three components of X and Y, respectively (variance is set to be $\sigma^2$ and $N_{\rm train}=10000$). Empirical studies on the modality gap defined by $|| n^{-1} \sum_{i \leq n} f(X_i) - g(Y_i) ||$ with $n$ test samples [Liang et al., 2022] in the table show the following:
>
>     - (1) When $\sigma \leq 0.2$, both the estimated ID, the similarity maximization phenomenon, and top-K accuracies are comparable to those with $\sigma=0$, indicating that our theory can be robust when the condition is approximately met. This will imply the future work on the robustness analysis of our framework, and we will add a detailed discussion on this in an updated version.
>     - (2) The computed modality gap increases with $\sigma$, and its magnitude becomes more pronounced when $\sigma > 0.4$. When $\sigma = 0$, the tiny modality gap may be an artifact of the model training process, as it depends on the number of training epochs and step size scheduling. Hence, our setting with non-empty $\mathcal{V}(\mathcal{H})$ can be viewed as a baseline scenario, and the modality gap can, in this sense, quantify the difficulty in multi-modal learning, which also motivates future theoretical work on the robustness analysis.
>     | $\sigma$                                      | 0     | 0.2   | 0.4   | 0.6   | 0.8   | 1.0   |
>     |----------------------------------------------|-------|-------|-------|-------|-------|-------|
>     | modality gap                                  | 0.056 | 0.078 | 0.075 | 0.105 | 0.159 | 0.178 |
>     | modality gap with self-normalized representations | 0.030 | 0.047 | 0.044 | 0.065 | 0.130 | 0.134 |
>
>
>
>
>
> - [alternative similarities and architectures: W3 & W4 & Q3]
>     - [similarities] To begin with, as shown in Figure 11 and 12, the empirical performance with cosine similarity and the similarity adopted in our paper are comparable. In addition, we would like to note that the key variational decomposition in Lemma 8, as well as the results in Lemma 10, are agnostic to the similarity measure and also apply to cosine similarity. Concretely, with a general similarity (including cosine similarity, or, equivalently, the function class maps to the unit hypersphere), as we have stated in Lemma 10 in Appendix C.2, we can also show that
>         - (1) with any $(f,g) \in \mathcal{V}(\mathcal{H})$, $L(f,g,\tau)$ is non-decreasing in $\tau$ and  $-L(f,g,\tau)/2$ converges to the maximum mutual information as $\tau \rightarrow 0$;
>         - (2) Reversely, by Lemma 10, for any $(f,g)$ as the approximate minimizer, similar to the proof in Appendix C 2.2, we can show that $(f,g)$ must maximize both mutual information and similarity, and $L(f,g,\tau)$ is non-decreasing in $\tau$. In addition, for intrinsic dimension adaptation with other similarities, specific technical tools may be needed. As a concrete example, with the cosine similarity, tools from potential minimization and geodesic projections on a hypersphere could be promising to obtain results similar to those in Appendix D.
>         - To sum up, the current similarity measure is a choice that has both comparable empirical performance with cosine similarity and elegant theoretical guarantees. We will add a more detailed discussion on variants of similarity measures in the revision.
>
>     - [architectures] Although our theory treats the function class $\mathcal{H}$ as fixed and is agnostic to the specification of $\mathcal{H}$, we expect $\mathcal{H}$ to have “expressive power” such that the set of “ideal” representation maps is not empty. We would like to clarify that the choice in experiments (i.e. the class of 5-layer ReLU MLPs) is only for illustrative purposes to justify our theory, and it turns out to be adequately powerful for datasets such as CITE-seq. In this sense, it is very meaningful to implement more powerful architectures in simulations and applications. We have additionally tried a small-scale Transformer (2 layers, 2 heads, feedforward dimension 64, and embedding dimension $\max(100, d_x, d_y)$), topped with a final linear layer with user-specified output dimension, on CITE-seq dataset (same setting with Figure 5), the results of which are presented in the following table. From the table, we can see that both level-1 and level-2 accuracies tend to saturate as the intrinsic dimension exceeds 15, which is similar to the results presented in Figure 5. These results indicate that our theory is agnostic to network architecture as long as the underlying architecture is sufficiently “expressive” so that the ideal representation maps can be well-approximated.
>      | output dimension    | 1    | 7    | 13   | 19   | 25   | 31   | 37   |
>      |---------------------|------|------|------|------|------|------|------|
>      | ID of f(X)          | 1.00 | 5.67 | 9.79 | 12.19| 14.15| 15.67| 15.97|
>      | ID of g(Y)          | 1.00 | 5.15 | 8.63 | 10.39| 11.75| 12.64| 12.96|
>      | cell type acc (1)   | 0.60 | 0.93 | 0.94 | 0.94 | 0.95 | 0.95 | 0.95 |
>      | cell type acc (2)   | 0.22 | 0.78 | 0.79 | 0.80 | 0.81 | 0.79 | 0.81 |
>
>
>
> - [imperfect pairing: W5 & Q4] We agree that extending our results to settings with noisy pairs, one-to-many pairings, or multi-view data is an interesting and important future direction.
>     - For the noisy pairing, the simple simulation in response to the modality gap may offer us some useful insight: (1) Within a moderate range of perturbations of matched features, contrastive learning tend to be robust in revealing the ideal shared features; (2) When the perturbation is large, the modality gap in learned representations cannot be ignored, in which case, either new algorithms or advanced technical tools ought to be designed. Moreover, noisy pairing may also be related to multi-modal data with shared and modality-specific features mentioned by Reviewer dKqv.
>     - For the one-to-many scenario (e.g., for the pair $(X,Y)=({\rm image}, {\rm caption})$, suppose there are additional/augmented pairs taking the form $(X,Y_{\rm aug})$ in the training set [Waseda et al., 2025]), if our goal is still to identify shared features and to ensure that our theory applies, one shortcut solution is to predefine an integration map $T_y$ (e.g., early fusion for the text modality), and define $T_y(Y,Y_{\rm aug})$ as the new inputs. Then, if the augmentation maps are invertible (e.g., permutations), if $V(H) \neq \varnothing$ is true for each (image, caption) pair, then it can still hold for $(X, T_y(Y,Y_{\rm aug}))$ in this simple setting. If more general scenarios with one-to-many and many-to-one pairs are of interest, more sophisticated statistical models should be introduced and a fine-grained set $\mathcal{V}(\mathcal{H})$ should be considered. It will be a compelling future direction to theoretically derive the theoretical guarantee in this setting to better understand the advantage of one-to-many data/augmentations in multi-modal learning. We will add detailed discussions on this aspect in the revision.
>
> **References**
> - Liang, Victor Weixin, Yuhui Zhang, Yongchan Kwon, Serena Yeung, and James Y. Zou. "Mind the gap: Understanding the modality gap in multi-modal contrastive representation learning."
> - Waseda, Futa Kai, Antonio Tejero-de-Pablos, and Isao Echizen. "Leveraging One-To-Many Relationships in Multimodal Adversarial Defense for Robust Image-Text Retrieval."

---

> > ### Comment · Reviewer_qdkz · 2025-08-05
> >
> > Thank you for the thoughtful and thorough response. I found the new experiments on the modality gap especially insightful and I appreciated the clear clarifications. I believe these additions would strengthen the paper, and I’m happy to update my score.

---

### Official Review · Reviewer_5L7J · 2025-07-02

**Clarity:** 3
**Significance:** 3
**Originality:** 4
**Rating:** 5
**Confidence:** 3

**Summary:**

This paper presents a novel theoretical analysis of multi-modal contrastive learning, particularly inspired by the success of models like CLIP. The authors investigate the properties of the learned representations when the contrastive loss function (infoNCE) is minimized jointly over the representation functions and the temperature parameter. The central claim is that this process not only maximizes the mutual information between modalities but also automatically adapts the learned representations to the intrinsic dimension of the shared latent variables, even when the user-specifies a higher-dimensional output space. The theoretical findings are supported by a comprehensive set of experiments on both synthetic and real-world datasets (CITE-seq, ImageNetV2, YFCC), demonstrating that the learned representations capture low-dimensional structures that are effective for downstream tasks.

**Questions:**

None

**Ethical Concerns:**

["NO or VERY MINOR ethics concerns only"]

**Final Justification:**

Based on the discussion with the authors, the updates will strengthen the overall contribution of the paper.

**Limitations:**

Yes.

**Quality:**

4

**Strengths And Weaknesses:**

Strengths:

1. The theoretical underpinnings of large-scale, self-supervised multi-modal models are a crucial and timely area of research. The paper astutely identifies a key phenomenon observed in practice—the learning of low-dimensional representations—and correctly points out the inability of existing theoretical frameworks (based on uniformity or simple mutual information maximization) to fully explain it.

2. By analyzing the infoNCE loss with an optimized temperature, it provides a compelling explanation for three key phenomena:
i. Similarity Concentration: Positive pairs achieve maximal similarity, while negative pairs are capped.
ii. Intrinsic Dimension Adaptation: The learned representations converge to the true intrinsic dimension of the shared information. This is the core contribution and a significant step beyond prior work which often required the true dimension to be known beforehand.
iii. Temperature Convergence: The optimization naturally drives the temperature parameter (τ) towards zero , which aligns with empirical observations.

3. The authors build their theory on a solid foundation. The introduction of the set of "ideal representations" V(H)—defined as those that achieve both alignment and maximal mutual information—is elegant. The use of a discretization argument to rigorously handle the mutual information of continuous representations is technically sound and allows them to sidestep the issue of infinite mutual information for aligned representations.

4. Comprehensive and Convincing Empirical Validation: The experiments are exceptionally well-designed to support the theoretical claims.
i. Synthetic Data: By using synthetic data with a known ground-truth intrinsic dimension (k*), the authors clearly demonstrate that both downstream accuracy and the estimated intrinsic dimension saturate at k*, regardless of the user-specified output dimension (provided d≥k*). This provides direct, unambiguous evidence for their central claim.
ii. Real-World Data: The application to diverse, real-world datasets like CITE-seq , ImageNetV2 , and YFCC  shows that these phenomena are not just artifacts of a synthetic setup but are present in practical applications.
iii. Ablation Studies: The appendix provides crucial supporting evidence. The experiment in Appendix G.2, which shows that minimizers are not meaningful when the central assumption (V(H) is not ∅) is violated, is particularly insightful and strengthens the paper by clearly defining the scope and limitations of the theory.

Weaknesses:
1. The entire theoretical framework rests on the non-emptiness of V(H)—the assumption that a representation simultaneously achieving perfect alignment and maximal mutual information exists within the chosen function class H. This is a strong assumption. While the paper does an excellent job justifying it by connecting the convergence of temperature to zero in experiments to this condition, the link could be made more explicit. A deeper discussion on what properties of a function class (e.g., a sufficiently deep/wide neural network) and dataset would lead to this assumption holding would further strengthen the paper's practical implications.

2. The definition of the infoNCE loss minimizer, using the set of near-minimizers O L,η(H) and the temperature threshold ϵ(η), is technically necessary to handle the asymptotic nature of the problem (τ→0). However, it is quite dense and may be difficult for readers to parse. While intuition is provided, a more streamlined or high-level explanation in the main text of Section 3.1 could improve readability and make the core results more accessible.

Overall: This is a high-quality paper that makes a significant and novel contribution to our understanding of multi-modal contrastive learning. It successfully bridges theory and practice by providing a rigorous explanation for the empirically observed phenomenon of intrinsic dimension adaptation. The claims are well-supported by elegant theory and validated by an extensive set of well-conducted experiments. While the core assumption is strong, the authors are transparent about it and provide compelling evidence for its relevance. This work will be of great interest to researchers in self-supervised learning, representation learning, and multi-modal AI.

---

> ### Author Rebuttal · Authors · 2025-07-30
>
> We thank the reviewer for the positive feedback on our work, especially the rigor of our theoretical analysis, and the comprehensiveness of our empirical validation. Below, we address the weaknesses you identified.
>
> - [W1: assumption on $\mathcal{V}(\mathcal{H})$] Thank you for pointing this out!
>     - Regarding the assumption on $\mathcal{V}(\mathcal{H})$, we agree that it is an idealized statistical setting, in which we could sharply understand the performance of multi-modal contrastive learning, and the established theory can also explain the observed phenomena, especially the convergence of temperature and the revealed intrinsic dimension, in some real-world datasets (e.g. CITE-seq and ImageNet, additional evidence is presented in Appendix G.4 and G.6).
>     - We agree with the reviewer that although our theory treats the function class $\mathcal{H}$ as fixed and is agnostic to the specification of $\mathcal{H}$, we may expect $\mathcal{H}$ to have “expressive power” such that the set of “ideal” representation maps is not empty. The choice in experiments (i.e. the class of 5-layer ReLU MLPs) is only for illustrative purposes to justify our theory, and it turns out to be adequately powerful for datasets such as CITE-seq. In this sense, it is very meaningful to implement more powerful architectures in simulations and applications. We have additionally tried a small-scale Transformer (2 layers, 2 heads, feedforward dimension 64, and embedding dimension $\max(100, d_x, d_y)$), topped with a final linear layer with user-specified output dimension, on CITE-seq dataset (same setting with Figure 5), the results of which are presented in the following table. From the table, we can see that both level-1 and level-2 accuracies tend to saturate as the intrinsic dimension exceeds 15, which is similar to the results presented in Figure 5. These results indicate that our theory is agnostic to network architecture as long as the underlying architecture is sufficiently “expressive” so that the ideal representation maps can be well-approximated.
> | output dimension    | 1    | 7    | 13   | 19   | 25   | 31   | 37   |
> |---------------------|------|------|------|------|------|------|------|
> | ID of f(X)          | 1.00 | 5.67 | 9.79 | 12.19| 14.15| 15.67| 15.97|
> | ID of g(Y)          | 1.00 | 5.15 | 8.63 | 10.39| 11.75| 12.64| 12.96|
> | cell type acc (1)   | 0.60 | 0.93 | 0.94 | 0.94 | 0.95 | 0.95 | 0.95 |
> | cell type acc (2)   | 0.22 | 0.78 | 0.79 | 0.80 | 0.81 | 0.79 | 0.81 |
> We will add a detailed discussion on this condition and its connection with architectures and datasets.
>
> - [W2: readability of Section 3.1] Thank you for your helpful suggestions on exposition! To improve clarity and readability, we will revise Section 3.1 in the updated version to include a more intuitive and high-level explanation before presenting the formal definitions. This will highlight the core intuition behind our asymptotic analysis in the regime as $\tau$ goes to zero and better motivate the role of the approximate minimizers that we defined.

---

> > ### Comment · Reviewer_5L7J · 2025-08-05
> > **Response to Author Rebuttal**
> >
> > Thank you for the clear and constructive rebuttal. The additional experiments with the Transformer architecture meaningfully support your claims and help contextualize the assumption on V(H). I also appreciate your plans to revise Section 3.1 for improved clarity—this will significantly aid accessibility. These updates strengthen the overall contribution of the paper.

---

### Comment · Area_Chair_kxbZ · 2025-08-04
**Author-reviewer discussion period ends soon**

Dear reviewers,

Author-reviewer discussion period ends soon. Please check the rebuttals and take an appropriate action.

AC

---

### Decision · Program_Chairs · 2025-09-17

**Decision:**

Accept (poster)

**Comment:**

The paper provides a theoretical analysis of multi-modal contrastive learning, focusing on the InfoNCE loss when jointly optimizing the representation functions and temperature. The central claim is that this setup not only maximizes cross-modal mutual information but also adapts to the intrinsic dimension of the shared latent variables, even when the output dimension is larger.

All reviewers agree that this work should be accepted. While the limitations of "alignment and uniformity" and "MI maximization" have been known, prior theoretical analyses have been insufficient. This paper makes a meaningful theoretical contribution on this topic, and given the scarcity of such contributions, the work is valuable. Some reviewers noted the lack of direct empirical improvements based on the findings, but also agreed that empirical gains are not necessary for this type of contribution.

Therefore, acceptance is recommended.